# Decidualization-empowered ECM hydrogel integrating sustained Tβ4 release drives endometrial regeneration in intrauterine adhesions

Yuxiang Liang [1,2] ✉, Zhaowei Yu[1], Shaobo Du[1], Yuqian Guo[1], Jing Li[1], Yujia Yan[1], Shanshan Jin[1], Wenjing Liang[1], Mengyuan Li[1], Ning Jin[1], Jiao Yang[1], Zhiwei Peng[1], Zhaoyang Chen[2], Hailan Yang[3], Zhizhen Liu[1] ✉, Qizhi Shuai[1] ✉, Liping Li[1] ✉ & Jun Xie [1] ✉

Intrauterine adhesions (IUA), a leading cause of female infertility, result from a pathological switch in the uterine injury response from regeneration to fibrotic scarring. Current treatments are often inadequate as they fail to address this fundamental shift. Here, we report a "decidualization-empowered" hydrogel that reverses this pathology by synergistically combining a bioactive extracellular matrix from decidualized endometrium (DEndo-UdECM) with the sustained release of the anti-fibrotic peptide Thymosin β4 (Tβ4). In a murine IUA model, a single administration of the hydrogel restores endometrial architecture, resolves fibrosis, and, most critically, leads to a near-complete recovery of fertility. Mechanistically, the hydrogel orchestrates a proregenerative niche by reprogramming macrophages to an M2 phenotype while dually inhibiting pyroptosis-driven inflammation and the canonical TGF-β/Smad3 fibrotic cascade. Together, these findings demonstrate that mimicking the biological intelligence of a physiological niche can resolve complex fibrotic diseases and restore organ function.

Intrauterine adhesions (IUA), characterized by fibrotic scarring of the endometrium, represent a major cause of female infertility and a formidable clinical challenge[1]. With incidence rates exceeding 30% in high-risk populations, it poses a significant global health burden, culminating in debilitating outcomes including menstrual disorders, recurrent pregnancy loss, and infertility[2,3]. Current treatments, primarily surgical resection, even when supplemented with physical barriers or hormonal therapies, are plagued by high rates of recurrence (often >60%) and fail to restore fertility in most cases[2,4]. This therapeutic impasse highlights a fundamental flaw in existing approaches: they do not address the underlying pathology—a pathological switch in the uterine microenvironment from a proregenerative to a pro-fibrotic state.

The persistence of IUA is driven by a self-sustaining vicious cycle of inflammation and fibrosis[5,6]. Following the initial injury, potent inflammatory cascades are initiated, involving NLRP3 inflammasome activation, M1 macrophage polarization, and excessive cytokine secretion.[7] This inflammatory milieu propagates detrimental cell

[1]Department of Biochemistry and Molecular Biology, Shanxi Key Laboratory of Birth Defect and Cell Regeneration, MOE Key Laboratory of Coal Environmental Pathogenicity and Prevention, Shanxi Medical University, Taiyuan, Shanxi, China. [2]Shanxi Key Laboratory of Human Disease and Animal Models, Experimental Animal Center, Shanxi Medical University, Taiyuan, Shanxi, China. [3]Department of Obstetrics and Gynecology, The First Affiliated Hospital of Shanxi Medical University, Taiyuan, Shanxi, China. ✉e-mail: liangyuxiang@sxmu.edu.cn; zhizhenliu2013@163.com; shuaiqizhi@sxmu.edu.cn; liping_8103@163.com; junxie@sxmu.edu.cn

death pathways, such as pyroptosis[8–10], and fuels the aberrant activation of the canonical TGF-β1/Smad3 pathway, which orchestrates excessive collagen deposition and matrix stiffening[11,12]. Ultimately, this process culminates in the formation of a non-permissive scar tissue that physically and biochemically impedes neo-angiogenesis, hinders the function of endometrial stem/progenitor cells, and obstructs functional regeneration[3,13,14].

Addressing the dual challenge of promoting regeneration while suppressing fibrosis demands an integrated biomaterial strategy[3,14]. However, current approaches are often too simplistic. Many function merely as passive physical barriers or deliver single growth factors, failing to address the multifaceted pathology[15–17], while simple hydrogels are limited by their inherent bio-inertness[6,18]. Decellularized extracellular matrix (dECM) hydrogels represent an advance by retaining intrinsic bioactivity from the source tissue[19,20]. While even tissue-specific uterine dECM (UdECM) has shown promise[21–25], such scaffolds are derived from a quiescent state and thus may lack the potent, specialized cues of the dynamically remodeling, pro-regenerative microenvironment required for true functional restoration.

Nature provides a compelling blueprint for scarless endometrial repair in physiological decidualization, a process that establishes a unique pro-regenerative and immunomodulatory milieu[12,26–28]. While prior studies inspired by this phenomenon have explored the therapeutic potential of decidual cells or their soluble derivatives[29–31], translating cell-based therapies faces clinical hurdles, and these approaches may not fully leverage the persistent cues embedded within the insoluble matrix framework[32,33]. We therefore hypothesized that a decellularized ECM derived directly from in vitro decidualized endometrium (DEndo-UdECM) could capture this unique, pro-regenerative matrix. This would offer a superior, cell-free, "decidualization-empowered" biomaterial that leverages the persistent physical and biochemical cues of the insoluble scaffold to drive functional repair.

However, to overcome the potent fibrotic drive in severe IUA, we reasoned that this pro-regenerative scaffold required a synergistic partner: a sustained anti-fibrotic agent. For this, we selected Thymosin β4 (Tβ4), a pleiotropic peptide known not only as a potent inhibitor of TGF-β signaling but also for its pro-angiogenic and immunomodulatory activities[34,35]. To overcome its short biological half-life, we engineered its prolonged delivery via PLGA microspheres to provide sustained therapeutic pressure[36]. This creates a dual-action system designed for a synergistic "division of labor": the DEndo-UdECM scaffold provides the immediate pro-regenerative blueprint, while Tβ4@PLGA acts as a sustained brake on fibrosis.

Here, we develop and validate a composite hydrogel that synergistically integrates the bioactive DEndo-UdECM scaffold with sustained Tβ4 release. We show that in a stringent murine IUA model, a single administration of this hydrogel not only restores endometrial architecture and resolves fibrosis but, most critically, leads to a near-complete recovery of fertility as summarized in Fig. 1. We further elucidate the underlying mechanisms, demonstrating that the hydrogel orchestrates a pro-regenerative microenvironment by reprogramming macrophages, dually inhibiting pyroptosis and TGF-β signaling, and restoring the molecular signature of endometrial receptivity.

## Results

### Establishment and characterization of a mouse uterine decidualization model

To obtain an ECM reflecting a dynamic uterine microenvironment, we first established a physiologically relevant mouse model of decidualization[37]. Pseudopregnant mice received intrauterine sesame oil injection on day 4 post-coitus, and uteri were harvested on day 8 (Supplementary Fig. 1a). Decidualization induced marked structural

changes, including pronounced bilateral uterine horn distension (Supplementary Fig. 1b). Quantitative analysis revealed significant increases in uterine length (2.516 vs 1.280 cm, $P < 0.0001$), area (2.296 vs 0.801 cm², $P < 0.0001$), and weight (814.91 vs 169.68 mg, $P < 0.0001$) in the decidualized (Dec) group compared to controls (Supplementary Fig. 1c–f). Molecular assessment employed markers for angiogenesis (CD31)[38], hormone response (Progesterone Receptor, PR)[39], stromal differentiation (HAND2)[40], and proliferation (PCNA). Immunofluorescence (IF) staining revealed distinct changes: CD31 expression showed an elevated and continuous pattern in decidualized uterine vessels (75.14% positive area fraction vs controls, $P < 0.0001$); PR displayed increased nuclear intensity in decidualized stromal cells ($P < 0.0001$); the PCNA-positive area fraction was significantly higher in the Dec group (12.56% vs 1.71%, $P < 0.0001$); and HAND2 expression was also significantly upregulated ($P < 0.0001$) (Supplementary Fig. 1g–j). These morphological and molecular changes confirmed the successful establishment of the mouse uterine decidualization model.

### Characterization of DEndo-UdECM scaffolds

Following model establishment, decellularized ECM scaffolds derived from decidualized endometrium (DEndo-UdECM) were prepared and systematically characterized. Macroscopically, DEndo-UdECM scaffolds appeared translucent white, contrasting with the opaque native tissue, while retaining the gross 3D architecture (Fig. 2a). Histological analysis using Hematoxylin and Eosin (H&E) staining demonstrated the absence of nuclear remnants in DEndo-UdECM (Fig. 2b). While collagen fiber density appeared slightly reduced visually compared to native tissue, the overall fibrillar architecture was maintained (Fig. 2b). Masson's trichrome staining confirmed the preservation of collagen fiber spatial organization (Fig. 2c), and quantitative analysis showed no significant difference in the collagen area fraction between native decidualized tissue and DEndo-UdECM ($P > 0.05$) (Fig. 2d). Effective decellularization was confirmed by a > 98% reduction in DNA content in DEndo-UdECM scaffolds (17.22 ± 4.940 ng/mg) compared to native tissue (1259.07 ± 216.582 ng/mg; $P < 0.0001$) (Fig. 2e). Key ECM components (Collagen I, Collagen IV, Fibronectin) were assessed by IF[21]. Compared to native non-decidualized tissue, decidualization significantly increased the area fraction of Collagen I (by ~20%; $P < 0.05$; Fig. 2f) and Collagen IV (by ~30%; $P < 0.01$; Fig. 2g). Fibronectin showed a non-significant upward trend in decidualized tissue ($P > 0.05$; Fig. 2h). The decellularization process resulted in a slight, non-significant decrease in the fluorescence intensity of these three ECM proteins ($P > 0.05$). The absence of nuclei in DEndo-UdECM was further confirmed by negative DAPI staining (Fig. 2f–h).

To elucidate compositional distinctions, quantitative proteomic analysis compared uterine decellularized ECM (UdECM) from control (CEndo-UdECM) and decidualized (DEndo-UdECM) endometrium. Proteomic profiling identified 198 proteins unique to CEndo-UdECM and 300 unique to DEndo-UdECM, with 8513 proteins common to both (Fig. 3a). Alignment with the Matrisome database identified 20 CEndo-specific, 24 DEndo-specific, and 348 shared matrisome proteins (Fig. 3b). Differential abundance analysis revealed significant matrisome remodeling during decidualization, with 99 proteins significantly upregulated and 126 significantly downregulated in DEndo-UdECM relative to CEndo-UdECM (Fig. 3c; adjusted $p$-value < 0.05, Fold Change > 1.5). Classification into core matrisome (Collagens, Glycoproteins, Proteoglycans) and matrisome-associated categories highlighted compositional shifts (Fig. 3d). DEndo-UdECM showed enrichment in the relative abundance of ECM regulators (13.4% vs 9.9%), Secreted factors (11.6% vs 3.4%), and Glycoproteins (20.9% vs 14.3%), while CEndo-UdECM maintained higher relative proportions of Collagens (42.3% vs 31.4%), Proteoglycans (11.5% vs 5.7%), and ECM-affiliated proteins (18.8% vs 17.1%). Analysis of the top 10 most abundant ECM proteins revealed shared structural elements and unique

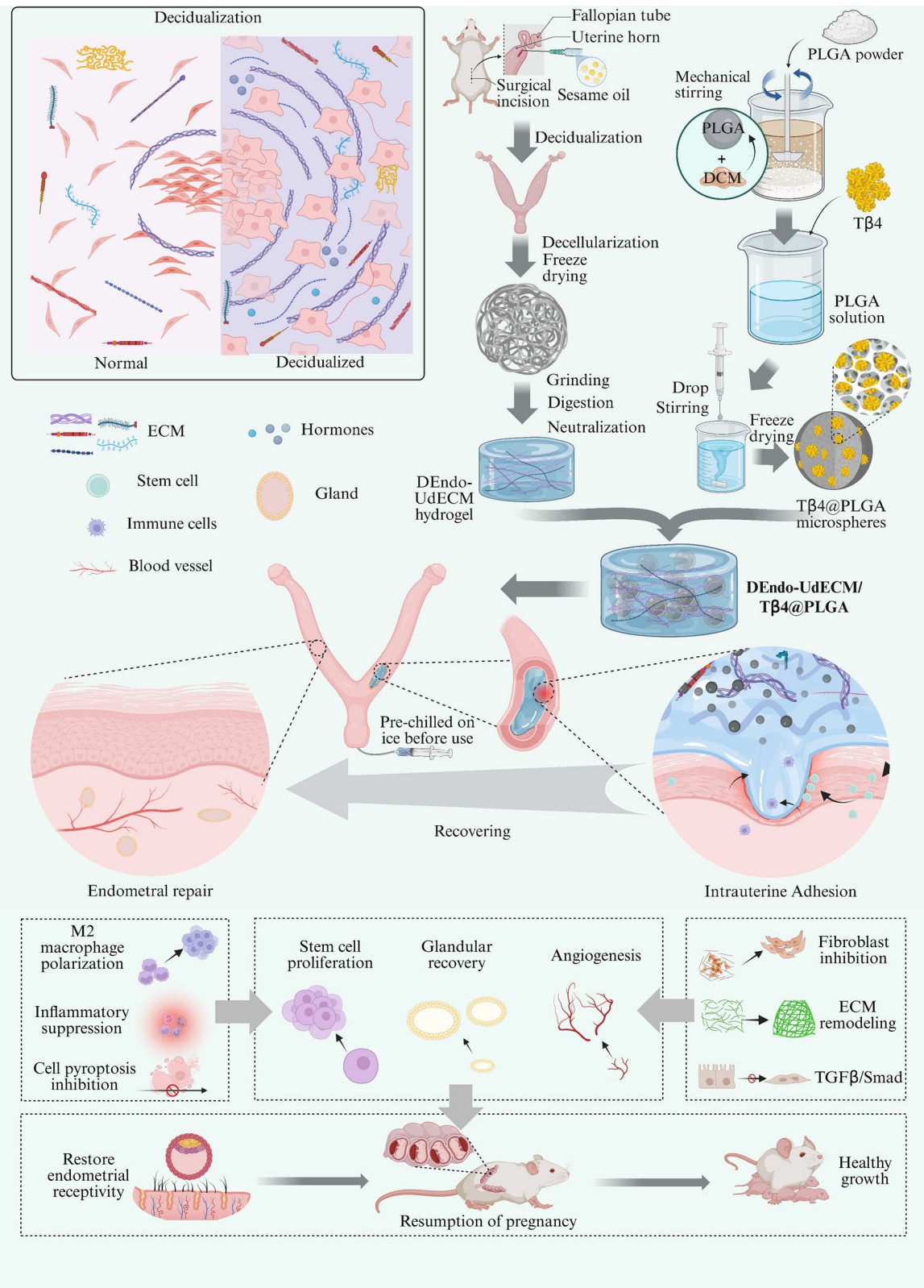

**Fig. 1 | Schematic diagram of material construction and therapeutic mechanism.** (Created in BioRender. Zhaowei, Y. (2025) https://BioRender.com/qx0t0ha).

signatures (Fig. 3e). While components like Collagen type I (Col1a1, Col1a2), Collagen type VI (Col6a1, Col6a2), Biglycan (Bgn), and Annexins (Anxa2, Anxa6) were prominent in both, CEndo-UdECM was distinguished by high levels of Tgm2, Col6a4, and Dcn. In contrast, DEndo-UdECM featured S100a10, Anxa1, and S100a6. Category-specific protein abundance analysis further detailed these

differences (Fig. 3f). DEndo-UdECM was enriched in Glycoproteins like Fibrinogen chains (Fga, Fgb), Fbn1, specific Laminin subunits (Lamb2, Lamc1, Lama5), and Postn, whereas Fn1 and TGFB1 characterized CEndo-UdECM. Collagen VI subunits predominated in CEndo-UdECM, while Collagen I levels were relatively higher in DEndo-UdECM. Among ECM regulators, while Serpinh1 and Tgm2 were shared, DEndo-UdECM

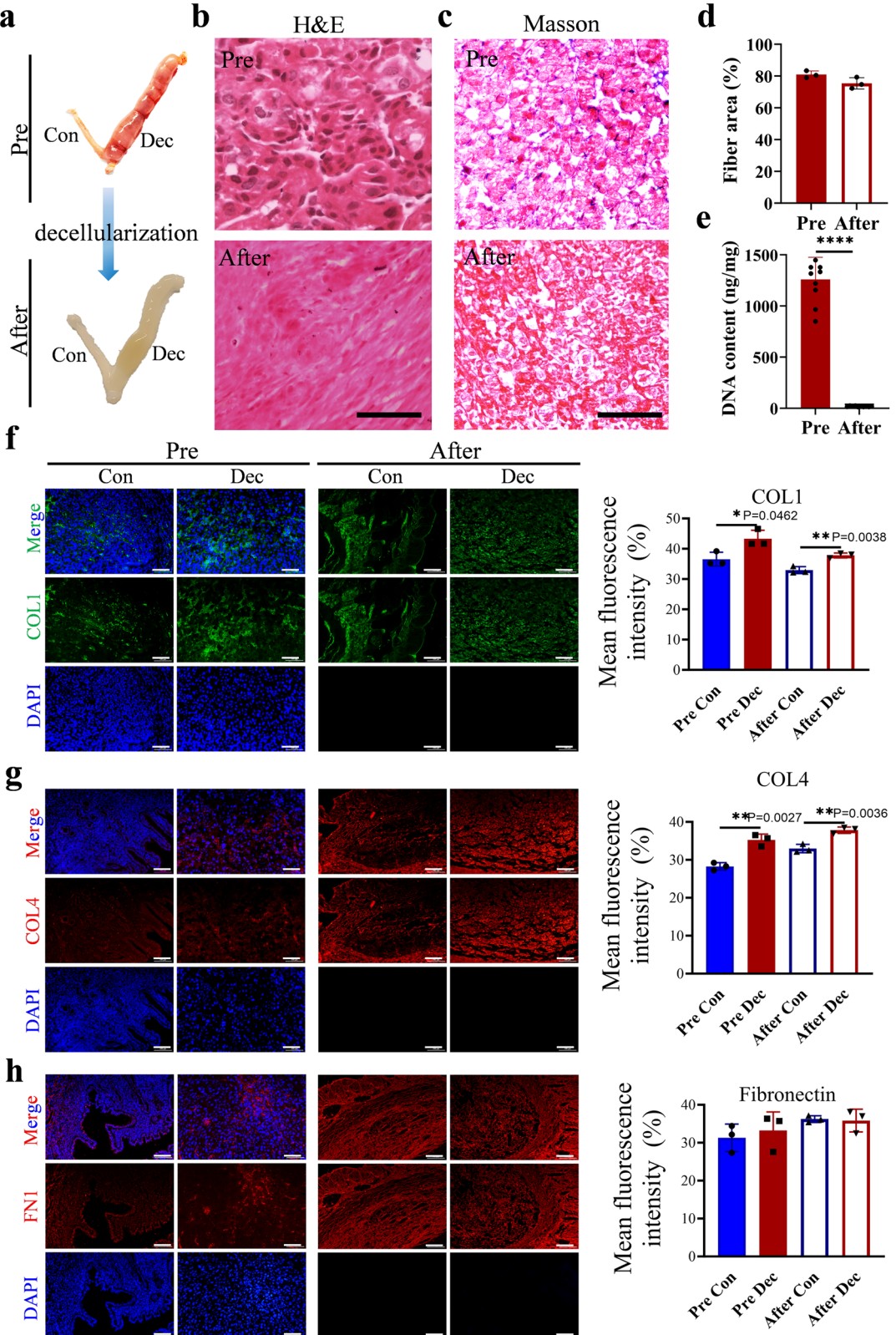

uniquely featured Pzp/A2M and Itih3. Shared proteoglycans included Bgn, Dcn, and Hspg2, but Prelp, Lum, and Ogn were enriched in CEndo-UdECM. Annexins (Anxa1, 2, 3, 5, 6) were abundant ECM-affiliated proteins in both. Notably, DEndo-UdECM showed specific enrichment in secreted S100 family members (S100a4, S100a6, S100a10, S100a11). Gene Ontology (GO) enrichment analysis of the 99 significantly upregulated matrisome proteins in DEndo-UdECM (Fig. 3g) revealed

significant associations (adjusted p-value < 0.05) with biological processes including "wound healing", "positive regulation of response to wounding", "collagen-containing extracellular matrix", and "cell-matrix adhesion". Pathways related to matrix dynamics ("regulation of peptidase activity", "regulation of collagen metabolic process", "negative regulation of TGFβ receptor signaling pathway"), hemostasis/angiogenesis ("coagulation", "fibrinolysis", "angiogenesis

**Fig. 2 | Validation of uterine decellularization and characterization of the resulting ECM scaffolds.** Uterine tissues from control (Con) and decidualized (Dec) horns were analysed before (Pre) and after the decellularization process. **a** Macroscopic images. **b** Haematoxylin and eosin (H&E) staining. **c** Masson's trichrome staining. **d** Quantification of collagen fibre area based on images in (**c**). **e** Quantification of residual DNA content. **f**–**h** Representative immunofluorescence images and quantification of mean fluorescence intensity (MFI) for Collagen I (COL1, green) (**f**), Collagen IV (COL4, red) (**g**), and Fibronectin (FN1, red) (**h**). Nuclei are counterstained with DAPI (blue). Scale bars, 100 μm (**b**, **c**); 50 μm (**f**–**h**). Images in (**a**–**c**) are representative of ($n = 6$) biologically independent samples with similar results. Data are presented as mean ± s.d. ($n = 6$ (**d**), $n = 10$ (**e**), $n = 3$ (**f**–**h**)) biologically independent samples. Statistical significance was assessed using an unpaired two-tailed Student's t-test. *$P < 0.05$, **$P < 0.01$, ***$P < 0.001$, ****$P < 0.0001$. Detailed statistical analyses, including exact $P$ values, test statistics ($F$ values and degrees of freedom), and 95% confidence intervals, are provided in the Source Data file.

involved in wound healing", "positive regulation of VEGF signaling pathway"), and cellular processes ("regulation of PI3K/PKB signal transduction", "regulation of epithelial cell proliferation", "negative regulation of apoptotic signaling pathway", "stem cell development") were also prominent. These distinct proteomic profiles indicate that the decellularization process preserved the overall ECM architecture while reflecting the unique compositional features induced by decidualization.

## Fabrication and characterization of DEndo-UdECM/Tβ4@PLGA composite hydrogels

The DEndo-UdECM/Tβ4@PLGA composite hydrogel system was fabricated by homogeneously suspending Tβ4@PLGA microspheres within the DEndo-UdECM precursor solution (Fig. 4a). Rheological measurements showed that the hydrogel precursor solution exhibited shear-thinning behavior, with viscosity decreasing as the shear rate increased (Fig. 4b), indicative of good injectability. This property was visually demonstrated by smoothly extruding the hydrogel precursor through a fine-gauge needle to form stable, high-resolution patterns (Supplementary Fig. 6a). To assess its gelation kinetics at physiological temperature, time-sweep rheology was performed. This revealed a rapid sol-gel transition, with the crossover of the storage modulus (G′) and loss modulus (G″) occurring at approximately 52 s (Fig. 4c), confirming its ability to form a stable scaffold quickly upon injection. The mechanical properties of the resulting gel were then characterized by frequency-sweep analysis, which showed that the storage modulus (G′, ≈ 1.0–1.3 kPa) was consistently higher than the loss modulus (G″, ≈ 0.6–0.7 kPa), indicative of a stable, elastic-dominant scaffold suitable for supporting soft tissue regeneration (Fig. 4d). SEM imaging of the pure DEndo-UdECM hydrogel revealed a porous, interconnected microstructure conducive to cell infiltration and nutrient exchange (Fig. 4i, left panel). Furthermore, its behavior in a physiological environment was assessed. The hydrogel demonstrated a controlled swelling profile, reaching a modest maximum swelling ratio of approximately 18.2% and indicating structural stability without the risk of uncontrolled expansion (Supplementary Fig. 6b). Its in vitro biodegradation profile showed near-complete degradation by day 14 (Supplementary Fig. 6c), a timeframe conducive to guiding endometrial repair.

To enable sustained therapeutic delivery, thymosin β4 (Tβ4) was encapsulated in PLGA microspheres. Particle size analysis revealed a highly consistent and narrow size distribution for both unloaded (mean diameter: 35.5 μm) and Tβ4-loaded (mean diameter: 35.6 μm) microspheres, confirming that drug loading did not significantly alter their physical dimensions (Fig. 4e, f). SEM imaging showed that both unloaded PLGA and Tβ4-loaded microspheres were spherical with smooth surfaces (Fig. 4g). To assess their degradation profile, the microspheres were incubated in vitro under physiological conditions and sampled at various time points. Bright-field microscopy showed a gradual loss of their sharp, spherical morphology over 10 days, while SEM imaging of parallel samples revealed the evolution from a smooth, intact surface to a more porous and eroded structure, indicating progressive degradation (Fig. 4h). Quantitative analysis determined the encapsulation efficiency (EE) to be 76.2 ± 1.6%, with the loading capacity (LC) reaching a plateau at initial Tβ4 concentrations above 300 ng/mL (Fig. 4i). When incorporated into the DEndo-UdECM hydrogel precursor, these microspheres were observed to be homogeneously dispersed throughout the resulting hydrogel matrix, as confirmed by cross-sectional SEM (Fig. 4j, right panel, Supplementary Fig. 7 and Supplementary Movie 1). Critically, the drug release profile was evaluated both for the microspheres alone and for the composite hydrogel system. While Tβ4-loaded microspheres alone exhibited a burst release with nearly 70% released within 3 days, their incorporation into the DEndo-UdECM hydrogel resulted in a significantly more sustained and controlled release profile. This dual-component system effectively mitigated the initial burst and prolonged delivery over the 10-day period, ensuring prolonged local availability of the therapeutic peptide (Fig. 4k). These experiments confirmed the successful construction and characterization of the composite therapeutic hydrogel system, providing the material basis for subsequent biological evaluation.

## DEndo-UdECM hydrogel promotes key cellular functions for endometrial regeneration in vitro

Following physicochemical characterization, we systematically investigated the in vitro biological effects of the hydrogel matrices to validate our design rationale, using gelatin as a baseline control. First, we confirmed the fundamental cytocompatibility of the materials. Live/Dead (AM/PI) staining revealed exceptionally high cell viability (> 98%) across all groups, with no significant difference observed between the Gelatin, CEndo-UdECM, and DEndo-UdECM hydrogels, establishing the biocompatibility of our ECM-derived materials (Fig. 5a, b; $P > 0.05$). Next, we assessed the core pro-regenerative functions. We began by evaluating cellular proliferation using two distinct but complementary assays. An EdU incorporation assay demonstrated that both CEndo-UdECM and DEndo-UdECM significantly promoted the proliferation of Ishikawa endometrial epithelial cells compared to the Gelatin control (Fig. 5c, d; $P < 0.05$). This finding was independently corroborated by Ki67 IF staining, which similarly showed a marked increase in the percentage of Ki67-positive cells in both ECM groups (Fig. 5e, f; $P < 0.01$). Critically, in both assays, the DEndo-UdECM group exhibited the most potent pro-proliferative effect, underscoring its superior regenerative potential. Cell migration, a critical process for wound closure, was evaluated using a scratch assay with endometrial stromal cells. Both CEndo-UdECM and DEndo-UdECM significantly accelerated wound closure over 12 h compared to the Gelatin control, with DEndo-UdECM again demonstrating the most robust effect (Fig. 5g, h; 12 h, $P < 0.0001$). Recognizing that robust angiogenesis is a cornerstone of endometrial regeneration, we then assessed the pro-angiogenic potential using a gold-standard tube formation assay with HUVECs. Qualitative observation over 12 h showed a clear progression towards more complex and stable capillary-like networks in the DEndo-UdECM group (Fig. 5i). To rigorously quantify these striking differences, we performed a comprehensive, multi-parameter analysis at multiple time points (3 h, 6 h, 12 h). This analysis provided unambiguous, time-dependent evidence of DEndo-UdECM's pro-angiogenic superiority. Compared to the Gelatin control, the DEndo-UdECM group induced a dramatic and statistically significant increase in all four key angiogenic metrics: Total tube length (Fig. 5j), Number of junctions (Fig. 5k), Number of meshes (Fig. 5l) Coverage area (Fig. 5m). Taken together, these robust, multi-faceted in vitro data provide

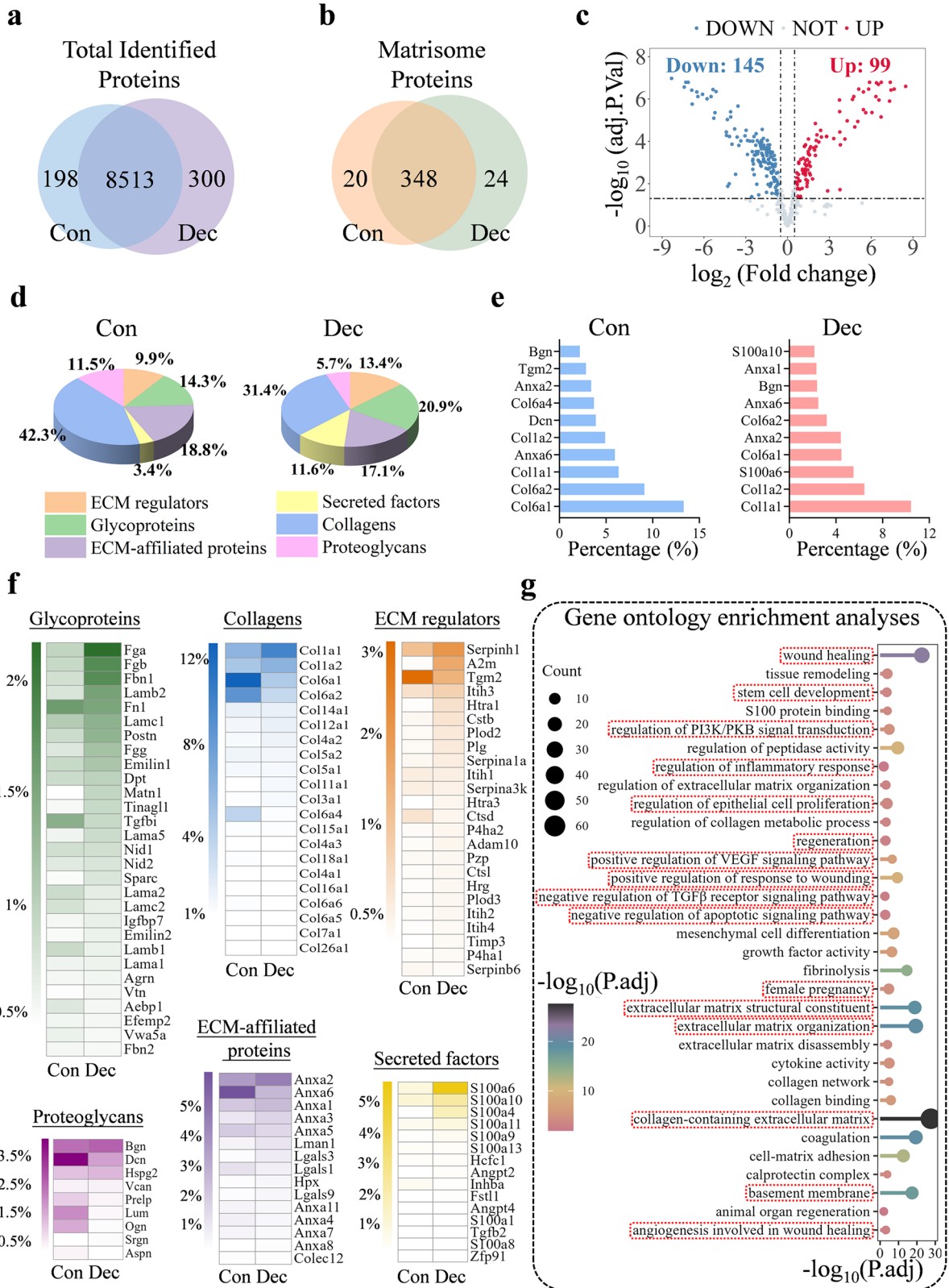

**a** Total Identified Proteins: Con 198, 8513, Dec 300

**b** Matrisome Proteins: Con 20, 348, Dec 24

**c** DOWN · NOT · UP; Down: 145, Up: 99; $-\log_{10}$ (adj.P.Val) vs $\log_2$ (Fold change)

**d** Con: 9.9%, 14.3%, 18.8%, 3.4%, 42.3%, 11.5%; Dec: 13.4%, 20.9%, 17.1%, 11.6%, 31.4%, 5.7%. ECM regulators, Glycoproteins, ECM-affiliated proteins, Secreted factors, Collagens, Proteoglycans

**e** Con and Dec bar charts, Percentage (%)

**f** Glycoproteins, Collagens, ECM regulators, ECM-affiliated proteins, Proteoglycans, Secreted factors heatmaps

**g** Gene ontology enrichment analyses; $-\log_{10}$(P.adj)

compelling evidence that the DEndo-UdECM matrix itself creates a potent pro-regenerative microenvironment by powerfully stimulating epithelial cell proliferation, stromal cell migration and proliferation, and, most critically, the formation of mature vascular networks. Separately, in an in vitro cellular fibrosis model induced by TGF-β1 in endometrial stromal cells, treatment with 50 nM Tβ4 significantly

decreased the expression levels of fibrosis markers α-SMA ($P < 0.05$), Col1a1 ($P < 0.0001$), Fibronectin ($P < 0.05$), COL4 ($P < 0.01$), and LAMININ ($P < 0.05$) (Supplementary Fig. 2a–i). To provide quantitative biochemical validation for these phenotypic observations, we performed Western Blot (WB) analysis. The results confirmed that Tβ4 treatment significantly suppressed the TGF-β1-induced

**Fig. 3 | Comparative proteomic analysis reveals a distinct, pro-regenerative matrisome in decidualized endometrium.** Proteomic data from decellularized uterine ECM derived from control (Con) and decidualized (Dec) uterine horns ($n = 3$ biologically independent samples per group). **a** Venn diagram of total identified proteins. **b** Venn diagram of matrisome-affiliated proteins. **c** Volcano plot of differentially abundant matrisome proteins in Dec versus Con UdECM. Differential abundance was assessed for each protein using a two-sided, moderated $t$-test. Significantly upregulated (red, $n = 99$) and downregulated (blue, $n = 145$) proteins are highlighted (adjusted $P < 0.05$ and $|\log_2 (\text{fold change})| > 1$). $P$ values were adjusted for multiple comparisons using the Benjamini–Hochberg procedure. **d** Relative abundance of major matrisome categories. **e** Top 10 most abundant matrisome proteins by relative abundance. **f** Heatmaps showing relative abundance of individual proteins within major matrisome subcategories. **g** Gene Ontology (GO) enrichment analysis for biological processes associated with proteins significantly upregulated in Dec UdECM. Enrichment was determined using a one-sided Fisher's Exact test, with $P$ values subsequently adjusted for multiple comparisons using the Benjamini–Hochberg procedure. Dot size corresponds to gene count; the colour scale indicates the adjusted $P$ value.

upregulation of both α-SMA and COL1A1 protein levels (Supplementary Fig. 2j–l; $P < 0.0001$), offering direct molecular evidence of its anti-fibrotic activity.

## DEndo-UdECM/Tβ4@PLGA hydrogel promotes endometrial regeneration and reverses fibrosis in a murine IUA model

To assess the in vivo therapeutic efficacy of our composite hydrogel, we established a severe murine model of IUA via 95% ethanol-induced chemical injury[9], followed by a single administration of the respective treatments (Fig. 6a). Fourteen days post-treatment, we evaluated endometrial regeneration and the resolution of fibrosis. Gross morphological assessment revealed that uteri from the Model group exhibited severe luminal stenosis and cavity dilation, pathologies that were only minimally improved by Tβ4@PLGA monotherapy. In stark contrast, the DEndo-UdECM/Tβ4@PLGA treatment group displayed a uterine morphology most comparable to the healthy Sham group, with no apparent stenosis or fluid retention (Fig. 6b). Histological analysis via H&E staining corroborated these macroscopic findings. The Model group displayed a severely thinned, damaged endometrium with extensive luminal adhesion and a notable absence of endometrial glands. While hydrogels or Tβ4@PLGA alone offered partial recovery, the DEndo-UdECM/Tβ4@PLGA composite hydrogel induced the most profound structural restoration, characterized by a well-organized, thick endometrial layer and numerous regenerated glands (Fig. 6c). Quantification confirmed these observations: the DEndo-UdECM/Tβ4@PLGA treatment led to a significant increase in both endometrial thickness (Fig. 6d) and the number of glands (Fig. 6e) compared to all other groups ($P < 0.0001$ vs. Model; $P < 0.0001$ vs. CEndo-UdECM/Tβ4@PLGA). To further validate glandular regeneration, we stained for FOXA2, a key marker for glandular epithelium. IF analysis showed a near-complete loss of FOXA2 signal in the Model group, which was robustly restored by the DEndo-UdECM/Tβ4@PLGA treatment to levels approaching the Sham group and significantly higher than all other treatments (Fig. 6f, g). Alongside structural regeneration, a critical component of endometrial repair is the resolution of pathological fibrosis[3]. Masson's trichrome staining revealed extensive collagen deposition (indicated by dense blue staining) in the Model group, signifying severe fibrosis. The DEndo-UdECM/Tβ4@PLGA hydrogel treatment resulted in the most substantial reduction in the fibrotic area, significantly outperforming all other groups (Fig. 6h, i; $P < 0.0001$ vs. Model; $P < 0.01$ vs. CEndo-UdECM/Tβ4@PLGA). To dissect the molecular underpinnings of this anti-fibrotic effect, we assessed the expression of key pro-fibrotic markers, alpha-smooth muscle actin (α-SMA) and Collagen I (COL1). IF staining showed a dramatic upregulation of both α-SMA and COL1 in the endometrial stroma of the Model group. Conversely, the DEndo-UdECM/Tβ4@PLGA treatment most effectively suppressed the expression of both markers (Fig. 6j), leading to a significant reduction in the mean fluorescence intensity of α-SMA (Fig. 6k) and COL1 (Fig. 6l) compared to the Model and other treatment groups.

Collectively, these results demonstrate that the DEndo-UdECM/Tβ4@PLGA hydrogel orchestrates a dual therapeutic action, concurrently promoting robust structural and glandular regeneration while potently suppressing the pathological fibrotic response that characterizes severe IUA.

## DEndo-UdECM/Tβ4@PLGA treatment effects on fertility in IUA mice

To assess functional recovery, the "gold standard" of fertility was evaluated by quantifying embryo implantation sites and live birth numbers after mating (Fig. 7a). The Sham group exhibited $8 \pm 1.0$ implantation sites per mouse (sum of both uterine horns), whereas the Model group showed only $2 \pm 0.2$ site. The Tβ4@PLGA group had $2 \pm 1.0$ sites ($P > 0.05$ vs. Model). CEndo-UdECM and DEndo-UdECM treatments increased the number of sites to $3 \pm 0.5$ and $5 \pm 0.5$ ($P < 0.0001$ vs. Model), respectively. The CEndo-UdECM/Tβ4@PLGA group had $5 \pm 1.0$ sites ($P < 0.01$ vs. Model). Notably, the DEndo-UdECM/Tβ4@PLGA group reached $7 \pm 1.0$ sites, significantly higher than the Model group ($P < 0.0001$) and approaching the levels of the Sham group (Fig. 7b, d). Consistent with implantation data, live birth rates were significantly decreased in the Model group compared to the Sham group (which had $12 \pm 2.0$ pups per litter) ($P < 0.0001$). The DEndo-UdECM/Tβ4@PLGA group achieved a live birth rate of $12 \pm 0.8$ pups per litter, comparable to the Sham group ($P > 0.05$ vs. Sham) and significantly higher than the Model group ($P < 0.0001$) and other treatment groups (Fig. 7c, e). To assess the long-term viability and health of the offspring, we monitored their postnatal development. Crucially, all live-born offspring, regardless of the treatment group, exhibited a 100% survival rate to weaning with no observable physical abnormalities (Table S2). This uniform health was further corroborated by comparative imaging of weaned pups, which revealed no discernible differences in body size or physical condition among the groups (Fig. 7f). Collectively, these fertility experiments confirm that the DEndo-UdECM/Tβ4@PLGA hydrogel not only restores implantation rates but, crucially, achieves a comprehensive functional recovery enabling the birth and healthy development of a normal-sized litter.

## The composite hydrogels exhibit systemic safety and local biocompatibility in vivo

To evaluate the translational potential of our hydrogel systems, we conducted a comprehensive safety assessment focusing on both systemic toxicity and local tissue compatibility. Systemic safety was first confirmed through histological analysis of major organs (heart, liver, spleen, lungs, and kidneys) harvested from the IUA model animals 14 days post-treatment. H&E staining revealed no evidence of inflammation, necrosis, or other pathological abnormalities in any hydrogel-treated group, with tissue architecture indistinguishable from controls (Supplementary Fig. 3a). For local biocompatibility, we subcutaneously implanted the hydrogels and monitored the tissue response over 21 days. H&E staining provided compelling evidence of the material's gentle integration; even at day 7, the fundamental skin structure remained intact and well-organized, showing no significant signs of cellular infiltration or structural disruption (Supplementary Fig. 3b). This seamless integration was followed by a complete, scar-free resolution by day 21. Furthermore, Masson's trichrome staining critically showed no evidence of excessive collagen deposition or the formation of a dense fibrous capsule, confirming the absence of a significant fibrotic foreign body response (Supplementary Fig. 3c). To dissect the molecular underpinnings, time-course immunohisto-chemical analysis for the pro-inflammatory cytokines IL-6 and IL-1β

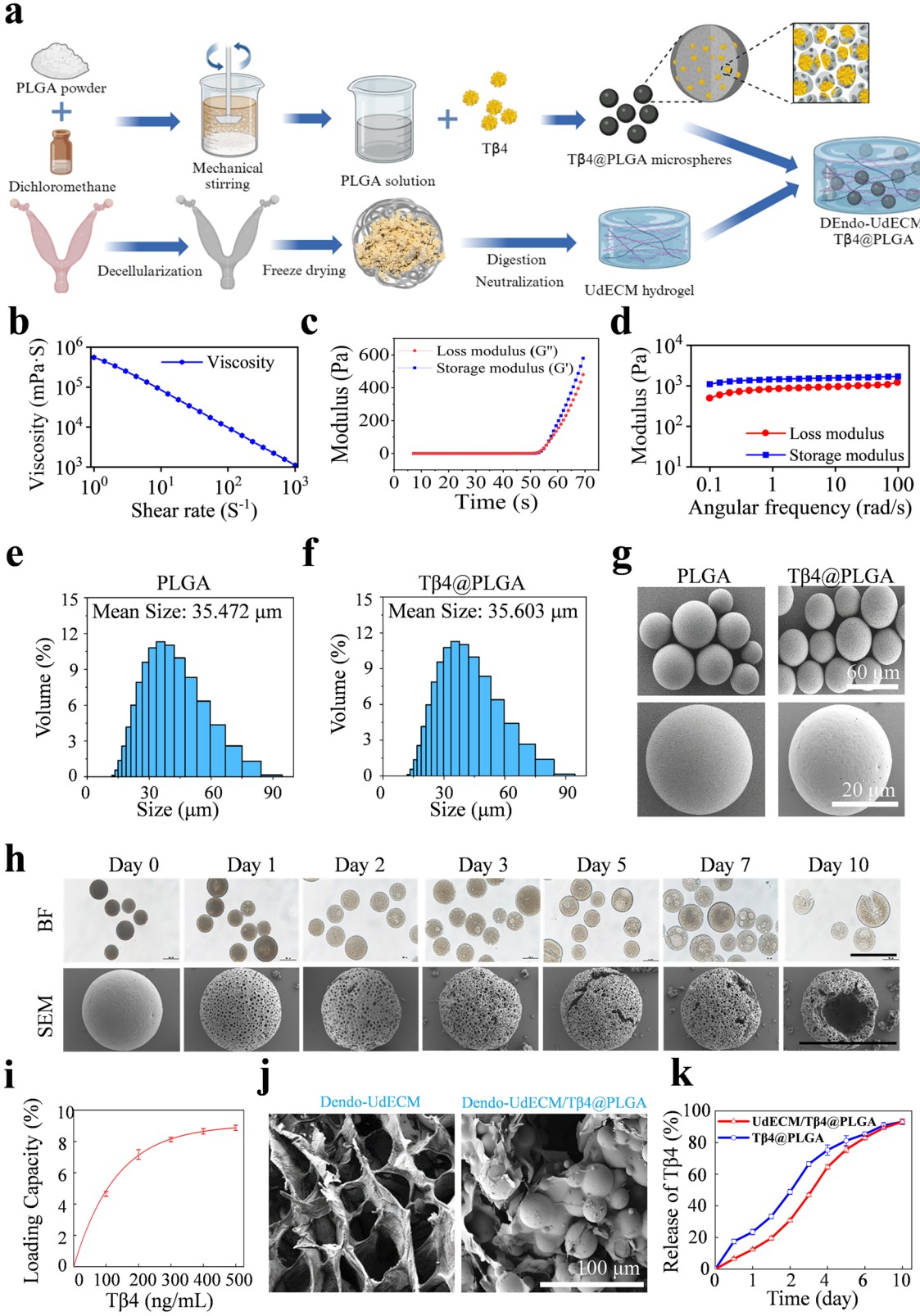

revealed only a transient, low-level expression at day 7 that was comparable to the PBS control and resolved to baseline by day 21, providing direct molecular evidence that our hydrogels do not elicit a chronic inflammatory state (Supplementary Fig. 3d, e). Taken together, these systemic and multi-level local safety assessments—validated at the histological, architectural, and molecular levels—robustly demonstrate in vivo biocompatibility of the Tβ4@PLGA@CEndo-UdECM and

Tβ4@PLGA@DEndo-UdECM hydrogels, supporting their strong potential for future clinical applications.

## DEndo-UdECM/Tβ4@PLGA treatment rebalances the immune microenvironment towards a pro-regenerative M2 phenotype

Given that macrophage polarization is a critical determinant of the inflammatory-reparative balance during tissue repair[3], we performed a

**Fig. 4 | Fabrication and characterization of the DEndo-UdECM/Tβ4@PLGA composite hydrogel. a** Schematic of the fabrication process for the composite hydrogel system (Created in BioRender. Zhaowei, Y. (2025) https://BioRender.com/ul4iqzn.). **b–d** Rheological characterization of the DEndo-UdECM hydrogel. **b** Viscosity of the precursor solution as a function of shear rate, demonstrating shear-thinning properties. **c** Time sweep rheology tracking the sol-gel transition at 37 °C. The crossover points of the storage (G′) and loss (G″) modulus indicates the gelation time. **d** Frequency sweep analysis showing the storage (G′) and loss (G″) moduli of the crosslinked hydrogel. **e**, **f** Size distribution histograms for blank PLGA (**e**) and Tβ4@PLGA (**f**) microspheres. **g** Scanning electron microscopy (SEM) images of blank PLGA and Tβ4@PLGA microspheres. Scale bars, 60 μm (top), 20 μm (bottom). **h** Imaging showing the morphological evolution of Tβ4@PLGA microspheres during in vitro degradation over 10 days. Top row: bright-field (BF) microscopy; bottom row: scanning electron microscopy (SEM). Scale bars, 100 μm. **i** Tβ4 loading capacity of PLGA microspheres as a function of initial Tβ4 concentration. **j** Cross-sectional SEM images of the acellular DEndo-UdECM hydrogel (left) and the composite hydrogel embedded with Tβ4@PLGA microspheres (right). Scale bar, 100 μm. **k** Cumulative in vitro Tβ4 release profiles from Tβ4@PLGA microspheres alone (blue) and from the composite hydrogel (red). Images in (**g**, **h**, and **j**) are representative of three independent experiments with similar results. In (**i** and **k**), data are presented as mean ± SD from $n = 3$ independent experiments. Source data are provided as a Source Data file.

comprehensive four-marker IF analysis to rigorously investigate the influence of our composite hydrogel on macrophage phenotype at the injury site. We assessed two canonical M1 markers (CD86 and iNOS) and two canonical M2 markers (Arginase-1 and CD206).

First, we examined the pro-inflammatory M1 phenotype. Expression of CD86 was significantly upregulated in the Model group compared to the Sham group ($P < 0.001$) (Fig. 8a, b). Tβ4@PLGA treatment significantly reduced CD86+ cell density ($P < 0.001$ vs. Model). Both CEndo-UdECM and DEndo-UdECM treatments further decreased CD86+ cell density ($P < 0.0001$ vs. Model). The DEndo-UdECM/Tβ4@PLGA group exhibited the lowest CD86+ cell density, comparable to the Sham group ($P > 0.05$ vs. Sham). Consistent with these findings, the expression of the M1 marker iNOS followed a nearly identical pattern, with the DEndo-UdECM/Tβ4@PLGA treatment most effectively suppressing its expression to levels approaching the Sham group (Fig. 8c, d).

Conversely, we assessed the pro-reparative M2 phenotype. Expression of Arginase-1 (Arg-1) was significantly diminished in the Model group compared to Sham (Fig. 8e, f). Tβ4@PLGA treatment alone did not significantly alter Arg-1+ cell density compared to the Model group ($P > 0.05$). Both CEndo-UdECM and DEndo-UdECM treatments significantly increased Arg-1+ cell density compared to the Model group. The DEndo-UdECM/Tβ4@PLGA group displayed the highest Arg-1+ cell density, similar to the Sham group. This pro-regenerative shift was powerfully corroborated by the analysis of CD206, another key M2 marker. The DEndo-UdECM/Tβ4@PLGA group demonstrated the most profound upregulation of CD206 expression, significantly higher than all other treatment and control groups (Fig. 8g, h).

To further corroborate these IF findings with a robust, single-cell quantitative method, we performed flow cytometric analysis on single-cell suspensions from the regenerated endometrial tissues. After gating on the total macrophage population (F4/80⁺), we quantified the relative proportions of M1-like (CD86⁺) and M2-like (CD206⁺) cells (Supplementary Fig. 8a). This analysis revealed a starkly elevated M1/M2 ratio in the Model group, confirming a dominant pro-inflammatory state. In stark contrast, treatment with the DEndo-UdECM/Tβ4@PLGA hydrogel induced a profound reduction in the M1/M2 ratio to a level even lower than that of the Sham group, significantly outperforming all other control treatments (Supplementary Fig. 8b). This flow cytometry data provides unequivocal quantitative validation of our immunohistochemical observations.

Taken together, this comprehensive four-marker analysis, powerfully corroborated by quantitative flow cytometry, unequivocally demonstrates that the DEndo-UdECM/Tβ4@PLGA treatment orchestrates a profound shift in the local immune microenvironment. It does so by concurrently suppressing the pro-inflammatory M1 phenotype (evidenced by reduced CD86 and iNOS) and promoting the pro-regenerative M2 phenotype (evidenced by elevated Arg-1 and CD206), thereby creating optimal conditions for endometrial repair.

## DEndo-UdECM/Tβ4@PLGA treatment effects on markers of regeneration

Effective tissue repair requires active promotion of regeneration alongside inhibition of detrimental processes. Therefore, the pro-regenerative capacity of the hydrogel was assessed by examining markers for epithelialization, stem/progenitor cells, angiogenesis, and proliferation. Co-staining of Pan-Cytokeratin 7 (Pan-CK, epithelial marker) and Aldehyde Dehydrogenase 1 Family Member A1 (ALDH1A1, stem/progenitor cell-associated marker) was performed (Fig. 9a). In the Model group, both Pan-CK⁺ epithelial structures and ALDH1A1⁺ cells were largely absent ($P < 0.0001$ vs. Sham for both) (Fig. 9–c). CEndo-UdECM and Dendo-UdECM single treatments resulted in scattered Pan-CK⁺ signals ($P < 0.001$ vs. Model) and a small number of ALDH1A1⁺ cells ($P < 0.001$ vs. Model). Tβ4@PLGA treatment showed no significant improvement in Pan-CK ⁺ and ALDH1A1⁺ signal ($P > 0.05$ vs. Model). The Dendo-UdECM/Tβ4@PLGA group showed the highest levels of both ALDH1A1⁺ cells ($P < 0.0001$ vs. Model, Cendo-UdECM/Tβ4@PLGA, and Dendo-UdECM) and Pan-CK⁺ epithelial coverage ($P < 0.0001$ vs. Model, Cendo-UdECM/Tβ4@PLGA, and Dendo-UdECM). Angiogenesis was assessed by CD34 (vascular endothelial marker) IF (Fig. 9d, e). CD34⁺ cells/vessels were largely absent in the Model group endometrium ($P < 0.0001$ vs. Sham). No significant improvement was observed in the Tβ4@PLGA ($P > 0.05$ vs. Model) or Cendo-UdECM ($P > 0.05$ vs Model) groups. Dendo-UdECM and Cendo-UdECM/Tβ4@PLGA treatments resulted in a small number of scattered CD34⁺ signals ($P < 0.0001$ vs. Model). Dendo-UdECM/Tβ4@PLGA treatment resulted in a significant increase in CD34⁺ cell density compared to the Model group ($P < 0.0001$), restoring levels comparable to the Sham group and significantly higher than other treatment strategies ($P < 0.0001$ vs. Cendo-UdECM/Tβ4@PLGA; $P < 0.0001$ vs. Dendo-UdECM). Cell proliferation was evaluated by Ki67 staining (Fig. 9f, g). Ki67⁺ proliferating cells were almost undetectable in the Model group ($P < 0.0001$ vs. Sham). No significant change was observed in the Tβ4@PLGA group ($P > 0.05$ vs. Model). Cendo-UdECM and Dendo-UdECM treatments showed a marked increase in Ki67⁺ cells ($P < 0.0001$ vs. Model). Cendo-UdECM/Tβ4@PLGA also increased Ki67⁺ cells ($P < 0.0001$ vs. Model). The Dendo-UdECM/Tβ4@PLGA treatment resulted in the highest Ki67+ cell density, comparable to the Sham group and significantly higher than other treatment strategies ($P < 0.0001$ vs. Cendo-UdECM/Tβ4@PLGA; $P < 0.0001$ vs. Dendo-UdECM). Assessment of markers related to epithelialization, stem/progenitor cells, angiogenesis, and proliferation indicated enhanced regenerative processes following Dendo-UdECM/Tβ4@PLGA treatment.

## Dendo-UdECM/Tβ4@PLGA treatment effects on pyroptosis and TGF-β/Smad3 signaling

To elucidate the molecular mechanisms underlying the therapeutic efficacy of our composite hydrogel, we investigated its impact on two critical pathological pathways implicated in IUA: pyroptosis-driven inflammation and TGF-β-mediated fibrosis[10] (Fig. 10a). WB analysis of

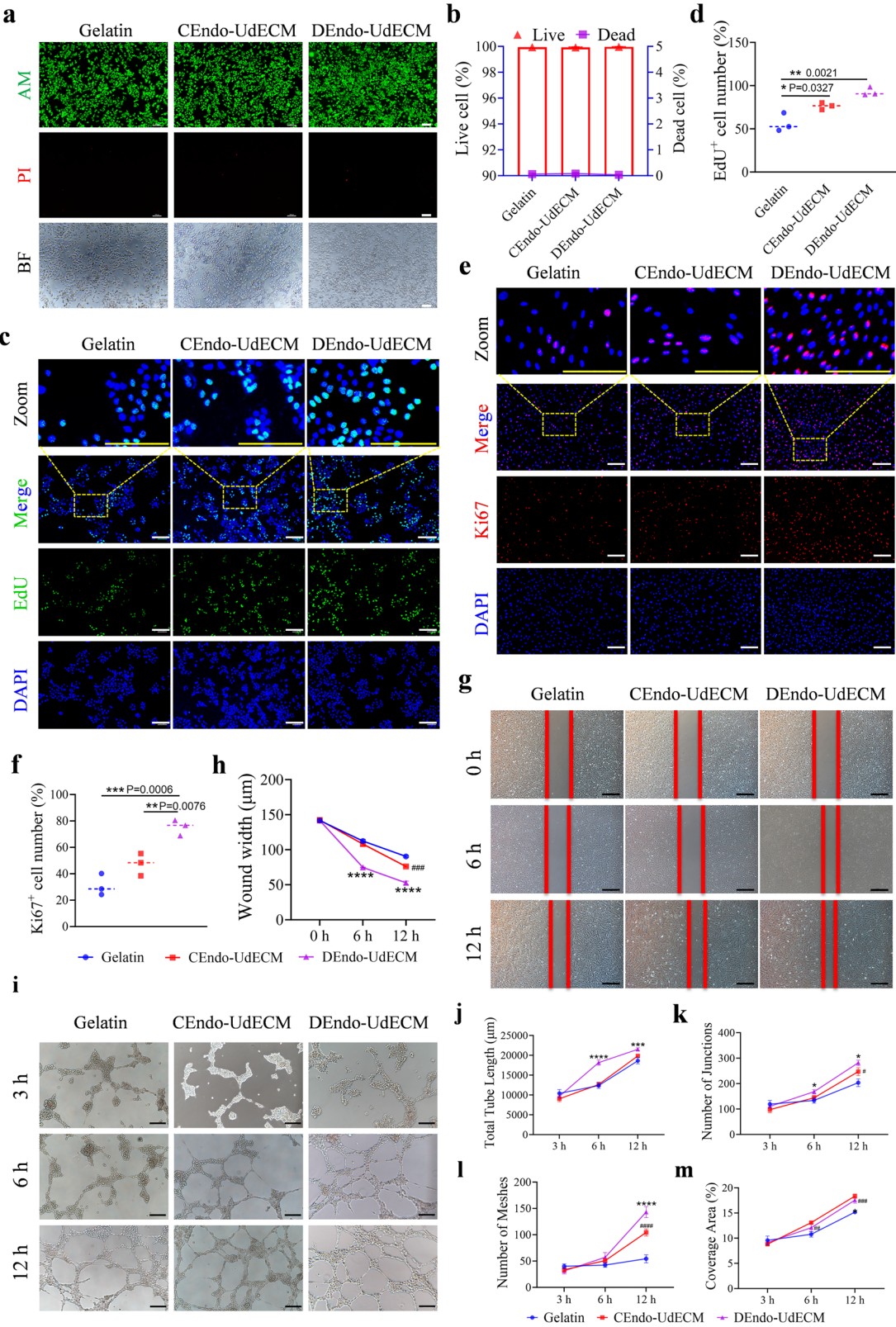

uterine tissues revealed a dramatic activation of the pyroptosis cascade in the Model group. This was evidenced by a significant upregulation of Pro-caspase-1 and, critically, a marked increase in the cleaved, active N-terminal fragment of Gasdermin D (GSDMD-NT), the ultimate executor of pyroptotic cell death, while full-length GSDMD levels remained relatively stable (Fig. 10b–e). Consequently, the downstream pro-inflammatory cytokines IL-1β and IL-18 were substantially elevated in the Model group (Fig. 10f, g). While individual components of the treatment showed modest effects, the DEndo-UdECM/Tβ4@PLGA hydrogel potently suppressed this entire inflammatory axis, significantly reducing the levels of Pro-caspase-1, GSDMD-NT, IL-1β, and IL-18 to levels approaching those of the healthy Sham group.

Concurrently, we assessed the canonical pro-fibrotic TGF-β/Smad signaling pathway. IF staining and WB analysis both confirmed a

**Fig. 5 | DEndo-UdECM hydrogel promotes pro-regenerative cellular responses in vitro. a**, **b** Cytocompatibility assessment of Ishikawa cells using a Live/Dead staining assay. **a** Representative fluorescence images showing live cells (Calcein-AM, green) and dead cells (Propidium Iodide/PI, red), with corresponding bright-field (BF) images. **b** Quantification of cell viability percentage. **c**, **d** Proliferation assessment of Ishikawa cells via EdU incorporation assay. **c** Representative fluorescence images showing EdU-positive proliferating cells (green) and all cell nuclei (DAPI, blue), with magnified zoom panels. **d** Quantification of the percentage of EdU-positive cells. **e**, **f** independent validation of cell proliferation using Ki67 immunofluorescence staining in human endometrial stromal cells. **e** Representative images showing Ki67-positive cells (red) and nuclei (DAPI, blue). **f** Quantification of the percentage of Ki67-positive cells. **g**, **h** Migration of mouse endometrial stromal cells (mESCs) evaluated by a scratch wound healing assay.

**g** Representative bright-field images at 0, 6, and 12 h. **h** Quantification of wound width over time. **i**–**m** Pro-angiogenic potential assessed by a gold-standard tube formation assay using HUVECs. **i** Representative bright-field images at 3, 6, and 12 h demonstrate the progressive formation of complex capillary-like networks. **j**–**m** Comprehensive quantitative analysis confirmed a significant increase in (**j**) total tube length, (**k**) number of junctions, (**l**) number of meshes, and (**m**) coverage area in the DEndo-UdECM group compared to controls. Scale bars, 100 μm. In (**b**, **d**, **f**, **h**, **j**–**m**), data are presented as mean ± s.d. from $n = 3$ biologically independent samples. Statistical significance was determined by one-way ANOVA with Tukey's post hoc test; *$P < 0.05$, **$P < 0.01$, ***$P < 0.001$, ****$P < 0.0001$. Gelatin was used as the control group. Detailed statistical analyses, including exact $P$ values, test statistics (F values and degrees of freedom), and 95% confidence intervals, are provided in the Source Data file.

significant upregulation of the master fibrogenic cytokine TGF-β1 in the injured endometrium of the Model group (Fig. 10k–m). This led to a corresponding increase in the phosphorylation of the downstream effector Smad3, as indicated by an elevated p-Smad3/Smad3 ratio (Fig. 10i). The DEndo-UdECM/Tβ4@PLGA treatment most effectively blocked this pathway, resulting in the strongest reduction of both TGF-β1 expression and Smad3 phosphorylation. This upstream inhibition translated directly to a significant downstream reduction in the key fibrotic proteins α-SMA and COL1A1 (Fig. 10h, j), quantitatively corroborating our histological anti-fibrosis findings.

In summary, these molecular analyses demonstrate that the DEndo-UdECM/Tβ4@PLGA hydrogel orchestrates endometrial repair through a powerful dual mechanism: it simultaneously quenches the pro-inflammatory fire of the pyroptosis pathway and dismantles the pro-fibrotic machinery of the TGF-β/Smad axis, thereby creating a microenvironment conducive to regeneration rather than scarring.

## DEndo-UdECM/Tβ4@PLGA hydrogel restores the molecular signature of endometrial receptivity

To provide a molecular basis for the observed restoration of fertility, we performed a comprehensive assessment of key markers that define the endometrial receptive state[41]. The IUA Model group exhibited a molecular profile characteristic of a non-receptive endometrium. Critical pro-receptivity factors were significantly downregulated, including Progesterone Receptor (PR) in both epithelial and stromal compartments, Estrogen Receptor α (ERα), and the essential embryo attachment molecules Osteopontin (OPN) and Integrin β3 (Supplementary Fig. 9a–i). In striking contrast, treatment with the DEndo-UdECM/Tβ4@PLGA hydrogel robustly restored the expression of all these positive markers, elevating their levels to closely approximate those seen in the healthy Sham group.

Conversely, we examined markers associated with a non-receptive or hostile endometrial state. In the Model group, the luminal epithelium displayed aberrantly high levels of the proliferation marker PCNA, indicating a failure to enter the necessary quiescent state required for the implantation window (Supplementary Fig. 9j–l). Furthermore, the anti-adhesion mucin MUC1, which can physically block embryo attachment, was significantly upregulated (Supplementary Fig. 9m, n). The DEndo-UdECM/Tβ4@PLGA treatment effectively reversed these pathological changes, normalizing the expression of PCNA and significantly downregulating MUC1 to levels comparable with the Sham group.

Collectively, these data demonstrate that our composite hydrogel not only physically repairs the endometrium but also fundamentally re-establishes the complex molecular milieu required for successful embryo implantation. This provides a clear mechanistic underpinning for the functional recovery of fertility observed in our study.

## Discussion

IUA represent a major clinical challenge, fundamentally stemming from a pathological shift where endometrial injury leads to fibrotic

scarring rather than functional regeneration[2]. Our findings directly address this core pathology by demonstrating that a biomaterial strategy empowered by the principles of decidualization can effectively reverse this process. We show that a composite hydrogel, integrating a bioactive scaffold from decidualized endometrium (DEndo-UdECM) with the sustained release of an anti-fibrotic peptide (Tβ4), orchestrates a profound regenerative response. In a stringent preclinical IUA model, this approach not only restored complex endometrial architecture but, most critically, achieved a near-complete recovery of fertility. This work provides compelling evidence that harnessing the intrinsic biological intelligence encoded within the decidual niche offers a powerful paradigm for functional uterine repair.

The cornerstone of our therapeutic strategy is the hypothesis that an ECM scaffold derived from a decidualized endometrium possesses a unique, intrinsically pro-regenerative molecular signature. To harness this, we first established and validated a robust artificial decidualization model in mice, which faithfully recapitulated the key morphological and molecular hallmarks of the physiological process, including increased vascularization (CD31) and upregulation of essential decidual markers like HAND2 and PR (Supplementary Fig. 1). This model was strategically chosen to isolate the intrinsic maternal uterine response, free from embryonic signals, providing a consistent and high-fidelity source for our biomaterial[42,43]. Following an optimized decellularization protocol, we successfully removed over 98% of the cellular DNA content while critically preserving the native ECM architecture and its core protein constituents, such as collagens and fibronectin (Fig. 2), ensuring the scaffold retained its structural and biological integrity[44,45].

Unbiased proteomic analysis provided a deep dive into the molecular basis for DEndo-UdECM's superiority. The analysis revealed a profound remodeling of the matrisome during decidualization, resulting in a distinct protein landscape in DEndo-UdECM compared to its non-decidualized counterpart (CEndo-UdECM) (Fig. 3a–c). Specifically, DEndo-UdECM was significantly enriched in crucial glycoproteins, secreted factors, and ECM regulators known to be involved in morphogenesis, immunomodulation, and cell-matrix interactions (Fig. 3d–f)[27,46,47]. G Critically, GO analysis of the upregulated proteins pinpointed an enrichment of pathways governing "wound healing", "angiogenesis", and the "negative regulation of TGF-β signaling" (Fig. 3g). Our proteomic findings are strongly supported by the established biology of the decidual niche. Decidualization involves profound remodeling of the ECM, creating a microenvironment distinct from both quiescent and fibrotic tissue. Foundational reviews highlight that decidual stromal cells become surrounded by a rich pericellular ECM, with significant increases in components like collagen IV and laminin, which are integral to establishing this functional niche[48]. The unique repertoire of collagens present at the maternal-fetal interface is not merely structural but plays active roles in immune modulation and angiogenesis[49]. This aligns with our observation of enriched pathways governing these processes. Compellingly, specific

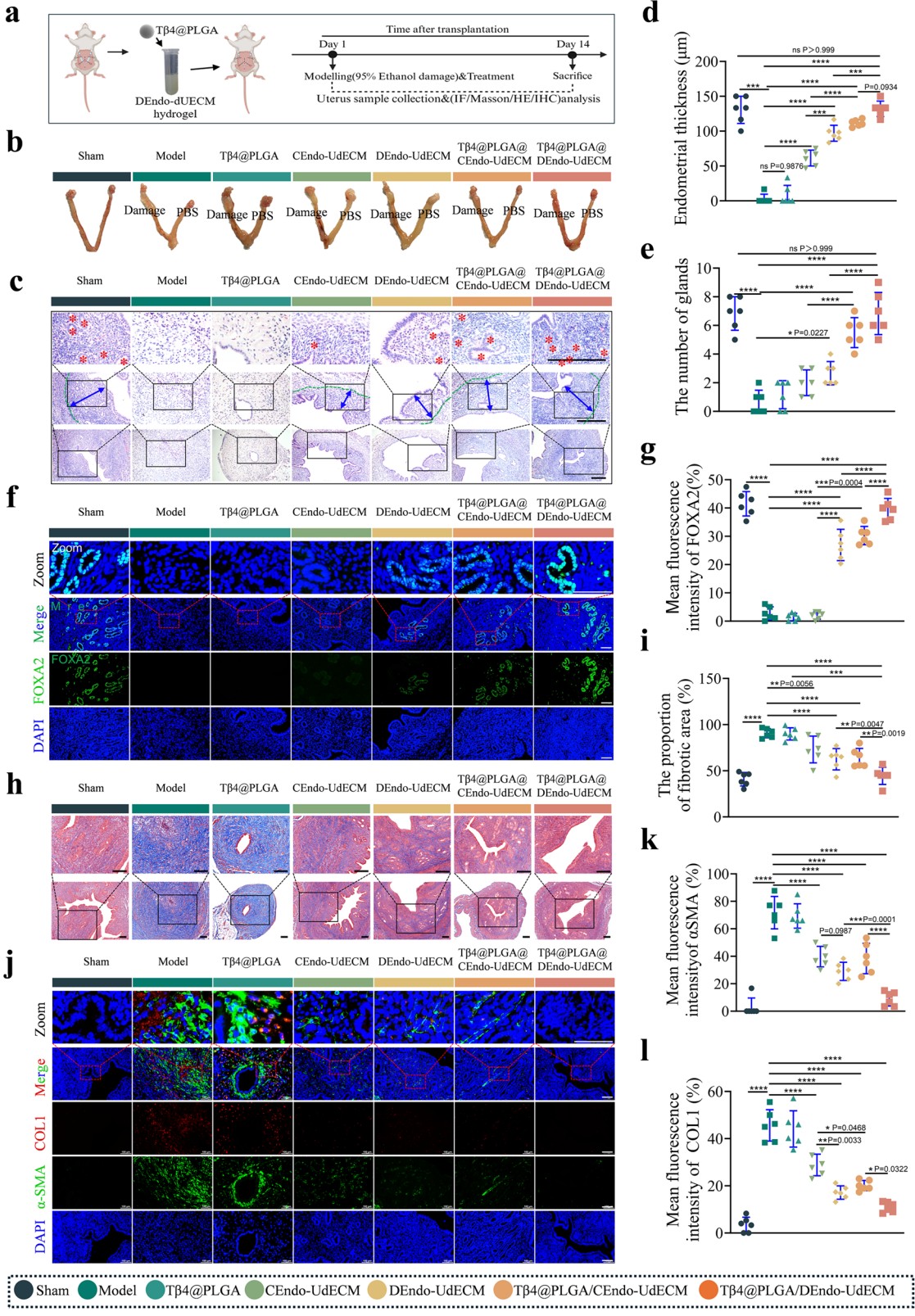

ECM components are not just present but are functionally indispensable; for instance, Laminin A5 is strongly upregulated under progesterone control, and its knockdown is known to impair the decidualization process itself, underscoring the inherent bioinstructive nature of these proteins[27]. By capturing this complex, insoluble matrix, our approach harnesses a pre-programmed, bioactive landscape poised for regeneration, demonstrating that DEndo-

UdECM is not merely a passive scaffold but a bioactive repository of instructive cues.

To augment these intrinsic properties and provide sustained antifibrotic pressure, we engineered a composite system by incorporating Tβ4-loaded PLGA microspheres into the DEndo-UdECM hydrogel (Fig. 4a). This design provided a controlled release of Tβ4 over approximately 10 days, a profile deliberately synchronized with the

**Fig. 6 | The DEndo-UdECM/Tβ4@PLGA hydrogel promotes functional endometrial regeneration and suppresses fibrosis in a murine injury model.**
**a** Schematic of the in vivo study design (Created in BioRender. Zhaowei, Y. (2025) https://BioRender.com/ul4iqzn). **b** Gross morphology of uteri 14 days post-treatment. **c** Representative H&E-stained uterine cross-sections. Red asterisks indicate endometrial glands. Blue double-headed arrows illustrate the measurement of endometrial thickness. **d, e** Quantification of endometrial thickness (**d**) and the number of endometrial glands per section (**e**). **f, g** Assessment of glandular function. **f** Immunofluorescence staining for the glandular marker FOXA2 (green). **g** Quantification of FOXA2 mean fluorescence intensity. **h, i** Evaluation of overall fibrosis. **h** Masson's trichrome staining, where collagen is stained blue and muscle/cytoplasm red. **i** Quantification of the fibrotic area as a percentage of the total tissue area. **j–l** Analysis of key fibrotic markers. **j** Co-immunofluorescence staining for α-SMA (green) and COL1 (red). **k, l** Quantification of the mean fluorescence intensity for α-SMA (**k**) and COL1 (**l**). In all fluorescence images (**f, j**), nuclei are counterstained with DAPI (blue). Scale bars in the representative images (**c, f, h, j**), 100 μm. In (**d, e, g, i, k, l**), data are presented as mean ± SD from $n = 6$ biologically independent animals. Statistical significance was determined by one-way ANOVA with Tukey's post hoc test; *$P < 0.05$, **$P < 0.01$, ***$P < 0.001$, ****$P < 0.0001$. Detailed statistical analyses, including exact P values, test statistics (F values and degrees of freedom), and 95% confidence intervals, are provided in the Source Data file.

hydrogel's 14-day degradation window (Fig. 4j and Supplementary Fig. 6c) to ensure therapeutic action throughout the critical phase of tissue remodeling. The resulting composite hydrogel exhibited ideal characteristics for minimally invasive intrauterine application, including shear-thinning behavior for injectability and rapid thermosensitive gelation at body temperature (Fig. 4b–d). Furthermore, its modest swelling ratio of under 20% (Supplementary Fig. 6b) ensures it can fill the uterine cavity and maintain intimate tissue contact without exerting harmful pressure. This integrated design creates a sophisticated, multifunctional system that combines the potent, endogenous biomimetic cues of the decidual ECM with the spatiotemporally controlled delivery of a powerful anti-fibrotic agent, establishing a robust platform for functional endometrial repair.

While direct in vivo tracking via fluorescent labeling is a common evaluation method, we recognized the significant risk that covalently modifying our complex, bioactive dECM scaffold could alter its native therapeutic properties or introduce labeling artifacts. Therefore, we contend that the most compelling evidence for the hydrogel's appropriate in vivo residence time is its demonstrated therapeutic efficacy. The significant improvements in endometrial regeneration, neo-vascularization, and ultimately, the restoration of fertility (Figs. 6 and 7), would be mechanistically implausible if the scaffold's degradation profile were unsuitable. A scaffold that degrades too rapidly would fail to guide tissue repair, while one that persists too long would impede tissue integration and likely elicit a chronic inflammatory response. Thus, the robust functional success observed in vivo strongly corroborates our in vitro findings, confirming that the hydrogel possesses a degradation rate finely tuned for its intended therapeutic application.

A key strength of our therapeutic design is the strategic "division of labor" between the hydrogel scaffold and the drug delivery microspheres, and our in vitro assays were meticulously designed to validate these distinct roles.

First, we confirmed the potent, immediate pro-regenerative capacity of the DEndo-UdECM scaffold itself—the system's "pro-regenerative engine." As hypothesized, DEndo-UdECM significantly outperformed CEndo-UdECM in promoting endometrial epithelial cell proliferation, stromal cell migration, and endothelial cell (HUVEC) tube formation (Fig. 5). At first glance, these potent pro-regenerative effects might seem paradoxical, given the terminally differentiated nature of the mature decidua. However, this apparent conflict is resolved when decidualization is viewed not as a static end-point, but as a dynamic, multi-phase process fundamentally analogous to controlled, scar-free wound healing[48,50]. Authoritative reviews highlight that the initial phase of decidualization is inherently proliferative and involves extensive tissue remodeling, with stromal cells first undergoing "extensive proliferation and differentiation" to build the necessary architecture[48]. Therefore, our findings do not contradict the biomimetic premise; on the contrary, they directly confirm that DEndo-UdECM successfully recapitulates the crucial, pro-regenerative signals of this early decidualization process, making it an ideal "engine" for therapeutic wound repair[19]. Conversely, we validated the critical role of Tβ4 as the system's long-term "anti-fibrotic brake." In a

dedicated assay mimicking a sustained pro-fibrotic challenge, the controlled release of Tβ4 effectively suppressed TGF-β1-induced fibrotic activation in endometrial stromal cells in vitro (Supplementary Fig. 2), consistent with its well-established anti-fibrotic functions[51]. This strategic design, combining a rapid pro-regenerative "engine" with a sustained anti-fibrotic "brake," provides a clear mechanistic rationale for the profound synergistic effects observed in our in vivo model, where both rapid healing and prevention of chronic scarring are essential for functional recovery.

Our in vivo investigation in a severe, ethanol-induced IUA model unequivocally demonstrates the superior therapeutic potential of the DEndo-UdECM/Tβ4@PLGA hydrogel[52]. This composite system orchestrated a multi-faceted repair process that significantly outperformed all other control and treatment groups. Histological analysis revealed a profound restoration of endometrial architecture, evidenced by a near-complete recovery of endometrial thickness and, critically, the regeneration of abundant, well-formed glands (Fig. 6b–e). This structural regeneration was underpinned by a potent anti-fibrotic effect; the DEndo-UdECM/Tβ4@PLGA treatment most effectively resolved the pathological collagen deposition and suppressed the expression of key fibrotic mediators α-SMA and COL1 (Fig. 6h–l). The integrated success in both promoting regeneration and inhibiting fibrosis highlights a powerful synergy between the bioinstructive, decidualized ECM niche and the sustained delivery of the anti-fibrotic peptide Tβ4[21,23,24]. Furthermore, the robust re-expression of the master transcription factor FOXA2 (Fig. 6f, g) signifies more than mere architectural rebuilding[53]; it indicates the re-establishment of glandular functional competence, which is indispensable for producing the histotroph required for embryo survival and implantation[41,54].

While structural and molecular restoration is crucial, the ultimate benchmark for any IUA therapy is the recovery of fertility[9]. Here, in rigorous mating studies, the DEndo-UdECM/Tβ4@PLGA treatment group exhibited a near-complete restoration of reproductive function, with implantation site numbers and live birth rates that were statistically indistinguishable from those of healthy, sham-operated animals (Fig. 7a–e). The birth of healthy, normally developing offspring further validates the quality and functionality of the regenerated endometrium (Fig. 7f). This result provides the most compelling evidence that our therapeutic strategy achieves true functional repair, capable of fully supporting all stages of pregnancy.

Crucially, this therapeutic efficacy is not compromised by safety concerns, a paramount consideration for clinical translation. A comprehensive safety evaluation demonstrated that the hydrogel materials induced no signs of systemic toxicity, with all major organs maintaining normal histological architecture (Supplementary Fig. 3a). More importantly, a dedicated subcutaneous implantation study confirmed the material's local biocompatibility. The hydrogels elicited only a mild, transient inflammatory response that fully resolved over time, and critically, there was no evidence of a chronic foreign body reaction or fibrotic capsule formation (Supplementary Fig. 3b, c). This intrinsic safety profile ensures that the hydrogel itself does not contribute to the inflammatory or fibrotic burden, but rather serves as a

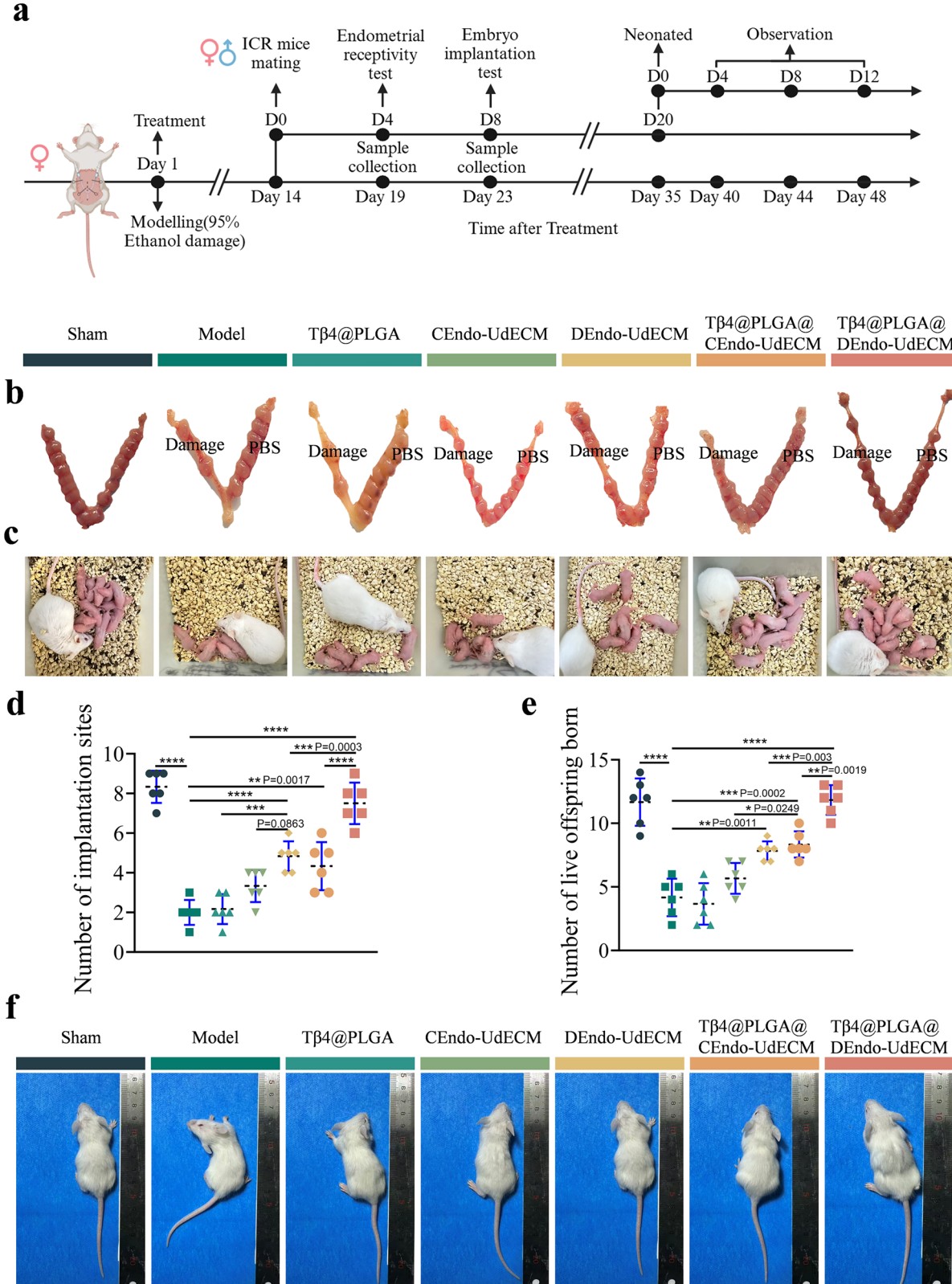

benign and effective platform to orchestrate a pro-regenerative microenvironment.

Our mechanistic investigation reveals that the superior efficacy of the DEndo-UdECM/Tβ4@PLGA system stems from its ability to orchestrate a complex, multi-pronged healing response. A key initial action is the potent reprogramming of the local immune microenvironment[55]. Our data show that the composite hydrogel dramatically shifts the macrophage population from a pro-inflammatory M1 phenotype (CD86+, iNOS+) towards a pro-regenerative M2 state (Arginase-1+, CD206+), creating a foundational anti-inflammatory niche necessary for successful repair[56] (Fig. 8). Crucially, this effect significantly surpassed that of the CEndo-UdECM/Tβ4@PLGA group, and DEndo-UdECM alone demonstrated an intrinsically stronger M2-polarizing capacity than CEndo-UdECM. This

**Fig. 7 | The DEndo-UdECM/Tβ4@PLGA hydrogel restores fertility in a murine injury model. a** Schematic of the fertility assessment timeline and experimental groups (Created in BioRender. Zhaowei, Y. (2025) https://BioRender.com/ul4iqzn). **b** Representative images of uterine horns showing embryo implantation sites (indicated by swelling) on Gestational Day 8 (GD8). The labels "Damage" and "PBS" denote the treated (right) and untreated control (left) uterine horns, respectively. **c** Representative images of live offspring born to treated female mice. **d**, **e** Quantification of implantation sites in the treated uterine horn (**d**) and the number of live offspring born per litter (**e**). **f** Representative images of weaned pups

from all treatment groups that produced live offspring. Pups from all groups that produced live offspring showed 100% survival and normal development (see details in Table S2). In (**d**, **e**), data are presented as scatter dot plots from $n = 6$ biologically independent animals, with data shown as mean ± s.d. Statistical significance was determined by one-way ANOVA with Tukey's post hoc test; *$P < 0.05$, **$P < 0.01$, ***$P < 0.001$, ****$P < 0.0001$. Detailed statistical analyses, including exact $P$ values, test statistics ($F$ values and degrees of freedom), and 95% confidence intervals, are provided in the Source Data file.

suggests that the decidualized ECM is not a passive scaffold but an active immunomodulatory agent[57].

This re-programmed immune niche, in turn, appears to mobilize the endometrium's endogenous regenerative machinery[28]. We observed a broad-spectrum stimulation of multiple repair facets in the DEndo-UdECM/Tβ4@PLGA group, including enhanced epithelial regeneration (pan-CK)[58], a significant increase in potential stem/progenitor cell populations (ALDH1A1)[13], robust angiogenesis (CD34)[6], and normalized cell proliferation (Ki67) (Fig. 9). The consistent superiority of this group over all others, including the CEndo-UdECM/Tβ4@PLGA cohort, points to a powerful synergy where the unique bioactive cues within the DEndo-UdECM niche are amplified by the supportive actions of Tβ4 to maximally activate the tissue's intrinsic healing capacity[26,57].

At the molecular level, our findings indicate that this regenerative cascade is driven by the dual suppression of two master pathological pathways[52]. The DEndo-UdECM/Tβ4@PLGA treatment most effectively quenched the pro-inflammatory fire of the pyroptosis pathway—reducing GSDMD-NT cleavage, Pro-caspase-1 activation, and downstream IL-1β/IL-18 production—while simultaneously dismantling the canonical TGF-β/Smad3 fibrotic signaling cascade (Fig. 10). The magnitude of this dual inhibition provides a clear molecular basis for the profound anti-inflammatory and anti-fibrotic outcomes observed in vivo[10]. Intriguingly, the intrinsic superiority of DEndo-UdECM over CEndo-UdECM in dampening these pathways suggests that its unique matrisome composition may actively disrupt these detrimental feedback loops[59].

The culmination of this orchestrated repair process—immunomodulation, tissue regeneration, and suppression of pathological signaling—is the re-establishment of the molecular signature of endometrial receptivity[60]. The DEndo-UdECM/Tβ4@PLGA treatment uniquely restored the delicate balance of this signature, characterized by the robust upregulation of essential pro-implantation markers (PR, ERα, OPN, Integrin β3) and the concurrent downregulation of anti-receptive factors (PCNA, MUC1) (Supplementary Fig. 9)[61–63]. This comprehensive molecular normalization, which closely mirrors the profile of a healthy endometrium, provides the direct mechanistic link between the hydrogel's multi-faceted biological actions and the ultimate functional outcome: the near-complete restoration of fertility.

Our study highlights a central trade-off in ECM-based therapies: decellularization retains the structural scaffold but removes soluble bioactive molecules. We deliberately focused on the insoluble matrix, and our results compellingly demonstrate that the DEndo-UdECM scaffold alone possesses a potent intrinsic bioactivity, sufficient to drive functional endometrial regeneration. This focus, however, defines a primary limitation: the absence of the full decidual secretome. A logical future direction is therefore the development of "next-generation" hybrid systems, combining the structural cues of our ECM with defined recombinant growth factors or cytokines to potentially unlock synergistic therapeutic effects.

Beyond the biomaterial itself, we acknowledge the limitations in our preclinical models. Our initial in vitro screening utilized Ishikawa and hESC cell lines to represent epithelial and stromal components.

While an established approach for preliminary assessment, these immortalized lines cannot fully recapitulate the complex signaling of a native endometrium. Therefore, our central claims of functional regeneration are rightly substantiated by the comprehensive in vivo data.

Finally, while our hydrogel has demonstrated potent efficacy, we acknowledge a common challenge for all natural biomaterials: potential batch-to-batch variability[44]. In anticipation of this, we established a rigorous, standardized production and quality control (QC) protocol. This protocol, including standardized assessments of final protein concentration, residual DNA content, and rheological properties, was implemented to minimize variance and ensure the reproducibility of our findings. Such stringent QC represents a foundational step towards future clinical translation.

In this study, we introduce and validate a pioneering "decidualization-empowerment" strategy for functional uterine repair. We demonstrate that a composite hydrogel, synergistically integrating a uniquely bioactive decidualized endometrial ECM with the sustained delivery of the anti-fibrotic peptide Tβ4, can effectively reverse the pathological hallmarks of severe IUA. This system does not merely provide a scaffold but actively orchestrates a complex pro-regenerative program: it reprograms the local immune microenvironment towards an M2 phenotype, quenches pathological inflammation and fibrosis by dually inhibiting the pyroptosis and TGF-β/Smad3 pathways, and ultimately restores the molecular signature of endometrial receptivity.

The unequivocal success of this approach, culminating in the near-complete restoration of fertility in a stringent preclinical model, establishes a benchmark for IUA therapy. In essence, our work provides a blueprint for leveraging the intrinsic biological intelligence of a physiological niche to resolve complex fibrotic diseases. It moves beyond simple scaffolding to active, instructive tissue engineering, offering a therapeutic paradigm for reproductive medicine. While further validation in larger animal models is necessary to pave the way for clinical translation, this study lays a robust foundation for developing next-generation biomaterials capable of tackling the formidable challenge of functional organ regeneration.

## Methods
### Animal experiments
Sexually mature female ICR mice (6–8 weeks old, $25–30 \times g$), vasectomized male ICR mice (8–10 weeks old), and fertile male ICR mice (8–10 weeks old) were used in this study. All animals were procured from the Experimental Animal Center of Shanxi Medical University (Animal Production License No: SCXK(Jin)2024-0004) and housed under specific pathogen-free conditions at Shanxi Medical University. The animal facility was maintained at $23 \pm 2\,°C$ with a humidity of 40–70% and a 14 h light/10 h dark cycle. All mice had ad libitum access to standard chow and water.

### Materials
Poly(lactic-co-glycolic acid) (PLGA, LA:GA = 50:50, Mw 15 kDa, cat. no. P1066) was purchased from Jinan Daigang Biomaterial. Polyvinyl Alcohol (PVA, Mw 30–70 kDa, cat. no. 363138), sesame oil (cat. no. S3507), Triton ×-100 (cat. no. T8787), Sodium Dodecyl Sulfate (SDS,

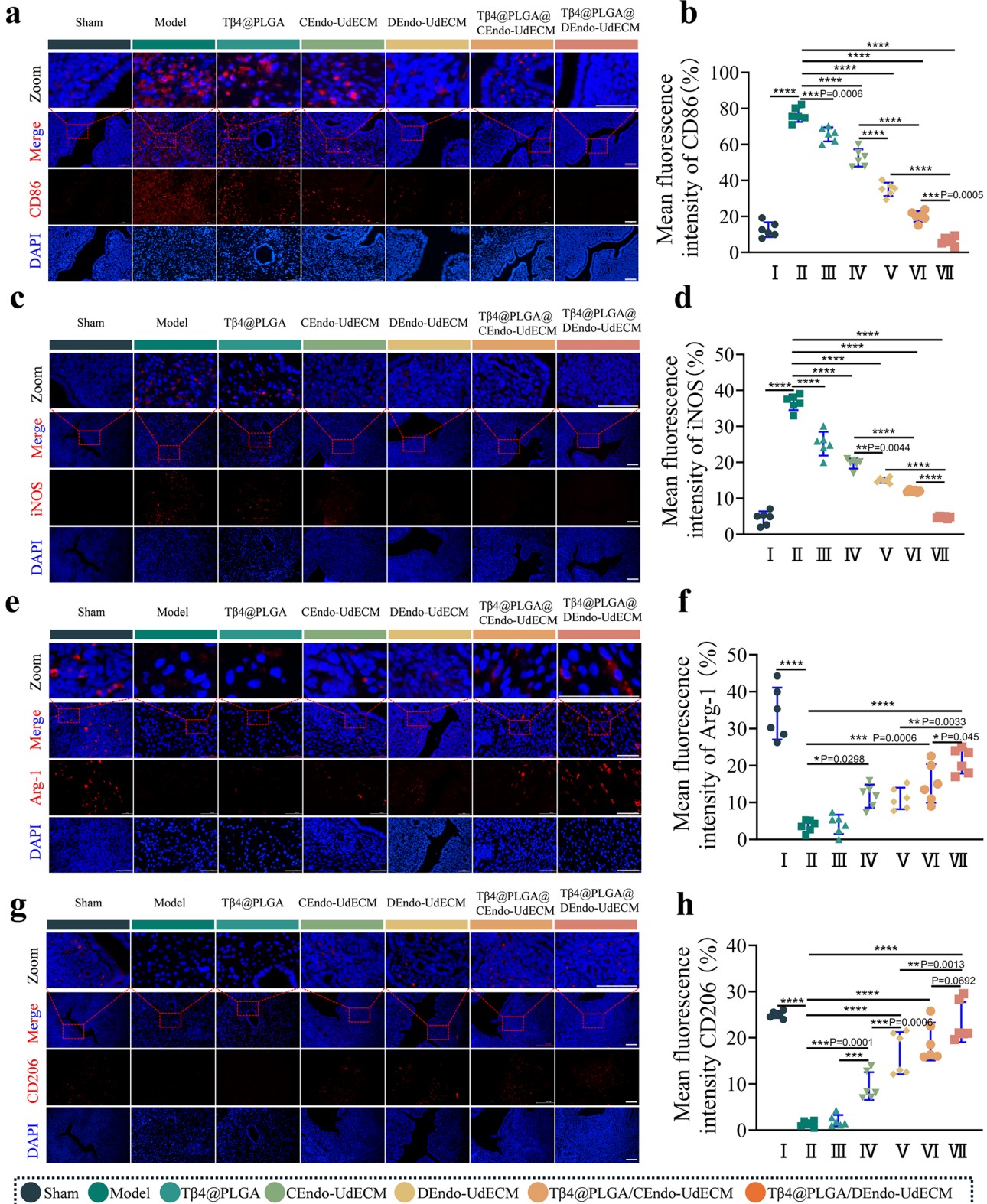

**Fig. 8 | The DEndo-UdECM/Tβ4@PLGA hydrogel remodels the immune microenvironment by shifting macrophage polarization towards a pro-regenerative M2 phenotype.** Immunofluorescence analysis of macrophage polarization in regenerated endometrial tissue 14 days post-treatment. **a–d** Representative images and corresponding quantification of M1-associated markers: CD86 (**a**, **b**) and inducible nitric oxide synthase (iNOS; **c**, **d**). **e–h** Representative images and corresponding quantification of M2-associated markers: Arginase-1 (Arg-1; **e**, **f**) and CD206 (**g**, **h**). In all immunofluorescence panels, markers are shown

in red, and cell nuclei are counterstained with DAPI (blue). Each panel includes high-magnification insets (Zoom) of the boxed areas. Scale bars, 100 μm. In (**b**, **d**, **f**, **h**), data are presented as scatter dot plots from n = 6 biologically independent animals, with data shown as mean ± s.d. Statistical significance was determined by one-way ANOVA with Tukey's post hoc test; *P < 0.05, **P < 0.01, ***P < 0.001, ****P < 0.0001. Detailed statistical analyses, including exact P values, test statistics (F values and degrees of freedom), and 95% confidence intervals, are provided in the Source Data file.

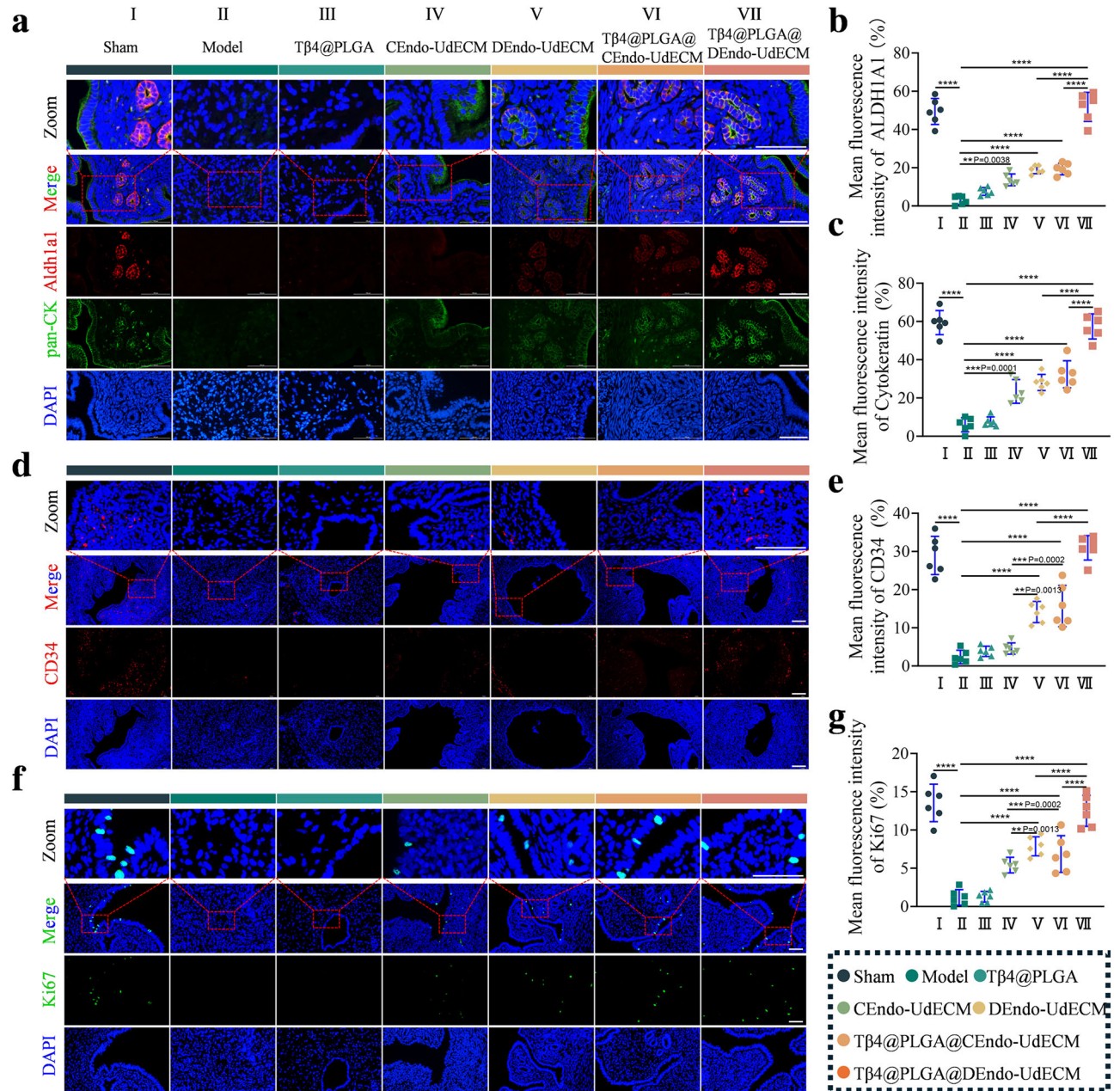

**Fig. 9 | DEndo-UdECM/Tβ4@PLGA hydrogel promotes multi-lineage endometrial regeneration.** Uterine tissues were analysed 14 days post-treatment across all seven experimental groups. **a** Representative immunofluorescence co-staining for the progenitor marker ALDH1A1 (red) and epithelial marker pan-Cytokeratin (pan-CK, green). Insets show high-magnification views. **b**, **c** Quantification of mean fluorescence intensity (MFI) for ALDH1A1 (**b**) and pan-CK (**c**). **d** Representative immunofluorescence for the angiogenesis marker CD34 (red). **e** Quantification of MFI for CD34. **f** Representative immunofluorescence for the proliferation marker

Ki67 (green). **g** Quantification of MFI for Ki67. In all immunofluorescence images, nuclei were counterstained with DAPI (blue). Scale bars, 100 μm. For all quantifications, data are presented as mean ± s.d. ($n$ = 6 mice per group); statistical significance was assessed by one-way ANOVA with Tukey's post-hoc test. *$P$ < 0.05, **$P$ < 0.01, ***$P$ < 0.001, ****$P$ < 0.0001. Detailed statistical analyses, including exact $P$ values, test statistics ($F$ values and degrees of freedom), and 95% confidence intervals, are provided in the Source Data file.

cat. no. L3771), Pepsin (cat. no. P7000), Gelatin (cat. no. G9391), and Endothelial Cell Growth Supplement (ECGS) were obtained from Sigma-Aldrich. DNase I (cat. no. EN0521) was from Thermo Fisher Scientific. Dichloromethane (DCM, cat. no. A606624), Hydrochloric Acid (HCl, cat. no. A100927), and Sodium Hydroxide (NaOH, cat. no. A500942) were acquired from Sangon Biotech. Recombinant human Thymosin β4 (Tβ4, cat. no. RP10658) was purchased from ABclonal, and Recombinant Human TGF-β1 (cat. no. 100-21) from PeproTech. For cell culture, DMEM/F12 Medium (cat. no. C11330500BT), Fetal Bovine Serum (FBS, cat. no. 10099141), Charcoal Stripped FBS (CS-FBS, cat. no.

12676029), and 0.25% Trypsin-EDTA (cat. no. 25200056) were purchased from Gibco. Hank's Balanced Salt Solution (HBSS, cat. no. H8264) was from Merck, and Penicillin-Streptomycin Solution (cat. no. SV30010) from HyClone. Matrigel Basement Membrane Matrix (cat. no. 356255) was obtained from Corning. For assays and staining, Paraformaldehyde (PFA, cat. no. P1110), H&E Staining Kit (cat. no. G1120), and Masson's Trichrome Staining Kit (cat. no. G1340) were from Solarbio. Calcein-AM/PI Live/Dead Cell Staining Kit (cat. no. C2015S), BeyoClick™ EdU Cell Proliferation Kit (cat. no. C0071S), RIPA Lysis Buffer (cat. no. P0013B), BCA Protein Assay Kit (cat. no. P0012), DAPI

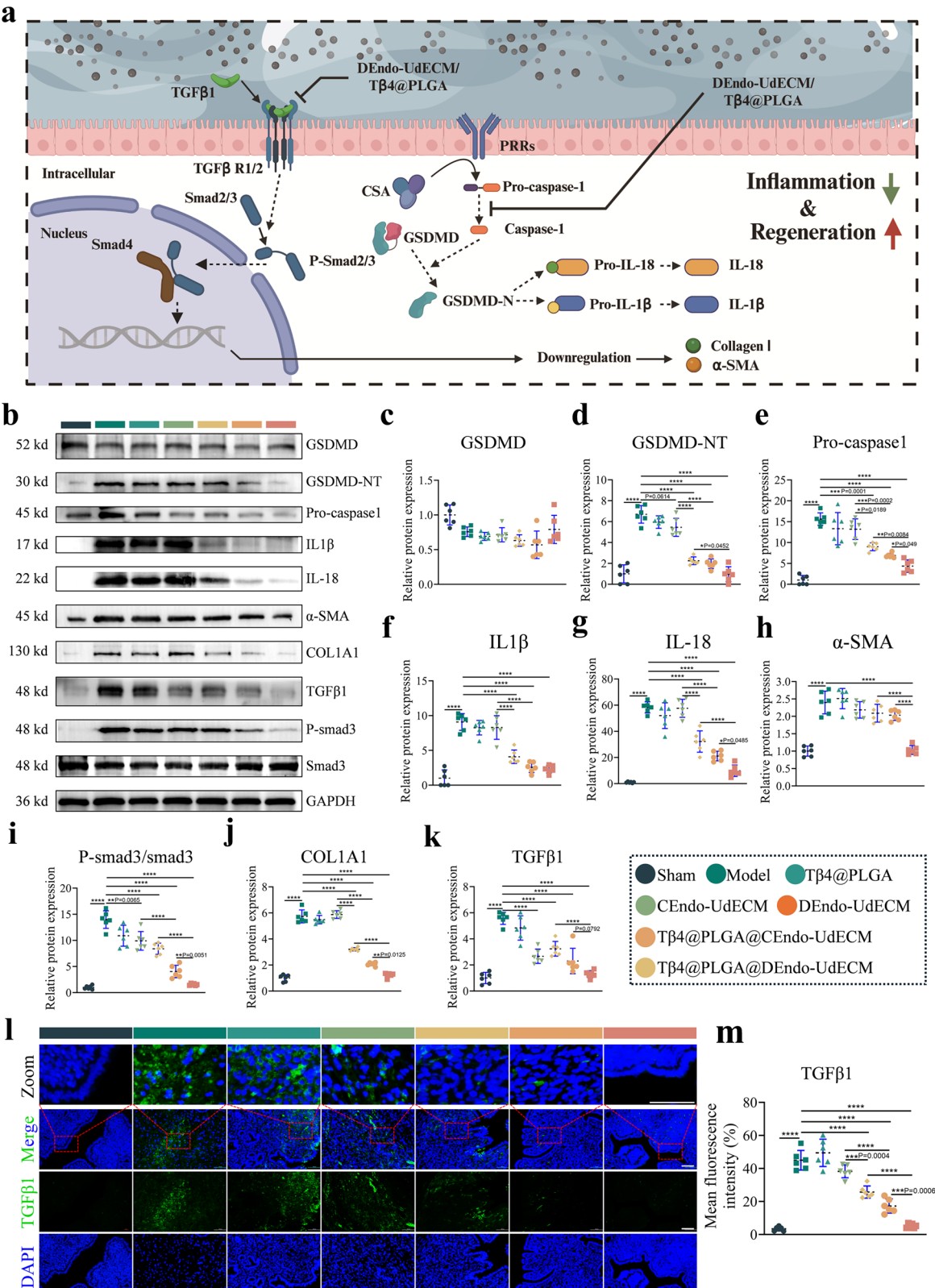

Staining Solution (cat. no. C1005), and Antifade Mounting Medium (cat. no. P0126) were purchased from Beyotime. The Tβ4 ELISA Kit (cat. no. JL19928) was from JONLNBIO, and the DNeasy Blood & Tissue Kit (cat. no. 69504) from Qiagen. Zombie Aqua™ Fixable Viability Kit (cat. no. 423101) was obtained from BioLegend. Detailed information regarding antibodies used for IF, immunohistochemistry (IHC), and WB analyses is provided in Supplementary Table 1.

### Preparation and characterization of decidualized uterine extracellular matrix (DEndo-UdECM)

Pseudopregnancy was induced in sexually mature female ICR mice via mating with vasectomized males (1:1 ratio), with the day of vaginal plug detection designated as day 1 (D1). On D4, artificial decidualization was triggered by intrauterine injection of 10 μL sesame oil into each uterine horn under isoflurane anesthesia. Fully decidualized uteri (DEndo-

**Fig. 10 | The DEndo-UdECM/Tβ4@PLGA hydrogel simultaneously suppresses pyroptosis and canonical TGF-β/Smad signaling. a** Schematic illustrating the dual mechanism of action, where the composite hydrogel inhibits both the TGF-β-driven fibrotic pathway and the pyroptotic inflammatory pathway (Created in BioRender. Zhaowei, Y. (2025) https://BioRender.com/ul4iqzn). **b–k** Western blot analysis of key signaling proteins in endometrial tissue lysates 14 days post-treatment. **b** Representative western blot images. **c–g** Quantification of proteins involved in the pyroptosis pathway: Gasdermin D (GSDMD; **c**), its N-terminal fragment (GSDMD-NT; **d**), pro-caspase-1 (**e**), and the mature inflammatory cytokines IL-1β (**f**) and IL-18 (**g**). **h–k** Quantification of proteins involved in the fibrosis pathway: α-smooth muscle actin (α-SMA; **h**), the ratio of phosphorylated Smad3 (P-Smad3) to total Smad3 (**i**), Collagen Type I Alpha 1 (COL1A1; **j**), and TGF-β1 (**k**). **l, m** Immunofluorescence analysis of TGF-β1 expression in endometrial tissue. **l** Representative images. TGF-β1 is shown in green, and cell nuclei are counterstained with DAPI (blue). High-magnification insets (Zoom) of the boxed areas are provided. Scale bars, 100 μm. **m** Quantification of TGF-β1 mean fluorescence intensity. In all quantitative plots (**c–k, m**), data are presented as scatter dot plots from $n = 6$ biologically independent animals, with data shown as mean ± s.d. Statistical significance was determined by one-way ANOVA with Tukey's post hoc test; *$P < 0.05$, **$P < 0.01$, ***$P < 0.001$, ****$P < 0.0001$. Detailed statistical analyses, including exact $P$ values, test statistics ($F$ values and degrees of freedom), and 95% confidence intervals, are provided in the Source Data file.

Uteri) were harvested on D8, while untreated, age-matched uteri served as controls (CEndo-Uteri). Successful decidualization was verified by histological assessment of endometrial thickening and vascularization, and IF staining for the decidual marker HAND2. For decellularization, fresh tissues were thoroughly rinsed with deionized water and subjected to a sequential protocol consisting of: (1) incubation in 1% Triton ×-100 containing 0.1% EDTA for 1.5 h; (2) incubation in 1% SDS containing 0.1% EDTA for 48 h; and (3) treatment with 200 U/mL DNase I in PBS containing 10 mM MgCl2 at 37 °C for 4 h. All steps except enzymatic digestion were performed at room temperature with continuous agitation, interspersed with extensive PBS washes. The resulting matrices were sterilized in 75% ethanol for 30 min, lyophilized, and ground into powder. Decellularization efficacy was evaluated by H&E and Masson's trichrome staining, and residual DNA was quantified using a DNeasy Blood & Tissue Kit (Qiagen) and a Nanodrop 2000 spectrophotometer (Thermo Fisher Scientific). Retention of key ECM components (COL1A1, COL4, Fibronectin) was confirmed via IF. To compare matrisome profiles, proteins were extracted using SDT lysis buffer (4% SDS, 100 mM Tris-HCl, 0.1 M DTT) and digested via the filter-aided sample preparation (FASP) method using trypsin. Peptides were separated on a C18 analytical column (75 μm × 15 cm) using a 120-min gradient on an UltiMate 3000 RSLCnano system and analyzed on a timsTOF Pro2 mass spectrometer (Bruker Daltonik) in diaPASEF mode. Data were processed using Spectronaut™ Pulsar X software, annotated against the MatrisomeDB database, and differentially expressed proteins (fold change > 2, adjusted $P < 0.05$) were analyzed via GO enrichment.

### Preparation and characterization of decidualized uterine extracellular matrix (DEndo-UdECM)

Pseudopregnancy was induced in sexually mature female ICR mice via mating with vasectomized males (1:1 ratio), with the day of vaginal plug detection designated as day 1 (D1). On D4, artificial decidualization was triggered by intrauterine injection of 10 μL sesame oil into each uterine horn under isoflurane anesthesia[61]. Fully decidualized uteri (DEndo-Uteri) were harvested on D8, while untreated, age-matched uteri served as controls (CEndo-Uteri). Successful establishment of the decidualization model was confirmed through histological staining (H&E, Masson's trichrome) and IF analysis of key molecular markers (CD31, PCNA, PR, HAND2). For decellularization[20,45], fresh tissues were thoroughly rinsed with deionized water and subjected to a sequential protocol consisting of: (1) incubation in 1% Triton ×-100 containing 0.1% EDTA for 1.5 h; (2) incubation in 1% SDS containing 0.1% EDTA for 48 h; and (3) treatment with 200 U/mL DNase I in PBS containing 10 mM MgCl2 at 37 °C for 4 h. All steps except enzymatic digestion were performed at room temperature with continuous agitation, interspersed with extensive PBS washes. The resulting matrices were sterilized in 75% ethanol for 30 min, lyophilized, and ground into powder. Decellularization efficacy was evaluated by H&E and Masson's trichrome staining, and residual DNA was quantified using a DNeasy Blood & Tissue Kit (Qiagen) and a Nanodrop 2000 spectrophotometer (Thermo Fisher Scientific). Retention of key ECM components

(COL1A1, COL4, Fibronectin) was confirmed via IF. To compare matrisome profiles, proteins were extracted using SDT lysis buffer (4% SDS, 100 mM Tris-HCl, 0.1 M DTT) and digested via the FASP method using trypsin. Peptides were separated on a C18 analytical column (75 μm × 15 cm) using a 120-min gradient on an UltiMate 3000 RSLCnano system and analyzed on a timsTOF Pro2 mass spectrometer (Bruker Daltonik) in diaPASEF mode. Data were processed using Spectronaut™ Pulsar X software, annotated against the MatrisomeDB database[64], and differentially expressed proteins (fold change > 2, adjusted $P < 0.05$) were analyzed via GO enrichment and Kyoto Encyclopedia of Genes and Genomes pathway using online tools such as DAVID or Metascape.

### Fabrication of Tβ4@PLGA microspheres

Tβ4-loaded PLGA microspheres (Tβ4@PLGA) were fabricated using an optimized water-in-oil-in-water (W/O/W) double emulsion solvent evaporation technique. Specifically, Tβ4 was dissolved in PBS to serve as the inner aqueous phase (W1, 5% w/v), while PLGA was dissolved in DCM to form the oil phase (O, 5% w/v). Under ice-bath conditions, 0.5 mL of W1 was emulsified into 4 mL of O using a probe sonicator (35% power, pulse mode: 4 s on/4 s off, 5 min) to generate the primary W/O emulsion. This primary emulsion was immediately injected into 200 mL of deionized water containing 2% (w/v) PVA (outer aqueous phase, W2) under stirring at 800 rpm to form the secondary W/O/W emulsion. The system was continuously stirred at 600 rpm for >12 h to allow for complete solvent evaporation and microsphere solidification. The resulting Tβ4@PLGA microspheres were collected via centrifugation, washed three times with deionized water to remove residual PVA and unencapsulated protein, and lyophilized for at least 10 h. The final powder was stored at −80 °C. Blank PLGA microspheres were prepared using the same procedure without the addition of Tβ4.

### Characterization of Tβ4@PLGA microspheres

The surface morphology of the microspheres was examined using scanning electron microscopy (SEM, ZEISS GeminiSEM 300, Germany) after sputter-coating with gold. Particle size distribution was determined using a dynamic light scattering instrument (DLS, Malvern Mastersizer 2000, UK) after resuspending the microspheres in water. To quantify LC and EE, a precise amount of lyophilized microspheres was dissolved in DCM, and Tβ4 was extracted into an aqueous buffer. The Tβ4 concentration was measured using a commercial ELISA kit according to the manufacturer's instructions. LC and EE were calculated using the following formulas:

$$LC(\%) = (\text{weight of encapsulated Tβ4}/\text{weight of microspheres}) \times 100\% \quad (1)$$

$$EE(\%) = (\text{weight of encapsulated Tβ4}/\text{initial input Tβ4}) \times 100\% \quad (2)$$

For in vitro release kinetics, 10 mg of microspheres were incubated in 1 mL of PBS (pH 7.4) at 37 °C with constant shaking (100 rpm). At predetermined time points (days 0, 1, 2, 3, 5, 7, and 10), the supernatant was collected for ELISA analysis and immediately replaced with

fresh pre-warmed PBS. Degradation behavior was assessed by incubating 20 mg of microspheres in 1 mL of PBS at 37 °C with gentle agitation (300 rpm). At defined intervals, aliquots were imaged using an inverted microscope (Nikon Ti2) to observe hydrated morphology. Parallel samples were washed with deionized water, lyophilized, and examined using SEM (Phenom ProX) at an accelerating voltage of 10 kV to monitor morphological erosion.

## Preparation of DEndo-UdECM/Tβ4@PLGA composite hydrogels

Lyophilized UdECM powder was solubilized in 0.01 M HCl at a concentration of 25 mg/mL containing pepsin (2.5 mg/mL; matrix-to-enzyme ratio of 10:1, w/w) and stirred magnetically at room temperature for 5 days to ensure limited digestion. Following digestion, the pH was neutralized to 7.2–7.4 using 10 N NaOH, and the ionic strength was adjusted with 10× PBS ($Ca^{2+}/Mg^{2+}$-free) to yield a final UdECM concentration of 10 mg/mL. The resulting pre-gel solution was maintained on ice to prevent premature gelation. Control 1% gelatin solutions were prepared in 1× PBS. To fabricate the composite hydrogels (DEndo-UdECM/Tβ4@PLGA and CEndo-UdECM/Tβ4@PLGA), pre-weighed Tβ4@PLGA microspheres were dispersed into the ice-cold UdECM pre-gel solution. The mixture was homogenized by gentle pipetting to ensure uniform distribution without introducing air bubbles.

## Physicochemical and functional characterization of composite hydrogels

Rheological properties were analyzed using a rotational rheometer (Anton Paar MCR 302) with a 25 mm parallel plate geometry (1 mm gap). Gelation kinetics were monitored via time-sweep oscillatory tests (1 Hz, 1% strain) as the temperature was ramped from 4 °C to 37 °C. Upon reaching equilibrium, frequency sweep assays (0.1–10 Hz, 1% strain) were conducted to determine the viscoelastic properties of the formed hydrogels. For microstructural analysis, hydrogels formed at 37 °C were fixed in 2.5% glutaraldehyde, dehydrated through a graded ethanol series (30–100%), and critical point dried or lyophilized. The internal morphology was visualized by SEM (ZEISS GeminiSEM 300) after gold sputter-coating. Injectability was assessed by extruding the pre-gel solution through a 27 G needle.

In vitro degradation was evaluated by incubating weighed hydrogels (approx. 100 μL) in PBS containing 0.01% $NaN_3$ at 37 °C with agitation. The remaining wet weight was recorded at defined intervals over 14 days. Swelling behavior was determined gravimetrically by immersing lyophilized hydrogels in PBS at 37 °C and calculating the swelling ratio [(Ws − Wd)/Wd × 100%] over 48 h. To quantify Tβ4 release, composite hydrogels (100 μL) were incubated in PBS at 37 °C. Supernatants were collected at predetermined time points, replaced with fresh buffer, and analyzed using a Tβ4 ELISA kit. Cumulative release was calculated relative to the total encapsulated protein. For 3D structural visualization, Rhodamine B-labeled Tβ4@PLGA microspheres were encapsulated within a FITC-labeled DEndo-UdECM hydrogel. Z-stack images were acquired using a spinning-disk confocal microscope (Evident SIM-ultimate) and reconstructed using cellSens software to visualize the spatial distribution of microspheres within the matrix.

## Cell culture

Human endometrial stromal cells (hESC, CRL-4003) were obtained from ATCC. The human endometrial adenocarcinoma cell line (Ishikawa) was kindly provided by Prof. Zengming Yang (Guizhou University), and human umbilical vein endothelial cells (hUVEC) were provided by Dr. Jianbing Liu (Shanxi Medical University). hESCs were cultured in DMEM/F12 supplemented with 10% charcoal-stripped fetal bovine serum (CS-FBS) and 1% penicillin-streptomycin (P/S). Ishikawa cells were maintained in DMEM/F12 containing 10% FBS and 1% P/S, while hUVECs were cultured in DMEM/F12 supplemented with 10% FBS, 30 μg/mL ECGS, and 0.1 mg/mL heparin. All cells were incubated at 37 °C in a humidified atmosphere containing 5% $CO_2$. The hESC line was authenticated by the supplier via short tandem repeat profiling. The Ishikawa and hUVEC lines were verified in-house based on cellular morphology and specific marker expression. All cell lines tested negative for mycoplasma contamination. Experiments were performed using hESCs at passages 3–8 and hUVECs at passages 3–6.

## In vitro anti-fibrotic and cell migration assays

To assess anti-fibrotic efficacy, hESCs ($5 \times 10^4$ cells/well) were seeded in 24-well plates and subjected to one of three conditions: basal medium (Control), medium containing 10 ng/mL TGF-β1[65] (Fibrosis Model), or medium containing 10 ng/mL TGF-β1 supplemented with 100 nM Tβ4[66] (Treatment). After 48 h, cells were fixed and immunostained for fibrosis markers α-SMA and COL1A1 (antibody details in Supplementary Table 1). Fluorescence intensity was quantified from at least five random fields using ImageJ software (NIH). For migration analysis, a scratch assay was performed on confluent hESC monolayers ($2 \times 10^5$ cells/well). Following the creation of a cell-free gap using a pipette tip, cells were washed and incubated in low-serum medium (1% CS-FBS) containing extracts from different hydrogels (prepared by incubating hydrogels in serum-free medium for 24 h). Wound closure was monitored at 0, 6, and 12 h using an inverted microscope (Nikon Ti2), and migration distance was analyzed using ImageJ.

## Cell proliferation and cytotoxicity analyses

A Transwell co-culture system (0.4 μm pore size, Corning) was established to evaluate the paracrine effects of the hydrogels. Pre-formed hydrogels (100 μL; CEndo-UdECM, DEndo-UdECM, or Gelatin) were placed in the upper chambers, while Ishikawa cells ($2 \times 10^4$ cells/well) were seeded in the lower chambers. For proliferation analysis, cells were co-cultured for 48 h and then incubated with 10 μM 5-ethynyl-2′-deoxyuridine (EdU) for 2 h using the BeyoClick™ EdU Cell Proliferation Kit with Alexa Fluor 488. Alternatively, cell proliferation was assessed via Ki67 IF staining. The percentage of proliferative cells ($EdU^+$ or $Ki67^+$) relative to total nuclei ($DAPI^+$) was calculated. Cytotoxicity was evaluated using a Calcein-AM/PI Live/Dead Cell Staining Kit. After 48 h of co-culture, cells were stained with Calcein-AM (2 μM) and propidium iodide (PI, 4 μM). Live (green) and dead (red) cells were visualized using fluorescence microscopy.

## In vitro angiogenesis assays

The angiogenic potential of the hydrogels was assessed using a tube formation assay. hUVECs ($1 \times 10^4$ cells/well) were seeded onto Matrigel-coated (30 min at 37 °C) 24-well plates. Transwell inserts containing the respective hydrogels were added to the wells to initiate co-culture. Tubular network formation was imaged at 3, 6, and 12 h using a Nikon Ti2 inverted microscope. Quantitative analysis was performed on binary images using the Angiogenesis Analyzer plugin in ImageJ. Key parameters, including total tube length, coverage area, number of junctions, and number of meshes, were calculated from three to five random fields per well.

## Animal models and study design

A murine model of intrauterine adhesion (IUA) was established using mechanical and chemical injury[9,21,67,68]. Female mice were anesthetized with isoflurane, and a midline abdominal incision was made to expose the left uterine horn. The uterine segment was clamped, and 50 μL of 95% ethanol was infused into the lumen for 1 min, followed by thorough flushing with sterile saline. The contralateral horn served as an uninjured internal control. Mice were randomized into seven groups (n = 24 per group): Sham, Model (PBS), Tβ4@PLGA (microspheres only), CEndo-UdECM, DEndo-UdECM, CEndo-UdECM/Tβ4@PLGA, and DEndo-UdECM/Tβ4@PLGA. Immediately post-injury, 30 μL of the respective therapeutic agents or controls were injected into the uterine cavity. To evaluate in vivo biocompatibility, healthy female ICR

mice ($n = 9$ per group) received subcutaneous dorsal injections (100 µL) of PBS, Tβ4@PLGA@CEndo-UdECM, or Tβ4@PLGA@DEndo-UdECM. Skin tissues were harvested at days 7, 14, and 21 for histological examination.

### Histological and morphometric analysis

Uterine tissues ($n = 6$ biologically independent animals per group) were harvested 14 days post-surgery, fixed in 4% PFA for 24 h, and embedded in paraffin following standard dehydration through a graded ethanol series. Serial transverse sections (5 µm) were stained with H&E to evaluate tissue architecture, glandular density, and inflammatory infiltration. Endometrial thickness was quantified using ImageJ software by averaging four equidistant perpendicular measurements taken from the myometrial-endometrial junction to the luminal surface in sections displaying a patent lumen. In cases of severe injury manifesting as luminal obliteration (observed in the Model and Tβ4@PLGA groups), endometrial thickness was biologically defined as zero to reflect regenerative failure, as illustrated in Supplementary Fig.4. To assess fibrosis, sections were subjected to Masson's trichrome staining. The endometrial stroma was manually delineated as the region of interest (ROI), strictly excluding the myometrium, and collagen deposition was visualized as blue staining. The area of fibrosis was isolated using the color deconvolution plugin in ImageJ, and the fibrosis percentage was calculated as the ratio of the blue-stained area to the total endometrial ROI area. Quantitative analyses were derived from at least three non-consecutive sections per animal, with 3–5 random non-overlapping fields evaluated per section.

### Flow cytometry analysis of macrophage polarization

Uterine tissues ($n = 3$ biologically independent animals per group) were harvested, minced into small fragments (~1–2 mm³), and dissociated using a sequential enzymatic digestion protocol. Tissues were incubated in HBSS containing 1% trypsin, 6 mg/mL dispase, and 1 mg/mL DNase I for 1 h at 4 °C, 1 h at room temperature, and 10 min at 37 °C, followed by secondary digestion with 0.15 mg/mL collagenase I for 35 min at 37 °C with gentle agitation. The reaction was neutralized with FACS buffer (DPBS with 2% FBS and 1 mM EDTA), and suspensions were filtered through a 70-µm strainer. Single-cell suspensions ($1–2 \times 10^6$ cells) were stained with the Zombie Aqua™ Fixable Viability Kit to exclude dead cells and incubated with anti-mouse CD16/32 to block Fc receptors. Surface staining was performed on ice for 30 min using fluorochrome-conjugated antibodies against CD45, F4/80, and CD86. Cells were subsequently fixed with 4% PFA, permeabilized with 0.2% Triton ×-100, and stained intracellularly with anti-CD206 for 35 min. Data were acquired on a BD FACSAria™ Fusion Flow Cytometer and analyzed using FlowJo software (v10.8).

The gating strategy was performed as follows: (1) exclusion of debris based on forward scatter (FSC) and side scatter; (2) exclusion of doublets using FSC-H versus FSC-A; (3) identification of live cells (Zombie Aqua-negative); (4) selection of leukocytes (CD45⁺); and (5) identification of macrophages (F4/80⁺). Within the F4/80⁺ population, M1-like (CD86⁺) and M2-like (CD206⁺) phenotypes were quantified based on fluorescence minus one controls. Representative gating strategies are provided in Supplementary Fig. 8. Detailed antibody information is listed in Supplementary Table 1.

### Reproductive performance and endometrial receptivity

To assess functional fertility restoration, female mice ($n = 6$ biologically independent animals per group) were co-housed with proven fertile male ICR mice (1:1 ratio) two weeks post-surgery. Vaginal plugs were monitored daily to confirm mating. On gestation day 19 (D19), pregnant dams were euthanized, and the number of viable pups was recorded. For implantation analysis, a separate cohort ($n = 6$ per group) was mated two weeks post-treatment. The presence of a vaginal plug was designated as gestation day 1 (D1). On D8, the uteri were

harvested, and the number of visible implantation sites, identified as distinct swellings along the uterine horns, was quantified.

To evaluate endometrial receptivity during the implantation window, uterine tissues were collected from a third cohort ($n = 6$ per group) on day 4 post-coitus (D4). The expression of key receptivity markers—including progesterone receptor (PR), estrogen receptor alpha (ERα), integrin β3, OPN, PCNA, and mucin 1 (MUC1)—was analyzed via IHC or IF. The detection method for each marker was selected based on validation assays to maximize signal specificity and preservation of morphological context. For IHC, antigen-antibody complexes were visualized using DAB chromogen and counterstained with hematoxylin. For IF, sections were incubated with appropriate fluorochrome-conjugated secondary antibodies and counterstained with DAPI. Quantitative analysis was performed on representative images based on staining intensity or the percentage of positive cells. Detailed antibody information is provided in Supplementary Table 1.

### Immunohistochemistry, immunofluorescence, and Western blotting

For histological analysis, uterine tissues were fixed, embedded in paraffin, and sectioned at 5 µm. Sections were deparaffinized in xylene, rehydrated through graded ethanol, and subjected to antigen retrieval using citrate buffer (pH 6.0) or EDTA buffer (pH 9.0) via heat induction. Non-specific binding was blocked using 5% normal serum for 1 h at room temperature.

For IF, sections were incubated with specific primary antibodies (detailed in Supplementary Table 1) overnight at 4 °C. Following washing, sections were incubated with species-specific Alexa Fluor-conjugated secondary antibodies for 1 h at room temperature. Nuclei were counterstained with DAPI, and images were acquired using a Nikon Ti2 fluorescence microscope.

For IHC, endogenous peroxidase activity was quenched prior to blocking. Sections were incubated with primary antibodies overnight at 4 °C. Detection was performed using a horseradish peroxidase (HRP)-conjugated secondary antibody system followed by color development with 3,3'-diaminobenzidine (DAB). Sections were counterstained with hematoxylin and imaged using light microscopy. Quantitative analysis of staining intensity, positive area, or cell counts was performed using ImageJ software.

For Western blotting, frozen uterine tissues were homogenized in RIPA lysis buffer supplemented with protease and phosphatase inhibitors. Lysates were clarified by centrifugation (12,000 rpm, 15 min, 4 °C), and protein concentrations were quantified using a BCA Protein Assay Kit. Equal amounts of protein (20–40 µg) were separated by SDS-PAGE and transferred onto PVDF membranes. Membranes were blocked with 5% non-fat milk or BSA and probed with primary antibodies against GSDMD, Caspase-1, IL-1β, IL-18, Smad3, p-Smad3, and GAPDH overnight at 4 °C (Supplementary Table 1). After incubation with HRP-conjugated secondary antibodies, protein bands were visualized using an enhanced chemiluminescence system (Tanon 4600). Band densities were quantified using ImageJ and normalized to GAPDH levels. Uncropped scans of the blots are provided in the Source Data file.

### Statistical analysis

Statistical analyses were performed using GraphPad Prism 8.0 software. All quantitative data are presented as mean ± standard deviation (SD) derived from at least three independent biological replicates. Data distribution and variance homogeneity were assessed using Shapiro–Wilk and Levene's tests, respectively. Comparisons between two groups were determined using two-tailed unpaired Student's t-tests. For comparisons involving multiple groups, one-way analysis of variance (ANOVA) was employed, followed by Tukey's or Dunnett's post hoc tests. Non-parametric data were analyzed using the Mann–Whitney $U$ test or Kruskal–Wallis test with Dunn's multiple

comparisons adjustment. A *P* value < 0.05 was considered statistically significant (*$P$ < 0.05, *$P$ < 0.01, ***$P$ < 0.001, ****$P$ < 0.0001). Exact *P* values are provided in the figure legends where possible. Quantification of images was consistently performed using ImageJ software (National Institutes of Health, Bethesda, MD, USA). Unless otherwise stated, all in vitro experiments were repeated independently at least three times with similar results, and the representative images (micrographs and Western blots) shown reflect these findings. Sample sizes for in vivo experiments are specified in the respective method descriptions.

### Ethical statement
All experimental protocols involving animals were reviewed and approved by the Animal Ethics Committee of Shanxi Medical University (Approval No: SYDL2023018). All procedures were performed in strict compliance with the ARRIVE guidelines and relevant animal welfare regulations.

### Reporting summary
Further information on research design is available in the Nature Portfolio Reporting Summary linked to this article.

## Data availability
The data and statistical evaluations supporting the findings of this study are available within the article, the Supplementary Information, and the Source Data file. The mass spectrometry proteomics data have been deposited to the ProteomeXchange Consortium via the PRIDE partner repository with accession PXD064077. Source data are provided with this paper.

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

## Acknowledgements

This work was supported by the National Nature Science Foundation of China[U23A20420] (J.X.); Shanxi Province Higher Education "Billion Project" Science and Technology Guidance Project [BYBLD006] (J.X.); National Key R&D program[2021YFC2301603] (J.X.)

## Author contributions

Y.X.L., Z.Z.L., QZ.S., LP.L., and J.X. conceived the concept and designed the experiments. Y.X.L., Z.W.Y., and SB.D. synthesized the Tβ4@PLGA microspheres and prepared the composite hydrogels. YX.L., Z.W.Y., Y.Q.G., J.L., and Y.J.Y. performed the material characterization and in vitro cellular assays. Z.W.Y., S.S.J., N.J J.Y., Z.W.P., and W.J.L. S.S.J., N.J. J.Y., Z.W.P., and W.J.L. established the IUA mouse model, performed the animal surgeries, and conducted the in vivo therapeutic evaluations. Z.Y.C., H.L.Y., N.J. Z.Z.L., and Q.Z.S. assisted with the histological staining, flow cytometry, and Western blot analyses. Y.X.L. and L.P.L. analyzed the data and wrote the original draft of the manuscript. Y.X.L. and J.X. supervised the study, provided resources, and reviewed and edited the manuscript. All authors discussed the results and approved the final version of the manuscript.

## Competing interests

Yuxiang Liang, Jun Xie, Mengyuan Li, Zhiwei Peng, and Zhizhen Liu have submitted a patent application to the China National Intellectual Property Administration (CNIPA) pertaining to the preparation and application of the composite hydrogel system described in this work (application number 202511595392.5). The remaining authors declare no competing interests.
