## [Transparent Peer Review file · Nature Communications]

Decidualization-Empowered ECM Hydrogel integrating sustained T β 4 release Drives Endometrial Regeneration in Intrauterine Adhesions

Corresponding Author: Dr yuxiang liang

Version 0:

Reviewer comments:

Reviewer #1

(Remarks to the Author)

This study introduced a newly developed decellularized ECM derived from decidualized uteri and were the first to demonstrate its efficacy in the treatment of IUA. DEndo-UdECM was shown to promote endometrial regeneration and angiogenesis both in vivo and in vitro. Building on these findings, the authors proposed a novel synergistic therapeutic strategy combining DEndo-UdECM with T β 4 peptides, which are known to enhance tissue repair, stimulate angiogenesis, and critically, inhibit inflammatory and fibrotic pathways, including the TGF- β 1/Smad3 signalling axis. In a mouse model of IUA, T β 4 was administered in combination with DEndo-UdECM as a mixed formulation, functioning as a dual-action system. This approach created a pro-regenerative microenvironment by modulating immune responses, specifically by promoting M2 macrophage polarization, while concurrently suppressing detrimental processes such as pyroptosis and canonical TGF- β /Smad3 signalling.

Major comment

1. The rationale for selecting decellularized ECM derived from decidualized endometrium (DEndo-UdECM) remains unclear to this reviewer. The authors stated that “prior studies have explored the therapeutic potential of decidual stromal cells or their derivatives (e.g., exosomes) for IUA, underscoring the value of harnessing decidualization-associated factors” (lines 116–118), and further hypothesise that DEndo-UdECM may retain key bioactive components of the decidual niche (lines 120–122). However, it is well established that non-structural derivatives such as soluble cytokines and exosomes are typically removed during the decellularization process. This raises concerns that DEndo-UdECM may not confer the same regenerative efficacy as reported for decidual cell-derived soluble factors in previous studies. The authors are encouraged to clarify the justification for using decidualized ECM under these considerations and to discuss potential limitations regarding the retention of bioactive molecules following decellularization.

2. The authors used uterus tissue subjected to sesame oil-induced decidualization as the source for DEndo-UdECM. Is there a specific reason why the authors chose to use a decidualization-induced model instead of utilizing naturally decidualized uterus from mated animals? While this model is commonly used to induce decidualization via artificial mechanical stimulation, it does not fully recapitulate the physiological context of natural decidualization process. I am not fully convinced that the ECM derived from artificially decidualized tissue can adequately mimic a true decidual niche. The authors should justify the choice of this model over physiologically decidualized uteri and discuss how the mode of decidualization might impact the ECM composition and its regenerative efficacy. In order to stick to this model system, the authors should provide a screening data that demonstrate the similarity between sesame oil-induced and naturally-induced decidualization.

3. In Results 2–4, the authors present that the use of DEndo-UdECM enhances wound healing in endometrial stromal cells and promotes the proliferation of Ishikawa cells (endometrial epithelial cells). However, these findings appear to contradict the well-established concept of the decidualization niche. Decidualization of the human endometrium is commonly defined as a process involving the dramatic morphological and functional differentiation of endometrial stromal cells (ESCs), which is essential for the establishment of a successful pregnancy. This process is typically characterized by cellular differentiation rather than proliferation, migration, or regenerative behaviour. These results therefore raise questions as to whether the DEndo-UdECM truly mimics the characteristics of a decidualized niche or not, as claimed by the authors.

4. Immunofluorescent images presented both in the main and supplementary figures are not clear at all. The zoomed-in panels appear blurry, as if they have been artificially smoothed or are out of focus. Please clarify whether this issue is due to image resolution degradation or suboptimal imaging during acquisition. All images should be revised and if magnified images are provided, it would be very much helpful.

Minor comments

1. In Figure 8B, the representative image for the model group does not appear to match the corresponding data points shown in the graph (Figure 8D). According to the experimental design, both uterine horns should have undergone modelling. Could the authors clarify whether each horn received a different treatment, or if there is an inconsistency between the image and the quantitative data?

2. In Figure 2H-I: The images show fibronectin expression, whereas the main text refers to laminin. This inconsistency should be corrected.

3. The manuscript currently includes 12 main figures, which makes the presentation somewhat fragmented and potentially overwhelming for the reader. I recommend consolidating the figures where possible to improve clarity and narrative flow. A more compact and focused presentation would enhance the overall readability and impact of the manuscript.

4. In Figure 7A and 7C, Masson's trichrome staining results appear visually similar across the different groups, with limited red staining observed. It is unclear whether the staining was optimally performed. Additionally, the authors should clarify whether collagen quantification was based on the entire cross-sectional area of the uterus or selected regions of interest. A more detailed explanation of the quantification method and representative staining quality would be helpful to validate the conclusions drawn from this data.

Reviewer #2

(Remarks to the Author)

The manuscript presents a conceptually innovative and well-structured study that introduces a “decidualization-empowered” strategy for intrauterine adhesion (IUA) therapy. By integrating extracellular matrix derived from decidualized endometrium (DEndo-UdECM) with sustained release of thymosin $\beta 4$ (T $\beta 4$) via PLGA microspheres, the authors demonstrate enhanced endometrial regeneration, suppression of fibrosis, and improved fertility restoration in a murine model. While the biological outcomes are thoroughly validated, several critical concerns, particularly related to materials characterization, experimental design transparency, and figure presentation, must be addressed to ensure reproducibility and mechanistic clarity. Therefore, the manuscript requires major revision.

1. Although the hydrogel system is described as injectable and biodegradable, the manuscript lacks comprehensive characterization of key material parameters that are essential for reproducibility and mechanistic interpretation. For instance, (a) There is no direct evidence that T $\beta 4$ is successfully loaded onto or encapsulated within the PLGA microspheres. SEM images of PLGA and T $\beta 4$ @PLGA microspheres look indistinguishable.

(b) The actual T $\beta 4$ loading capacity per unit weight of PLGA microsphere is not reported. Was encapsulation efficiency confirmed via ELISA or other quantification methods?

(c) There is no information on microsphere porosity: Are the particles solid or porous? If porous, what is the average pore size, and does it fall within micro-, meso-, or macroporous range?

(d) The distribution of microspheres within the hydrogel is not evaluated. Is it uniform? Was any imaging used to confirm homogeneous dispersion?

(e) The mechanical properties of the hydrogel (e.g., compressive modulus, tensile strength) are not reported. Given that the hydrogel is injected into the uterine cavity, its mechanical support capacity is critical for functional application.

(f) The swelling behavior of the hydrogel is not reported. Given the secretory and fluid-rich environment of the uterine cavity, it is essential to assess the anti-swelling capacity of the hydrogel.

2. The manuscript claim that “time-series SEM visualized the gradual degradation of the embedded microspheres over time” lacks methodological detail and raises interpretive concerns. SEM is inherently a dry-state imaging technique and cannot capture dynamic degradation in situ. The manuscript does not clarify whether the microspheres were extracted from the hydrogel, whether samples were freeze-dried and fractured, or how the time series was established. As such, it remains unclear whether the observed morphological changes truly reflect degradation kinetics within the hydrogel or simply post-processing artifacts.

3. Figure 1, the schematic appears to depict a surgically opened uterine cavity, which is inconsistent with the experimental protocol. Please revise accordingly. The timeline in the bottom row proceeds right-to-left; for clarity and convention, consider reordering it to left-to-right, ending with “healthy growth.”

4. Figures 2, 5–7, 9–12, and S1, DAPI staining across multiple figures is too faint to discern nuclear localization. Please adjust imaging brightness/contrast or use enhanced labeling to improve clarity.

5. Figure 4E, the meaning of “Con” and “Dec” is not defined.

6. Figure 5, the meaning of “Geltin” is unclear and lacks a corresponding legend. Please clarify. Is “Mre” meant to indicate “merge”? Ensure consistent and explicit labeling. EdU (A, C) and AM-PI staining (B, D) appear to use endometrial cancer

cells. Given the focus on regeneration rather than malignancy, this may be inappropriate—please justify or replace. The figure layout does not match the order of data presentation. Reorganize panels accordingly (e.g., C ↔ B, F ↔ H, J ↔ G) to enhance logical flow. Include quantitative analysis of angiogenesis, such as average capillary length or number of branch points in tube formation assays.

7. Figure 6, a schematic illustration showing how endometrial thickness was measured (e.g., anatomical landmarks, measurement direction, number of sections) would improve reproducibility.

8. Line 609, the statement that “functional restoration of glands was evidenced by the near-complete recovery of the key transcription factor FOXA2” should be elaborated. Please explain the physiological relevance of FOXA2 and how it reflects glandular function.

9. Figure 7, Panel order needs reorganization for narrative consistency. The blue channel indicating DNA (DAPI) staining is not labeled. A large discrepancy is observed between fibrotic marker expression in IHC and immunofluorescence—please explain potential methodological or interpretive reasons.

10. Figure 8, only the postnatal development of pups from the DEndo-UdECM/Tβ4@PLGA group is shown. Were offspring from other treatment groups monitored? Comparative data should be provided to validate the functional benefit.

11. Figure 9, consider supplementing M2 macrophage polarization data with flow cytometry to enhance rigor and quantification.

12. Figure 10, the single-channel image for CD34 lacks visible fluorescence signal. Verify if signal acquisition or channel intensity was appropriate. If necessary, adjust exposure or contrast.

13. Figure 11, a schematic depiction of the TGFβ1/Smad signaling pathway and its activation by the treatment would help contextualize the molecular findings.

14. Figure 12, clarify the rationale for choosing IHC versus immunofluorescence for each marker. Is it based on antibody availability, signal specificity, or detection sensitivity?

15. Figure S3, the toxicity assay employed is insufficient to conclude biocompatibility. Standard in vivo biocompatibility evaluation should be conducted—e.g., subcutaneous implantation followed by histology and IHC at defined time points.

Reviewer #3

(Remarks to the Author)

This manuscript presents the development of a decellularized extracellular matrix hydrogel incorporating Tβ4-loaded PLGA microspheres. Notably, the study introduces an innovative approach by utilizing decidualized mouse uterus for the preparation of dECM hydrogel, demonstrating favorable regenerative efficacy. However, the scientific evidence remains insufficiently robust and the experimental design requires refinement. Given the current scope and depth of the work, I conclude that this study does not meet the standards for publication in Nature Communications and therefore recommend rejection. Detailed comments are provided below.

Major remarks:

1. Section 3.1 mentions that the dECM hydrogel exhibits thermosensitive gelation properties. Experimental validation of this property, including quantification of the sol-gel transition time, should be provided.
2. The manuscript only presents the release curve of Tβ4 from microspheres. Please also show the release behavior of Tβ4 from the hydrogel.
3. The in vitro and in vivo degradation properties of the composite hydrogel should be investigated.
4. Figure 5, the authors only evaluated in vitro regeneration effects of CEndo-UdECM and DEndo-UdECM. Since the introduction highlights Tβ4's role in enhancing tissue repair and promoting angiogenesis, DEndo-UdECM/Tβ4@PLGA group should be added
5. A large number of immunofluorescence and histological experiments were used in in vitro and in vivo experiments to observe the phenotype. Please add Western blotting experiments for in vitro experiments to increase credibility
6. The in vitro anti-fibrosis experiments are insufficient. Additional studies are needed to evaluate the overall effect of the hydrogel.
7. All immunofluorescence images suffer from poor clarity and overexposure. Please improve the image quality. Non-specific fluorescence signals are evident in multiple immunofluorescence images, particularly in Figures 9A, 10A, and 11H.
8. As the manuscript described, the established animal model is in fact an endometrium injured-model, other than an IUA model which should present thin endometrium and severe fibrosis.

Minor remarks:

1. Caption of Figure 2, “Characterization of UdECM scaffolds before and after decidualization” , herein “decidualization” should be “decellularization”.
2. Figure 11A, please add the number of samples in each group on the western blot bands.
3. Figure 11A, please add western blot bands of Total Smad3, and Fibrosis related proteins (e.g. FN, COL1A1...)
4. In figure. 5B, C, D, the PI/AM staining results of the gelatin hydrogel were not displayed.

Reviewer #4

(Remarks to the Author)

Version 1:

Reviewer comments:

Reviewer #1

(Remarks to the Author)

Overall, the revised manuscript presents a coherent and mechanistically substantiated narrative that advances the study from a phenomenological report to a conceptually unified demonstration of a “decidualization-empowered” bioactive matrix. The work now offers clear scientific novelty, methodological rigor, and tangible translational relevance in uterine regenerative medicine. Minor refinements—such as clarifying methodological details related to ECM batch variability—could further enhance reproducibility, but no substantive deficiencies remain.

Reviewer #2

(Remarks to the Author)

The authors have revised some figures and text in response to the previous review and have addressed a few of the questions raised. However, many critical issues remain unresolved, and several responses contain inaccuracies or appear to downplay key concerns. Therefore, I recommend major revision or rejection of the manuscript.

1. Fig. 4i: The image does not clearly demonstrate a homogeneous distribution of T β 4@PLGA microspheres; local aggregation appears evident. Fluorescence labeling combined with confocal microscopy is recommended to more accurately verify the spatial distribution of the microspheres within the hydrogel.
2. The authors claim that the hydrogel is self-healing, yet no experimental evidence is provided to support this statement.
3. Fig. S6c: The x-axis label should indicate days rather than hours.
4. Time-series SEM images: If these images represent the degradation behavior of standalone microspheres rather than those embedded within the hydrogel, the data should be relocated to the earlier section describing microsphere characterization. In addition, the methodological description in lines 236–255 remains insufficient, which hinders proper evaluation of this experiment. Please provide detailed procedural information.
5. Regarding the use of endometrial cancer cells (Ishikawa cells) in the experiments from the original Fig. 5 to validate the material's ability to promote normal tissue regeneration, the authors provided no response. This is a critical issue. Furthermore, in this section of their response, the four references cited under "Regarding the Rationale for Using HUVECs" do not actually contain content related to tube formation assays.
6. We recommended incorporating quantitative statistical analysis of the tube formation assay results. However, the authors failed to implement this. Instead, they cited four references unrelated to tube formation assays to assert that such assays are a gold standard. For reference, quantifiable parameters for tube formation assays include, but are not limited to, the following:
 - Tube length: Measurement of the total length of tubular structures, reflecting endothelial cell migration and assembly capability.
 - Coverage area: Calculation of the area occupied by the tubular network, indicating the extent of angiogenesis.
 - Number of loops/meshes: Enumeration of loop or mesh structures within the tubular network, reflecting vascular network complexity.
 - Number of nodes/branch points: Counting of branching points in the tubular structures, indicating the richness of vascular branching.
7. The authors partially addressed our concerns regarding endometrial thickness measurements. According to their response, they measured endometrial thickness using the myometrium as the boundary. However, would it be more appropriate to perform these measurements on Masson-stained sections?
8. In the original Fig. 8, the "Postnatal development of pups in the DEndo-UdECM/T β 4@PLGA combination treatment group" was presented. What about the offspring in other treatment groups? Were there any differences? This question was not directly answered. The authors stated, "We have added a new column for 'Postnatal Survival Rate (%)' to our new Supplementary Table (Table S1), where every group with live births is listed with a value of '100%'." However, the current Table S1 contains antibody-related data, not the relevant survival data. Thus, our query remains unaddressed. Additionally, we still recommend including offspring data in the figure, similar to Fig. 7c, to enhance data reliability.
9. We suggested that the authors use flow cytometry to detect M2-polarized macrophages in the original Fig. 9, as this would improve data credibility. While the authors provided additional immunohistochemical images for other markers, they did not directly address the use of flow cytometry. We maintain that employing multiple methodologies would enhance data reliability and still recommend using flow cytometry, a relatively more objective method, for statistical analysis.
10. Please provide IHC analysis of tissue sections after subcutaneous implantation to evaluate whether the material induces an inflammatory response. This is crucial for assessing biocompatibility, for example, by examining inflammatory markers such as IL-1.

Reviewer #3

(Remarks to the Author)

The revised manuscript seems to be improved much after addressing the reviewers' comments and is ready to be considered for acceptance.

Reviewer #4

(Remarks to the Author)

Version 2:

Reviewer comments:

Reviewer #2

(Remarks to the Author)

The revised manuscript seems to be improved much after addressing the reviewers' comments and is ready to be considered for acceptance.

Reviewer #4

(Remarks to the Author)

Response to Reviewers for Manuscript ID NCOMMS-25-36427A

Title: **Decidualization-Empowered ECM Hydrogel integrating sustained T β 4 release Drives Endometrial Regeneration in Intrauterine Adhesions**

Authors: Yuxiang Liang^{#12}, Zhaowei Yu^{#1}, Shaobo Du¹, Yuqian Guo¹, Jing Li¹, Yujia Yan¹, Shanshan Jin¹, Wenjing Liang¹, Mengyuan Li¹, Jiao Yang¹, Zhiwei Peng¹, Zhaoyang Chen², Zhizhen Liu^{*1}, Qizhi Shuai^{*1}, Liping Li^{*1}, Jun Xie^{*1}

*In this response letter, all revisions made to the manuscript text are highlighted in **blue** for clarity.*

Point-by-Point Responses to Reviewer #1:

Reviewer #1 (Remarks to the Author):

This study introduced a newly developed decellularized ECM derived from decidualized uteri and were the first to demonstrate its efficacy in the treatment of IUA. DEndo-UdECM was shown to promote endometrial regeneration and angiogenesis both in vivo and in vitro. Building on these findings, the authors proposed a novel synergistic therapeutic strategy combining DEndo-UdECM with T β 4 peptides, which are known to enhance tissue repair, stimulate angiogenesis, and critically, inhibit inflammatory and fibrotic pathways, including the TGF- β 1/Smad3 signalling axis. In a mouse model of IUA, T β 4 was administered in combination with DEndo-UdECM as a mixed formulation, functioning as a dual-action system. This approach created a pro-regenerative microenvironment by modulating immune responses, specifically by promoting M2 macrophage polarization, while concurrently suppressing detrimental processes such as pyroptosis and canonical TGF- β /Smad3 signalling.

Dear Reviewer #1,

We sincerely thank you for your time, your thorough review, and for providing such an **insightful and accurate summary of our work's key contributions**. Your recognition of the novelty in our "decidualization-empowered" ECM and the dual-action mechanism of our synergistic therapeutic strategy is greatly appreciated.

Your constructive comments have been invaluable in helping us to significantly improve the clarity, depth, and overall impact of our manuscript. We have carefully considered all of your suggestions and have addressed them point-by-point below.

Reviewer #1, Major Comment #1:

The rationale for selecting decellularized ECM derived from decidualized endometrium (DEndo-UdECM) remains unclear to this reviewer. The authors stated that “prior studies have explored the therapeutic potential of decidual stromal cells or their derivatives (e.g., exosomes) for IUA, underscoring the value of harnessing decidualization-associated factors” (lines 116–118), and further hypothesize that DEndo-UdECM may retain key bioactive components of the decidual niche (lines 120–122). However, it is well established that non-structural derivatives such as soluble cytokines and exosomes are typically removed during the decellularization process. This raise concerns that DEndo-UdECM may not confer the same regenerative efficacy as reported for decidual cell-derived soluble factors in previous studies. The authors are encouraged to clarify the justification for using decidualized ECM under these considerations and to discuss potential limitations regarding the retention of bioactive molecules following decellularization.

Response:

We sincerely thank the reviewer for this insightful and fundamental question. It touches upon the very core of our hypothesis and provides us with a welcome opportunity to elaborate on the scientific rationale behind our "decidualization-empowered" strategy. We agree that this justification is critical and needs to be articulated with greater clarity in the manuscript.

Our response is structured around four key points:

1. Full Acknowledgment of Limitations Regarding Soluble Factors:

We completely agree with the reviewer's premise. The decellularization process is, by design, intended to remove cellular components, which inevitably leads to a significant loss of soluble, non-structural molecules like cytokines, growth factors, and exosomes. We acknowledge that these soluble factors are indeed potent mediators of regeneration,

as demonstrated in numerous elegant studies. This is an important limitation of our approach, and we have now explicitly addressed it in a new "Limitations" subsection in the Discussion.

2. The Core Rationale: Harnessing the Unique *Insoluble* Decidual Matrix:

Our central hypothesis is not based on the retention of soluble factors, but rather on capturing the **unique, complex, and functionally instructive *insoluble* extracellular matrix (ECM) microenvironment** that is actively constructed during decidualization. This process involves a profound transformation of the endometrium, resulting in an ECM that is structurally and biochemically distinct from its non-decidualized counterpart or from the fibrotic scar tissue found in IUA. We posit that this preserved, tissue-specific matrix provides essential physical and biochemical cues to guide endogenous cells towards a regenerative, rather than a fibrotic, fate.

3. Strong Evidence from the Literature Supporting a Functionally Active Decidual ECM:

The functional importance of this actively remodeled, insoluble decidual ECM is well-supported by existing literature:

* As summarized in the landmark review by **Gellersen and Brosens (Endocr Rev, 2014)**, decidualization leads to the formation of a rich pericellular ECM with significant increases in key insoluble components like **collagen IV and laminin**, which are integral to establishing the functional decidual niche.

* The review by **Shi et al. (Int J Biol Sci, 2020)** details the diverse repertoire of collagens actively produced by decidual cells, highlighting a significant upregulation of **collagen type IV**, which plays active roles in immune modulation and angiogenesis through interactions with cell-surface receptors.

* Most compellingly, the study by **Yang et al. (Int J Mol Sci, 2022)** provides direct functional evidence. They show that **Laminin A5**, an insoluble ECM protein, is strongly upregulated during decidualization and that its knockdown *impairs* the process. This demonstrates that specific insoluble ECM components are not merely passive scaffolds but are functionally essential for establishing the decidual phenotype.

4. Our Own Proteomic Data Corroborates This Hypothesis:

To directly test our hypothesis, we performed a preliminary proteomic analysis

comparing our DEndo-UdECM to a control UdECM derived from non-decidualized tissue. The results were striking: our analysis confirmed the retention and significant enrichment of numerous ECM proteins in the DEndo-UdECM, with **99 ECM proteins found to be upregulated**. This provides direct evidence from our own system that the decellularization of decidualized tissue successfully captures a unique and potentially superior insoluble protein profile.

In summary, our rationale is to leverage the **functionally instructive, insoluble matrix scaffold** that is the hallmark of physiological, scarless endometrial remodeling. We believe this provides a more robust and persistent set of regenerative cues compared to the transient effects of soluble factors alone.

Reference:

Gellersen B, Brosens JJ. Cyclic decidualization of the human endometrium in reproductive health and failure. *Endocr Rev.* 2014 Dec;35(6):851-905.

Shi JW, Lai ZZ, Yang HL, Yang SL, Wang CJ, Ao D, Ruan LY, Shen HH, Zhou WJ, Mei J, Fu Q, Li MQ. Collagen at the maternal-fetal interface in human pregnancy. *Int J Biol Sci.* 2020 May 25;16(12):2220-2234.

Yang ZS, Pan HY, Shi WW, Chen ST, Wang Y, Li MY, Zhang HY, Yang C, Liu AX, Yang ZM. Regulation and Function of Laminin A5 during Mouse and Human Decidualization. *Int J Mol Sci.* 2021 Dec 24;23(1):199.

Action Taken:

Based on the reviewer's excellent feedback and our refined manuscript structure, we have performed the following major revisions:

- **Streamlined the Introduction:** We have significantly refined the Introduction to crisply present our core hypothesis regarding the "decidualization-empowered" scaffold, without delving into excessive mechanistic detail. This makes the introduction more focused and impactful.
- **Strengthened Discussion Section 3.1:** We have relocated the in-depth discussion of the decidual ECM's biological basis, supported by key literature, to **Discussion Section 3.1**. This strategic move allows us to more effectively integrate this established rationale with our own proteomic findings, creating a much stronger and more coherent scientific argument.
- **Added a "Limitations" Section:** As suggested, we have added a new subsection to the Discussion to transparently acknowledge the limitation regarding the loss of soluble factors.

- **Updated Bibliography:** All relevant references have been integrated and updated.
-

Revisions in the Manuscript (Introduction):

The revised introduction now crisply presents the core hypothesis as follows:

Nature provides a compelling blueprint for scarless endometrial repair in physiological decidualization, a process that establishes a unique pro-regenerative and immunomodulatory milieu[12, 26-28]. While prior studies inspired by this phenomenon have explored the therapeutic potential of decidual cells or their soluble derivatives[31-33], translating cell-based therapies faces clinical hurdles, and these approaches may not fully leverage the persistent cues embedded within the insoluble matrix framework[34, 35]. We therefore hypothesized that a decellularized ECM derived directly from *in vitro* decidualized endometrium (DEndo-UdECM) could capture this unique, pro-regenerative matrix. This would offer a superior, cell-free, “decidualization-empowered” biomaterial that leverages the persistent physical and biochemical cues of the insoluble scaffold to drive functional repair.

Revisions in the Manuscript (Discussion):

3.1 A decidualization-empowered ECM hydrogel engineered for spatiotemporal control of endometrial repair

“Our proteomic findings are strongly supported by the established biology of the decidual niche. Decidualization involves profound remodeling of the ECM, creating a microenvironment distinct from both quiescent and fibrotic tissue. Foundational reviews highlight that decidual stromal cells become surrounded by a rich pericellular ECM, with significant increases in components like collagen IV and laminin, which are integral to establishing this functional niche[29]. The unique repertoire of collagens present at the maternal-fetal interface is not merely structural but plays active roles in immune modulation and angiogenesis[30]. This aligns with our observation of enriched pathways governing these processes. Compellingly, specific ECM components are not just present but are functionally indispensable; for instance, Laminin A5 is strongly upregulated under progesterone control, and its knockdown is known to impair the decidualization process itself, underscoring the inherent bio-instructive nature of these proteins[27]. By capturing this complex, insoluble matrix, our approach harnesses a

pre-programmed, bioactive landscape poised for regeneration, demonstrating that DEndo-UdECM is not merely a passive scaffold but a bioactive repository of instructive cues.”

3.5 Limitations

“A limitation of utilizing decellularized matrix is the inevitable loss of soluble bioactive molecules, such as growth factors and cytokines, which are secreted by decidual cells and contribute significantly to their regenerative and immunomodulatory properties as shown in previous studies. While our focus was on the retained insoluble matrix, future research could explore strategies to combine DEndo-UdECM with specific beneficial soluble factors.”

Reviewer #1, Major Comment #2:

The authors used uterus tissue subjected to sesame oil-induced decidualization as the source for DEndo-UdECM. Is there a specific reason why the authors chose to use a decidualization-induced model instead of utilizing naturally decidualized uterus from mated animals? While this model is commonly used to induce decidualization via artificial mechanical stimulation, it does not fully recapitulate the physiological context of natural decidualization process. I am not fully convinced that the ECM derived from artificially decidualized tissue can adequately mimic a true decidual niche. The authors should justify the choice of this model over physiologically decidualized uteri and discuss how the mode of decidualization might impact the ECM composition and its regenerative efficacy. In order to stick to this model system, the authors should provide a screening data that demonstrate the similarity between sesame oil-induced and naturally-induced decidualization.

Our Response:

We sincerely thank the reviewer for this insightful and critical question regarding our choice of the sesame oil-induced artificial decidualization (AD) model. We agree that natural pregnancy decidualization (NPD) represents the physiological “gold standard,” and your concern that an artificial model may not fully recapitulate the natural process is entirely valid and deserves a thorough justification.

Our choice of the AD model was a deliberate and well-considered decision based on two primary factors: (1) **The experimental necessity to isolate a pure, uterine-derived ECM**, and (2) **Strong evidence from existing literature, now further corroborated by our own new bioinformatic analysis, demonstrating the high fidelity of the AD model.**

1. Rationale for an Embryo-Free Model:

- **Isolation of the Intrinsic Uterine Response:** Our primary goal was to investigate the regenerative potential of the ECM derived *specifically* from the decidualized uterine stroma, independent of any confounding contributions from the embryo. In a natural pregnancy, the decidua is in constant crosstalk with the conceptus. The AD model, by using a sterile stimulus, allows us to obtain a “pure” sample of uterine decidual ECM.
- **Practical Advantages (Yield and Consistency):** The AD model induces a uniform and widespread decidualization of the entire uterine horn, providing a significantly greater yield of consistent tissue for ECM extraction compared to the small, discrete implantation sites in an NPD model. This is a critical practical advantage for ensuring the reproducibility and scalability of our ECM preparation.

2. High-Fidelity Validation: Existing Literature and New Supporting Data:

While the AD model lacks embryonic signals, multiple studies have established that it faithfully recapitulates the key molecular events of natural decidualization (Herington et al., *Endocrinology*, 2009). More directly, a comprehensive transcriptomic analysis by Wang et al. (*Genes*, 2020) found a massive overlap in gene expression profiles between the AD and NPD models, concluding that **“AD is a reliable model for mouse decidualization.”**

To directly address your request for “screening data,” and to bolster this point even further, we have performed a **new, targeted re-analysis of the publicly available RNA-seq data from Wang et al. (2020)**. We specifically focused on the **“Matrisome”**—the complete set of genes encoding the ECM and its associated proteins, which is the most relevant gene set for our study.

Our new analysis reveals:

- The relative proportions of different Matrisome subcategories (e.g., collagens, glycoproteins, proteoglycans) that are significantly regulated during decidualization are **strikingly similar** between the AD and NPD models (see new **Figure S5a**).
- Crucially, the heatmap analysis shows that the specific ECM-related genes that are upregulated (**Figure S5b**, shown below) or downregulated (**Figure S5c**) are **highly consistent** between the AD and NPD models.

This new analysis provides direct, quantitative evidence that the changes in the ECM composition during oil-induced decidualization are a high-fidelity proxy for those that occur during natural pregnancy.

Reference:

Wang C, Zhao M, Zhang WQ, Huang MY, Zhu C, He JP, Liu JL. Comparative Analysis of Mouse Decidualization Models at the Molecular Level. *Genes (Basel)*. 2020 Aug 13;11(8):935.

Herington JL, Underwood T, McConaha M, Bany BM. Paracrine signals from the mouse conceptus are not required for the normal progression of decidualization. *Endocrinology*. 2009 Sep;150(9):4404-13.

New Figure S5 | Bioinformatic comparison reveals a highly similar ECM matrisome between artificial and natural decidualization models. Re-analysis of public transcriptomic data (GEO: GSE122376) comparing the artificial decidualization (AD) and natural pregnancy decidualization (NPD) models.

(a) Percentage distribution of differentially expressed matrisome genes across major ECM categories shows a highly similar profile between AD (blue) and NPD (red) models. (b, c) Heatmaps of upregulated (b, blue) and downregulated (c, orange) matrisome genes reveal

strongly concordant expression patterns between the AD and NPD models. The results validate that the AD model effectively recapitulates the ECM remodeling events of natural decidualization.

In summary, our choice of the AD model is a justified strategy that allows for the reproducible collection of pure uterine-derived ECM. The validity of this model is supported not only by previous landmark studies but also by our new, targeted bioinformatic analysis focusing specifically on the ECM, which directly addresses the core of your concern.

Action Taken:

To ensure this critical justification is transparent to all readers, we have substantially revised the **Discussion** section to incorporate this rationale and reference our new supporting data.

Revisions in the Manuscript (The revised text in the Discussion now reads as follows):

“The cornerstone of our approach lies in the high-fidelity replication of the decidual microenvironment within our biomaterial. ... A critical aspect of our study design was the use of this sesame oil-induced artificial decidualization (AD) model rather than a natural pregnancy decidualization (NPD) model. This was a deliberate choice driven by experimental necessity and strong supporting evidence. The AD model allows for the isolation of the intrinsic uterine decidual response, free from the confounding paracrine signals and cellular contributions of an embryo. This is essential for producing a consistent and pure uterine-derived ECM with high yield, a significant practical advantage over the small, discrete implantation sites in NPD. The validity of this approach is robustly supported by comparative molecular studies. A comprehensive RNA-seq analysis by Wang et al. directly compared AD and NPD models, finding an overwhelming transcriptomic overlap (1,977 consistently expressed genes) and concluding that “AD is a reliable model for mouse decidualization”[12]. To further substantiate this, our targeted re-analysis of this transcriptomic data confirmed a high degree of similarity in the expression profiles of ECM-related genes (the Matrisome) between the AD and NPD models (Fig. S5), validating our model choice

at the molecular level. This aligns with earlier work showing that the uterus can mount a proper decidual response without requiring signals from the conceptus[13]. Therefore, we are confident that the ECM derived from the AD model is a highly relevant proxy for the natural decidual niche, ideal for evaluating its intrinsic, maternally-derived regenerative properties. ...”

We are confident that this comprehensive response, supported by new, targeted bioinformatic analysis, fully addresses your valid concerns and strengthens the foundation of our study. We thank you again for pushing us to clarify this important point.

Reviewer #1, Major Comment #3:

In Results 2–4, the authors present that the use of DEndo-UdECM enhances wound healing in endometrial stromal cells and promotes the proliferation of Ishikawa cells (endometrial epithelial cells). However, these findings appear to contradict the well-established concept of the decidualization niche. Decidualization of the human endometrium is commonly defined as a process involving the dramatic morphological and functional differentiation of endometrial stromal cells (ESCs), which is essential for the establishment of a successful pregnancy. This process is typically characterized by cellular differentiation rather than proliferation, migration, or regenerative behaviour. These results therefore raise questions as to whether the DEndo-UdECM truly mimics the characteristics of a decidualized niche or not, as claimed by the authors.

Our Response:

We are sincerely grateful to the reviewer for this highly insightful and thought-provoking question. You have correctly identified an apparent paradox: our findings show that DEndo-UdECM promotes proliferation and migration, while the classical view of decidualization focuses on terminal differentiation. We agree this is a critical point and appreciate this opportunity to clarify our conceptual framework, which is strongly supported by recent, authoritative reviews in implantation biology.

Our central argument is that viewing decidualization solely as a static, terminally differentiated state is an oversimplification. Instead, we posit that **decidualization is a dynamic, multi-phase process fundamentally analogous to a controlled, scar-free wound healing and tissue remodeling event**. Our DEndo-UdECM is designed to mimic the crucial **initial and supportive phases** of this process, which are intensely pro-regenerative.

1. The Initial Phase of Decidualization is Inherently Proliferative and Inflammatory:

The literature clearly shows that stromal cell proliferation is not contradictory to decidualization, but is an indispensable prerequisite for building the decidual tissue.

- **Cha et al. (*Nature Medicine*, 2012)** explicitly state that “Stromal cells surrounding the implanting blastocyst undergo **extensive proliferation and differentiation**.”
- Furthermore, they describe implantation as a physiological “proinflammatory reaction,” which is essential for initiating tissue remodeling.
- **Gellersen & Brosens (*Endocrine Reviews*, 2014)** reinforce this by describing decidualization as a “biphasic process, characterized initially by an acute-phase inflammatory response,” which is essential for establishing receptivity.

Therefore, a biomaterial that successfully mimics the initial decidual niche *should be* pro-proliferative and promote a controlled remodeling response, precisely as we observed.

Reference:

Cha J, Sun X, Dey SK. Mechanisms of implantation: strategies for successful pregnancy. *Nat Med*. 2012 Dec;18(12):1754-67.

Gellersen B, Brosens JJ. Cyclic decidualization of the human endometrium in reproductive health and failure. *Endocr Rev*. 2014 Dec;35(6):851-905.

2. Therapeutic Context is Key: The Goal is Regenerative Healing, Not Terminal Differentiation:

In the pathological context of a scarred IUA uterus, the primary therapeutic need is for **regenerative wound healing**—rebuilding the cellular architecture and suppressing fibrosis. The decidual process provides the perfect biological template for this.

- As **Gellersen & Brosens (2014)** propose, cyclic decidualization and menstruation bestow “extraordinary plasticity on the endometrium, enabling it to mount a response tailored to individual embryos” and to engage in “**cyclic scar-free repair.**”
- Our therapeutic goal is to co-opt this natural, anti-scarring healing program to repair the pathological wound of IUA. The promotion of proliferation and migration by DEndo-UdECM is the necessary first step of this guided regeneration.

In summary, the pro-regenerative effects of our DEndo-UdECM do not contradict the concept of a decidual niche. On the contrary, they **confirm that our material successfully mimics the highly active and supportive initial phase of the decidualization process.** This is precisely the therapeutic action required to overcome the fibrotic state of IUA and restore a functional endometrium.

Action Taken:

To address this crucial conceptual point and enhance the manuscript's intellectual framework, we have incorporated a dedicated paragraph into the **Discussion** section. This addition explicitly clarifies our rationale: that our DEndo-UdECM is designed to mimic the highly pro-regenerative *initial phase* of decidualization, which is fundamentally analogous to a controlled wound healing process. This reframes the apparent paradox into a key strength of our therapeutic strategy and provides a stronger theoretical foundation for our findings.

Revisions in the Manuscript (Discussion 3.2):

“3.2 A Strategic 'Division of Labor': Validating the Dual-Action Mechanism In Vitro

“..., At first glance, these potent pro-regenerative effects might seem paradoxical, given the terminally differentiated nature of the mature decidua. However, this apparent conflict is resolved when decidualization is viewed not as a static end-point, but as a dynamic, multi-phase process fundamentally analogous to controlled, scar-free wound healing[48, 50]. Authoritative reviews highlight that the initial phase of decidualization

is inherently proliferative and involves extensive tissue remodeling, with stromal cells first undergoing “extensive proliferation and differentiation” to build the necessary architecture[48]. Therefore, our findings do not contradict the biomimetic premise; on the contrary, they directly confirm that DEndo-UdECM successfully recapitulates the crucial, pro-regenerative signals of this early decidualization process, making it an ideal 'engine' for therapeutic wound repair[51]. ...”

Reviewer #1, Major Comment #4:

Immunofluorescent images presented both in the main and supplementary figures are not clear at all. The zoomed-in panels appear blurry, as if they have been artificially smoothed or are out of focus. Please clarify whether this issue is due to image resolution degradation or suboptimal imaging during acquisition. All images should be revised and if magnified images are provided, it would be very much helpful.

Response:

We sincerely thank the reviewer for this critical feedback and for the constructive suggestion to include magnified images. We agree that the clarity of the images in the original submission was suboptimal and we deeply appreciate the opportunity to rectify this.

Upon a thorough investigation, we confirmed that the reviewer's suspicion was correct: the blurriness was not due to suboptimal image acquisition but was inadvertently introduced by **resolution degradation during the final figure export process**.

To address this fundamental issue, we have undertaken a comprehensive revision. We have gone back to the **original, high-resolution raw image files** for all immunofluorescent figures throughout the manuscript and its supplement. We meticulously re-assembled every panel using a **lossless TIFF format workflow** to preserve every detail of the original data.

During this process, we carefully considered the reviewer's excellent suggestion about adding magnified insets. We found that once the underlying resolution issue was resolved, the clarity of the **existing panels (including the original zoomed-in views)** **improved so dramatically** that they now clearly and crisply display the critical

subcellular details. We believe these comprehensively revised, high-resolution figures now effectively serve the purpose of high-magnification views and fully address the reviewer's core concern for clarity.

We are confident that these revised figures now meet the high standards of *Nature Communications* and accurately represent our findings in a visually compelling manner.

Action Taken:

- **Comprehensive Re-rendering of All Images:** We have re-generated **all** immunofluorescent images (in both main and supplementary figures) directly from the original, high-resolution source data. The entire figure preparation workflow was revised to exclusively use the lossless TIFF format, eliminating any compression artifacts and dramatically improving clarity.
 - **Replacement of All Figures:** All relevant figures in the revised manuscript have been replaced with these new, high-quality versions. We have ensured that the level of detail now visible in the standard panels is sufficient to support our conclusions.
-

Revisions in the Manuscript (Figures):

All immunofluorescent images presented in the main figures (e.g., Figures 2, 5–11) and supplementary figures (e.g., Figure S1, S2) have been replaced with high-resolution versions generated from the original raw data. This change is visual and does not alter the scientific conclusions drawn from the data.

Reviewer #1, Minor Comment #1:

In Figure 8B, the representative image for the model group does not appear to match the corresponding data points shown in the graph (Figure 8D). According to the experimental design, both uterine horns should have undergone modelling. Could the authors clarify whether each horn received a different treatment, or if there is an inconsistency between the image and the quantitative data?

Response:

We thank the reviewer for this insightful comment and for pointing out the need for clarification. The reviewer is correct in observing that only one uterine horn appears modeled in the representative image (Figure 8B).

This is because we employed a **unilateral (single-sided) intrauterine adhesion (IUA) model** in our study. This is a well-established and widely used method in rodent IUA research, where the un-operated contralateral horn serves as an **internal, healthy control**, thus strengthening the validity of the findings.

Consequently, all our treatments and subsequent analyses, including the quantification of implanted embryos presented in Figure 8D, were performed **exclusively on the surgically modeled uterine horn**.

To prevent any ambiguity for our readers, we have now revised the manuscript to make this experimental detail explicitly clear.

Action Taken:

- We have amended the description in the Methods section (under subsection “Animal Model and Treatment”) to explicitly state that a **unilateral model** was established in the **right uterine horn**, with the left horn serving as an internal control.
- We have updated the legend for Figure 7 (formerly Figure 8) to specify that the quantitative data represents the number of implanted embryos in the **treated (right) uterine horn**.
- We are confident that these revisions now resolve the apparent inconsistency and accurately reflect our experimental design.

Revisions in the Manuscript (Methods & Figure Legend):

Methods, Subsection “Animal Model and Treatment”: A severe murine unilateral model of intrauterine adhesions (IUA) was established in the right uterine horn via intrauterine infusion of 95% ethanol, following our previously published methodology. The contralateral left horn served as a healthy internal control.

Figure 7 Legend (formerly Figure 8): Quantitative analysis of the number of implanted embryos in the treated (right) uterine horn on day 8 of pregnancy.

Reviewer #1, Minor Comment #2:

In Figure 2H-I: The images show fibronectin expression, whereas the main text refers to laminin. This inconsistency should be corrected.

Response:

We sincerely thank the reviewer for identifying this inconsistency. This was indeed a typographical error in the main text. The immunofluorescent images in Figure 2H-I correctly show the expression of **Fibronectin (FN1)**, which is a key component of the endometrial ECM that we intended to investigate in this context.

Action taken:

- We have corrected the manuscript text to accurately refer to “**Fibronectin**” instead of “laminin” in the corresponding results section describing Figure 2.

We apologize for this oversight and appreciate the opportunity to correct it.

Reviewer #1, Minor Comment #3:

The manuscript currently includes 12 main figures, which makes the presentation somewhat fragmented and potentially overwhelming for the reader. I recommend consolidating the figures where possible to improve clarity and narrative flow. A more compact and focused presentation would enhance the overall readability and impact of the manuscript.

Response:

We sincerely thank the reviewer for this excellent and constructive suggestion. We completely agree that a more streamlined and focused presentation would significantly enhance the manuscript's narrative flow and impact.

Following this valuable advice, we have undertaken a careful reorganization of our figures. Specifically, we have consolidated the original Figure 6 (which focused on **structural endometrial repair**) and the original Figure 7 (which detailed the **anti-fibrotic effects**) into a single, cohesive **new Figure 6**.

This new, integrated figure now provides a more comprehensive and powerful visual narrative of uterine tissue regeneration. It allows the reader to simultaneously appreciate both the structural recovery (e.g., endometrial thickness, gland restoration) and the concurrent suppression of fibrosis (e.g., Masson's staining, specific marker expression). This consolidation reinforces the key message that our therapeutic strategy achieves not just architectural repair, but true functional regeneration by actively combating fibrosis.

Action Taken:

- We have merged the original Figure 6 and Figure 7 into a new, integrated Figure 6, creating a more comprehensive overview of the therapeutic outcomes.
 - The main text in the Results and Discussion sections has been revised accordingly to reflect this new figure organization, ensuring a smooth and logical narrative.
 - This consolidation has reduced the total number of main figures, leading to a more streamlined and impactful presentation, as astutely suggested by the reviewer.
-

Revisions in the Manuscript (Figure & Text):

New Figure 6 Title:

DEndo-UdECM/Tβ4@PLGA Hydrogel Promotes Comprehensive Uterine Repair by Enhancing Regeneration and Suppressing Fibrosis.

Manuscript Text:

The manuscript text has been revised throughout the Results and Discussion sections to refer to the new, consolidated Figure 6. The narrative has been adjusted to cohesively present the data on both structural repair and anti-fibrosis, reflecting the integrated nature of the new figure.

Reviewer #1, Minor Comment #4:

In Figure 7A and 7C, Masson's trichrome staining results appear visually similar across the different groups, with limited red staining observed. It is unclear whether

the staining was optimally performed. Additionally, the authors should clarify whether collagen quantification was based on the entire cross-sectional area of the uterus or selected regions of interest. A more detailed explanation of the quantification method and representative staining quality would be helpful to validate the conclusions drawn from this data.

Response:

We sincerely thank the reviewer for their meticulous observation and for holding our data to the highest standard. This is a critical point, and addressing it has substantially improved the quality and rigor of our fibrosis assessment. We agree with the reviewer's assessment of both the staining quality and the need for a more detailed methodological description. We have made significant improvements in both areas.

1. Regarding Staining Quality and Visual Representation:

Upon re-evaluating our original images, we concurred that the staining quality could be optimized. The previous images, as the reviewer correctly pointed out, exhibited a degree of over-saturation in the blue channel which partially masked the nuanced differences in collagen deposition.

To address this definitively, **we have performed a new round of Masson's trichrome staining**. The new, higher-quality images have now replaced the previous ones in the revised manuscript (now presented in **the new Figure 6h**). We are confident the reviewer will find that these new images exhibit superior color balance and contrast, clearly distinguishing the blue-stained collagen fibers from the red-stained muscle and cytoplasm. The differences between the densely fibrotic Model group and the substantially repaired treatment groups are now visually striking and unequivocal.

2. Regarding the Quantification Method and Region of Interest (ROI):

This is another excellent point that touches upon the specificity of our analysis. We apologize for not describing this with sufficient detail in the original manuscript.

Our quantification was strategically focused on the **endometrial stroma**, deliberately excluding the myometrium. This targeted ROI selection is critical for pathological relevance. Intrauterine adhesions (IUA) are a disease primarily affecting the endometrium, and it is the fibrotic scarring within this specific layer that impairs

fertility. The myometrium has a fundamentally different native collagen architecture, and including it would introduce a significant confounding variable, diluting the specific anti-fibrotic effect we aim to measure within the functionally relevant tissue layer.

Action Taken:

- **Figure Replacement with New Experimental Data:** The previous Masson's trichrome staining images in the manuscript have been **entirely replaced with new, high-quality, and more representative images** from our repeated experiments. These new images offer superior contrast and clarity, making the reduction in fibrosis visually unequivocal.
 - **Substantial Revision of the Methods Section:** We have significantly revised the Methods section to enhance clarity, detail, and reproducibility.
 - We now provide a detailed, step-by-step protocol for fibrosis quantification, which explicitly states that the **endometrial stroma was specifically selected as the Region of Interest (ROI)**.
 - We have included the **scientific rationale** for this targeted approach, emphasizing its pathological relevance to IUA.
 - The precise software (**ImageJ/Fiji with the Colour Deconvolution plugin**) and formula used to calculate the percentage of fibrotic area are now clearly described.
-

1. Revisions in the Manuscript (Methods Section):

“4.2 Histological and Morphometric Analysis.

For fibrosis assessment, Masson's trichrome staining was performed according to the manufacturer's protocol. Quantitative analysis of fibrosis was conducted using ImageJ/Fiji software (NIH, USA). For each uterine cross-section, a Region of Interest (ROI) was manually drawn to exclusively encompass the endometrial stroma, deliberately excluding the myometrium. This targeted approach is pathologically relevant as IUA primarily affects the endometrium, and excluding the myometrium prevents confounding from its distinct native collagen architecture. The Colour

Deconvolution plugin was then used to separate the blue channel (collagen) from the red channel (cytoplasm/muscle). The percentage of fibrotic area was calculated using the formula: Fibrotic Area (%) = (Blue-stained Collagen Area / Total Endometrial Stromal Area) × 100%. At least five random fields from each of six independent samples per group were analyzed...”

2. Revisions in the Figure 6 and legend:

Figure 6 | DEndo-UdECM/Tβ4@PLGA Hydrogel Promotes Comprehensive Uterine Repair by Enhancing Regeneration and Suppressing Fibrosis (h) Representative Masson’s trichrome staining of uteri from each treatment group (Sham, Model, Tβ4@PLGA, CEndo-UdECM, DEndo-UdECM, CEndo-UdECM/Tβ4@PLGA, and DEndo-UdECM/Tβ4@PLGA).

Point-by-Point Responses to Reviewer #2:

Reviewer #2 (Remarks to the Author):

The manuscript presents a conceptually innovative and well-structured study that introduces a “decidualization-empowered” strategy for intrauterine adhesion (IUA) therapy. By integrating extracellular matrix derived from decidualized endometrium (DEndo-UdECM) with sustained release of thymosin β 4 (T β 4) via PLGA microspheres, the authors demonstrate enhanced endometrial regeneration, suppression of fibrosis, and improved fertility restoration in a murine model. While the biological outcomes are thoroughly validated, several critical concerns, particularly related to materials characterization, experimental design transparency, and figure presentation, must be addressed to ensure reproducibility and mechanistic clarity. Therefore, the manuscript requires major revision.

Dear Reviewer #2

We are sincerely grateful to you for your meticulous review and for providing such insightful and constructive feedback. We are particularly encouraged by your positive assessment of our study as "**conceptually innovative**" and your recognition that our "**biological outcomes are thoroughly validated.**"

You have precisely identified the areas where the manuscript required significant improvement. We completely agree that robust **materials characterization, enhanced transparency in experimental design, and higher-quality figure presentation** are absolutely essential for ensuring the "**reproducibility and mechanistic clarity**" that a study of this nature demands.

Guided by your expert recommendations, we have undertaken a comprehensive revision. This involved not only conducting several new experiments to bolster our materials data but also meticulously rewriting relevant methods sections and systematically improving all figures to meet the highest standards of clarity. We have addressed every point you raised in detail below, and we are confident that these substantial revisions have transformed the manuscript into a much stronger, more rigorous, and more reproducible piece of work.

Reviewer #2, Major Comment #1:

Original Comment: *Although the hydrogel system is described as injectable and biodegradable, the manuscript lacks comprehensive characterization of key material parameters that are essential for reproducibility and mechanistic interpretation. For instance, (a) There is no direct evidence that Tβ4 is successfully loaded onto or encapsulated within the PLGA microspheres. SEM images of PLGA and Tβ4@PLGA microspheres look indistinguishable. (b) The actual Tβ4 loading capacity per unit weight of PLGA microsphere is not reported. Was encapsulation efficiency confirmed via ELISA or other quantification methods? (c) There is no information on microsphere porosity: Are the particles solid or porous? (d) The distribution of microspheres within the hydrogel is not evaluated. Is it uniform? (e) The mechanical properties of the hydrogel (e.g., compressive modulus, tensile strength) are not reported. (f) The swelling behavior of the hydrogel is not reported.*

Response:

We sincerely thank the reviewer for this comprehensive and exceptionally insightful critique of our material characterization. We completely agree that a robust characterization of our composite hydrogel system is paramount for reproducibility, mechanistic understanding, and clinical translation. The reviewer's detailed comments have spurred us to significantly strengthen this aspect of our work. We have performed additional experiments and provided substantial clarifications for each point raised below.

(a & b) Regarding Evidence and Quantification of Tβ4 Encapsulation

Response:

We thank the reviewer for these critical questions. We agree that the morphological similarity in SEM images between bare and Tβ4-loaded microspheres is not, by itself, proof of encapsulation. This similarity is expected due to the low mass ratio of the peptide to the polymer.

The definitive evidence for successful encapsulation is twofold:

1. **Functional Evidence:** The ability of the microspheres to provide a sustained release of their cargo. Our data in **Figure 4j** clearly demonstrates a controlled

release of T β 4 over 10 days, which would be impossible if the peptide were not successfully encapsulated.

2. **Direct Quantitative Evidence:** To address the reviewer's point directly, we have now provided a comprehensive quantitative analysis. We have corrected our terminology to use the standard terms “**Loading Capacity (LC)**” and “**Encapsulation Efficiency (EE)**”. Our ELISA-based quantification yielded a high **Encapsulation Efficiency (EE) of 76.16 \pm 1.63%** and a **Loading Capacity (LC) of 2.15 \pm 0.11%**. This robust quantitative data unequivocally confirms that a significant amount of T β 4 was successfully encapsulated within the microspheres.

Action Taken:

- We have added text to the Results section to clarify that the lack of morphological difference in SEM is expected.
- We have corrected the terminology from “Loading efficiency” to the standard terms “**Loading Capacity (LC)**” and “**Encapsulation Efficiency (EE)**” throughout the manuscript, including in the legend for the revised **Figure 4h**.
- We have added the newly calculated EE value to the Results section.
- We have included a new, detailed sub-section titled “**Quantification of T β 4 Loading Capacity and Encapsulation Efficiency**” in the Supplementary Materials and Methods, with full protocols and formulas to ensure reproducibility.

(c) Regarding Microsphere Porosity

Response:

We appreciate the reviewer’s attention to this structural detail. Our high-resolution SEM imaging reveals that the T β 4@PLGA microspheres fabricated using the double emulsion (w/o/w) method are **solid, non-porous structures** with a smooth surface. T β 4 release is therefore governed by bulk erosion and diffusion through the PLGA polymer matrix, rather than through a porous network.

Action Taken:

- We have added a sentence to the Results section to explicitly state this finding, referencing **Figure 4i**.
-

(d) Regarding Microsphere Distribution within the Hydrogel**Response:**

We thank the reviewer for this question. The distribution of microspheres within the hydrogel matrix was indeed evaluated. As shown in the cross-sectional SEM image of the composite hydrogel (**Figure 4i**), the T β 4@PLGA microspheres are distributed in a **homogeneous manner** throughout the porous network of the DEndo-UdECM hydrogel, ensuring a consistent and predictable release of T β 4 throughout the regenerative site.

Action Taken:

- We have revised the text in the Results section to more explicitly highlight the “**homogeneous dispersion of microspheres within the hydrogel matrix,**” referencing **Figure 4i**.
-

(e) Regarding the Mechanical Properties of the Hydrogel**Response:**

This is an excellent point. For an injectable, soft, self-healing hydrogel like ours, traditional compressive or tensile tests are not the most representative methods. Instead, **rheological analysis is the gold standard**. We did perform this crucial analysis, and the results were presented in the original **Figure 4c**. We regret that our initial description did not adequately convey their significance.

The data clearly shows that the storage modulus ($G' \approx 1 \text{ kPa}$) is significantly higher than the loss modulus (G''), which is the hallmark of a stable, cross-linked gel network capable of providing physical support post-injection. This modulus value is comparable to the stiffness of native soft tissues, including the endometrium, making it robust enough to resist clearance yet soft enough to be biocompatible.

Action Taken:

- We have substantially revised the Results section to more prominently feature and interpret the rheological data from **Figure 4c**, explaining why G' and G''

are the appropriate metrics and how the obtained values confirm the hydrogel's suitability for intrauterine application.

- We have refined the legend for **Figure 4c** to be more descriptive and impactful.
-

(f) Regarding the Swelling Behavior of the Hydrogel

Response:

We sincerely thank the reviewer for this insightful comment. We agree that characterizing the swelling behavior is crucial. Accordingly, we have performed a swelling assay in PBS at 37°C.

As shown in the new **Figure S6b**, our hydrogel exhibits a controlled and highly desirable swelling profile. It reaches equilibrium with a peak swelling ratio of **approximately 18%**. This limited swelling is a critical safety and functional feature, ensuring the hydrogel can conform to the uterine cavity without exerting excessive pressure on the delicate surrounding tissue.

Action Taken:

- We have conducted a hydrogel swelling assay and included the results as a new **Figure S6b**.
 - We have added a detailed protocol for the swelling assay to the Materials and Methods section.
 - We have incorporated a description and interpretation of these findings into the Results and Discussion sections.
-

Revisions in the Manuscript:

(In the main text, Results section 2.3):

“Furthermore, its behavior in a physiological environment was assessed. The hydrogel demonstrated a controlled swelling profile, reaching a modest maximum swelling ratio of approximately 18.2% and indicating excellent structural stability without the risk of uncontrolled expansion (**Fig. S6b**).”

“SEM imaging showed that both unloaded PLGA and T β 4-loaded microspheres were spherical with smooth surfaces (**Fig. 4g**).”

“Quantitative analysis determined the encapsulation efficiency (EE) to be $76.2 \pm 1.6\%$, with the loading capacity (LC) reaching a plateau at initial T β 4 concentrations above 300 ng/mL (**Fig. 4h**). When incorporated into the DEndo-UdECM hydrogel precursor, these microspheres were observed to be homogeneously dispersed throughout the resulting hydrogel matrix, as confirmed by cross-sectional SEM (**Fig. 4i**, right panel). Critically, the drug release profile was evaluated both for the microspheres alone and for the composite hydrogel system. While T β 4-loaded microspheres alone exhibited a burst release with nearly 70% released within 3 days, their incorporation into the DEndo-UdECM hydrogel resulted in a significantly more sustained and controlled release profile. This dual-component system effectively mitigated the initial burst and prolonged delivery over the 10-day period, ensuring prolonged local availability of the therapeutic peptide (**Fig. 4j**). To assess their degradation profile, the microspheres were incubated in vitro under physiological conditions and sampled at various time points. Bright-field microscopy showed a gradual loss of their sharp, spherical morphology over 10 days, while SEM imaging of parallel samples revealed the evolution from a smooth, intact surface to a more porous and eroded structure, indicating progressive degradation (**Fig. 4k**).”

(In the main text, Discussion section 3.1):

“Furthermore, its modest swelling ratio of under 20% (Fig. S6b) ensures it can fill the uterine cavity and maintain intimate tissue contact without exerting harmful pressure. This integrated design creates a sophisticated, multifunctional system that combines the potent, endogenous biomimetic cues of the decidual ECM with the spatiotemporally controlled delivery of a powerful anti-fibrotic agent, establishing a robust platform for functional endometrial repair.”

(In the Supplementary Materials and Methods 2.3 •Characterization):

A new sub-section has been added:

“Quantification of T β 4 Loading Capacity and Encapsulation Efficiency

The Loading Capacity (LC) and Encapsulation Efficiency (EE) of T β 4 in the PLGA microspheres were determined via an ELISA-based quantification method. Briefly, a precisely weighed sample of lyophilized T β 4@PLGA microspheres (W_microspheres) was dissolved in dichloromethane (DCM). The T β 4 was then extracted into an aqueous

buffer. The concentration of T β 4 (C_T β 4) in the aqueous phase was measured using a commercial ELISA kit according to the manufacturer's instructions. The total weight of actual encapsulated T β 4 (W_actual) was calculated. The LC and EE were calculated using the following formulas:

$$\text{Loading Capacity (LC) (\%)} = (W_{\text{actual}} / W_{\text{microspheres}}) \times 100\%$$

$$\text{Encapsulation Efficiency (EE) (\%)} = (W_{\text{actual}} / W_{\text{initial_T}\beta 4}) \times 100\%$$

Where W_initial_T β 4 represents the total amount of T β 4 initially added during the fabrication process. All measurements were performed in triplicate.”

(In the Supplementary Materials and Methods 2.4 •Characterization):

A new sub-section has been added:

“Hydrogel Swelling Assay: The swelling behavior of the DEndo-UdECM hydrogel was evaluated using a gravimetric method. Lyophilized hydrogel samples were weighed to obtain their initial dry weight (W_d). Subsequently, the samples were immersed in PBS (pH 7.4) and incubated at 37° C. At predetermined time points (0, 3, 6, 12, 24, 36, and 48 h), samples were retrieved, and excess surface water was gently removed with filter paper before recording the swollen weight (W_s). The swelling ratio was calculated using the following formula: Swelling Ratio (%) = [(W_s - W_d) / W_d] × 100%. All experiments were performed in triplicate.”

(New Figure Legend):

Figure S6 | Physicochemical characterization of dECM-based hydrogels. (a) Visual demonstration of the injectability of the DEndo-UdECM hydrogel precursor. (b) Swelling kinetics of the DEndo-UdECM hydrogel in PBS (37°C), showing equilibrium is reached within 36 hours. (c) Comparative in vitro degradation profiles of hydrogels derived from non-decidualized (CEndo-UdECM) and decidualized (DEndo-UdECM) uterine ECM over 14 days. Data are presented as mean ± SD (n = 3).

Figure 4 | Fabrication and characterization of DEndo-UdECM/TB4@PLGA composite hydrogels. (a) Schematic of the fabrication process. (b) Viscosity versus shear rate for the DEndo-UdECM precursor solution. (c) Time-sweep rheology showing the sol-gel transition of the precursor at 37°C. (d) Storage modulus (G') and loss modulus (G'') of the DEndo-UdECM hydrogel post-gelation. (e, f) Size distribution of blank PLGA (e) and TB4@PLGA (f) microspheres. (g) SEM images of blank PLGA and TB4@PLGA microspheres. Scale bars, 60

μm (top), $20\ \mu\text{m}$ (bottom). **(h)** T β 4 loading capacity of PLGA microspheres at varying initial T β 4 concentrations. Data are mean \pm SD ($n = 3$). **(i)** Cross-sectional SEM images of the DEndo-UdECM hydrogel (left) and the composite hydrogel with embedded T β 4@PLGA microspheres (right). Scale bar, $100\ \mu\text{m}$. **(j)** *In vitro* cumulative T β 4 release from T β 4@PLGA microspheres alone and from the composite hydrogel. Data are mean \pm SD ($n = 3$). **(k)** *In vitro* degradation of T β 4@PLGA microspheres over 10 days, imaged by Bright-field (BF) microscopy (top) and SEM (bottom). Scale bars, $100\ \mu\text{m}$.

Reviewer #2, Major Comment #2:

The manuscript claim that “time-series SEM visualized the gradual degradation of the embedded microspheres over time” lacks methodological detail and raises interpretive concerns. SEM is inherently a dry-state imaging technique and cannot capture dynamic degradation in situ. The manuscript does not clarify whether the microspheres were extracted from the hydrogel, whether samples were freeze-dried and fractured, or how the time series was established. As such, it remains unclear whether the observed morphological changes truly reflect degradation kinetics within the hydrogel or simply post-processing artifacts.

Our Response:

We sincerely thank the reviewer for this insightful comment and apologize for the lack of clarity in our original description. The reviewer is entirely correct in pointing out that SEM cannot perform live, dynamic imaging of degradation. Our initial phrasing was imprecise and potentially misleading.

The experiment was, as the reviewer correctly inferred, a separate *in vitro* degradation study where microsphere samples were incubated in PBS at 37°C . At predefined time points (Day 0, 1, 3, 5, 7, and 10), samples were collected, lyophilized, and then imaged using SEM to capture their morphological changes. This was complemented by bright-field imaging of the hydrated microspheres at the same time points. The intention was to show the structural evolution of the microspheres over time, which correlates with their drug release profile.

To rectify this significant oversight, we have undertaken a comprehensive revision of the manuscript to accurately describe this experiment and its methodology.

Action Taken:

- **Rewritten the Results Section:** We have completely rewritten the corresponding section in the Results to accurately describe this experiment as a separate *in vitro* degradation study of the microspheres, not as imaging of microspheres within the hydrogel.
- **Corrected Terminology:** We have replaced the inaccurate term “time-series SEM” with a more precise description, such as “SEM imaging revealed the morphological evolution of the microspheres during *in vitro* degradation.”
- **Added Methodological Detail:** A new, detailed protocol for the *in vitro* microsphere degradation study has been added to the Methods section to ensure full transparency and reproducibility.

Revisions in the Manuscript (highlighted in blue):**1. In the Results section 2.3:**

“To assess their degradation profile, the microspheres were incubated *in vitro* under physiological conditions and sampled at various time points. Bright-field microscopy showed a gradual loss of their sharp, spherical morphology over 10 days, while SEM imaging of parallel samples revealed the evolution from a smooth, intact surface to a more porous and eroded structure, indicating progressive degradation (Fig. 4k).”

2. In the Supplementary Materials and Methods Section 2.3 Characterization, a new subsection has been added:**In Vitro Degradation of Tβ4@PLGA Microspheres**

To evaluate the degradation behavior, 20 mg of lyophilized Tβ4@PLGA microspheres were suspended in 1 mL of PBS (pH 7.4) and incubated at 37°C with gentle shaking. At specified time intervals (0, 1, 3, 5, 7, and 10 days), the microspheres were collected by centrifugation. For morphological analysis, one portion of the collected microspheres was directly observed using bright-field microscopy. Another portion was washed with deionized water, flash-frozen in liquid nitrogen, lyophilized, and then sputter-coated with gold for SEM imaging (Phenom ProX).

Reviewer #2, Major Comment #3:

Figure 1, the schematic appears to depict a surgically opened uterine cavity, which is inconsistent with the experimental protocol. Please revise accordingly. The timeline in the bottom row proceeds right-to-left; for clarity and convention, consider reordering it to left-to-right, ending with “healthy growth.”

Response:

We are very grateful to the reviewer for these valuable suggestions to improve the clarity and accuracy of our schematic in Figure 1. The reviewer’s observations are entirely correct, and we apologize for the inconsistencies in the original illustration. We have now thoroughly revised the schematic to ensure it accurately reflects our experimental protocol.

Specifically, we have made two key changes:

1. **Depiction of IUA Induction and Treatment:** We have corrected the illustration to no longer depict a surgically opened uterine horn. The revised schematic now accurately shows a **trans-cervical approach**, where the hydrogel is delivered via a catheter into the uterine cavity. This correctly represents our minimally invasive procedure. We have also used red shading to more clearly signify the area of endometrial injury, rather than a surgical opening.
 2. **Timeline Reordering:** As suggested, we have reordered the timeline at the bottom to follow the conventional **left-to-right progression**. The sequence now starts with “IUA Model,” proceeds through “Hydrogel Treatment,” and concludes with “Fertility Restored,” providing a more intuitive and logical flow.
-

Action Taken:

- The schematic in **Figure 1** has been completely redesigned to accurately depict the trans-cervical administration and the nature of the endometrial injury.
- The timeline in **Figure 1** has been reordered to a standard left-to-right format for improved clarity.

Revisions in the Manuscript (highlighted in blue):

In the Main Figures:

The entire schematic in Figure 1 has been replaced with the revised, corrected version. This change is visual and directly addresses the reviewer's points without altering the figure caption or main text.

We are confident that the revised Figure 1 is now a much clearer and more accurate representation of our study's conceptual framework and experimental design. We thank the reviewer again for their constructive feedback.

Figure 1 | Schematic diagram of material construction and therapeutic mechanism.

Created in <https://BioRender.com>

Reviewer #2, Major Comment #4:

Figures 2, 5–7, 9–12, and S1, DAPI staining across multiple figures is too faint to discern nuclear localization. Please adjust imaging brightness/contrast or use enhanced labeling to improve clarity.

Response:

We sincerely thank the reviewer for this critical observation. Upon re-examination, we agree that the DAPI staining in several of our original immunofluorescence images was indeed too faint, compromising the clear visualization of nuclear localization. This was an oversight in our final figure preparation, and we appreciate the opportunity to correct it.

To address this issue, we have gone back to the original, high-resolution raw images for all mentioned figures. We have carefully re-processed them by appropriately adjusting the brightness and contrast levels for the DAPI channel. This was done meticulously to enhance the visibility of the nuclei without introducing artifacts or oversaturating the signal. The improved, clearer versions of these figures have now replaced the old ones in the revised manuscript. We are confident that the nuclear staining is now distinct and clearly visible, greatly improving the overall quality and interpretability of our data.

Action taken:

- The original raw images for all mentioned figures (now renumbered as **Figures 2, 5, 6, 7, and S1**) were revisited.
 - The brightness and contrast of the DAPI channel were meticulously adjusted for each image to enhance nuclear visibility without creating artifacts.
 - All affected figures in the manuscript have been replaced with the improved, high-clarity versions.
-

Revisions in the Manuscript (highlighted in blue):

In the Main and Supplementary Figures:

The images in Figures 2, 5, 6, 7 (previously Figures 5–7, 9–12) and Supplementary Figure S1 have been replaced with re-processed versions exhibiting enhanced DAPI

channel clarity. This change is visual and improves data presentation without altering the scientific conclusions.

Reviewer #2, Major Comment #5:

Figure 4E, the meaning of “Con” and “Dec” is not defined.

Our Response:

We thank the reviewer for pointing out this lack of clarity. We apologize for the oversight in not defining these abbreviations, which made the figure difficult to interpret. As the reviewer correctly anticipated in a later comment, and upon our careful re-check, these abbreviations "Con" and "Dec" actually appear in **Figures 2 and 3**, not Figure 4E.

"Con" stands for the **control, non-decidualized uterus**, while "Dec" stands for the **decidualized uterus**. This distinction is fundamental to our study's premise.

To rectify this, we have now explicitly defined these abbreviations in the legends of all relevant figures (now renumbered as Figures 2 and 3). This ensures that the experimental groups are clearly and unambiguously described.

Action Taken:

- We have carefully reviewed the manuscript and confirmed the use of "Con" and "Dec" in the figures previously numbered as 2 and 3.
 - The figure legends for these figures (now renumbered as **Figure 2 and Figure 3**) have been revised to explicitly define "Con" (Control, non-decidualized uterus) and "Dec" (Decidualized uterus).
-

Revisions in the Manuscript (highlighted in blue):

In the Figure Legends:

Legend for Figure 2: Representative images of uterine horns from the control (Con, non-decidualized) and decidualized (Dec) groups...

Legend for Figure 3: Comparative analysis of protein expression in control (Con, non-decidualized) and decidualized (Dec) uterine tissues...

Reviewer #2, Major Comment #6:

Figure 5, the meaning of “Geltin” is unclear and lacks a corresponding legend. Please clarify. Is “Mre” meant to indicate “merge”? Ensure consistent and explicit labeling. EdU (A, C) and AM-PI staining (B, D) appear to use endometrial cancer cells. Given the focus on regeneration rather than malignancy, this may be inappropriate—please justify or replace. The figure layout does not match the order of data presentation. Reorganize panels accordingly (e.g., C ↔ B, F ↔ H, J ↔ G) to enhance logical flow. Include quantitative analysis of angiogenesis, such as average capillary length or number of branch points in tube formation assays.

Our Response:

We are very grateful to the reviewer for this detailed and multi-faceted feedback on Figure 5. These comments on labeling, cell choice rationale, figure layout, and quantitative methods are all exceptionally insightful. We have performed a comprehensive revision of the figure and its associated text to address every point raised.

- **(a) Regarding Labeling Clarity:** We sincerely apologize for the typographical errors. "Geltin" was a typo for "**Gelatin**," and "Mre" was indeed intended to be "**Merge**." The color choice in the original image unfortunately obscured the full text. We have corrected these labels in the revised figure for absolute clarity.
- **(b) Regarding the Rationale for Using HUVECs:** This is an excellent point. While primary uterine endothelial cells represent a more direct physiological model, we chose Human Umbilical Vein Endothelial Cells (HUVECs) for this *in vitro* screening assay based on established best practices in the field of angiogenesis research. HUVECs are the **gold standard** for tube formation assays due to their high reproducibility, stability, and well-characterized pro-angiogenic responses. This allows for robust, quantifiable, and comparable results that are benchmarked against a vast body of literature. The use of HUVECs in evaluating the angiogenic potential of biomaterials for uterine repair is well-documented in numerous high-impact studies (e.g., Cai X, *et al.*, *EBioMedicine*, 2022; Zhao H, *et al.*, *Hum Reprod*, 2024). This strategic choice ensures the reliability and broad relevance of our findings.

Reference :

Cai X, Jiang Y, Cao Z, Zhang M, Kong N, Yu L, Tang Y, Kong S, Deng W, Wang H, Sun J, Ding L, Jiang R, Sun H, Yan G. Mst1-mediated phosphorylation of Nur77 improves the endometrial receptivity in human and mice. *EBioMedicine*. 2023 Feb;88:104433.

uane PT, Paterson I, Reeves B, Adlam D, Berneau SC, Renshall L, Brosens JJ, Kimber SJ, Brison DR, Aplin JD, Westwood M. Glucose influences endometrial receptivity to embryo implantation through O-GlcNAcylation-mediated regulation of the cytoskeleton. *Am J Physiol Cell Physiol*. 2024 Sep 1;327(3):C634-C645.

Lu J, Zhang M, Liu Z, Guo L, Huang P, Xia W, Li J, Lv J, Cheung HH, Ding C, Li H, Huang B. NSUN2-Mediated m5C Methylation Impairs Endometrial Receptivity. *Lab Invest*. 2024 Apr;104(4):100327.

Zhao H, Yu M, Li Q, Chen G, Liu X, Bao H. Excessive proliferating cell nuclear antigen attenuates endometrial adhesive capacity and decidualization in patients with recurrent implantation failure. *Hum Reprod*. 2024 May 23;deae111.

- **(c) Regarding Figure Layout:** We agree with the reviewer that the original layout was not optimal for narrative flow. We have completely reorganized the figure to present the data more logically.
- **(d) Regarding the Quantitative Analysis of Angiogenesis:** We thank the reviewer for this suggestion. In fact, our original quantitative analysis was indeed based on the **number of branch points**, which is a rigorous and standard metric for this assay. We regret that this was not made sufficiently clear in the Methods section. We have now provided a more detailed description of this quantification method in the Methods section under “Tube Formation Assay” to further improve methodological transparency.

We are confident that these comprehensive revisions—addressing labeling, cell choice justification, layout, and clarification of quantitative analysis—have resolved all the issues raised and significantly strengthened our manuscript.

Action Taken:

- Typographical errors ("Geltin," "Mre") in the figure have been corrected.
 - The entire figure (now **Figure 5**) has been reorganized for improved logical flow and clarity.
 - A sentence justifying the use of HUVECs has been added to the Results section where this experiment is first described.
 - The Methods section has been significantly expanded to provide a detailed, step-by-step protocol for the quantification of the tube formation assay, explicitly stating the use of "branch points" as the metric.
-

Revisions in the Manuscript (highlighted in blue):

1. In the Main Figures:

The entire Figure 5 and its corresponding legend have been replaced with the revised, reorganized version containing corrected labels.

2. In the Results Section:

(In the paragraph describing the in vitro angiogenesis results) ...To assess the pro-angiogenic potential of our hydrogels, we utilized a well-established in vitro tube formation assay with Human Umbilical Vein Endothelial Cells (HUVECs), the gold-standard cell line for this application ensuring high reproducibility and comparability...

3. In the Methods Section (under "Tube Formation Assay"):

Quantitative analysis was performed on three to five random fields per well. Using ImageJ software (v1.53, NIH, USA), the total number of branch points (defined as the intersection of three or more tubules) was counted by an investigator blinded to the treatment conditions. The average count per well was used for statistical comparison across groups.

Figure 5 | In Vitro Evaluation of the Pro-Regenerative Effects of UdECM Hydrogel. (a) Representative images of EdU staining in Ishikawa cells treated with UdECM. (b) Representative images of AMPI staining assessing the in vitro biocompatibility of UdECM in Ishikawa cells. (c) Quantification of EdU-positive cells. (d) Quantification of AM and PI-positive cells. (e) Representative images of scratch assays demonstrating the effect of UdECM

on mouse endometrial stromal cell migration. (f) Representative images of tube formation assays demonstrating the effect of UdECM on hUVEC tube formation. (g) Representative images of Ki67 staining in Ishikawa cells treated with UdECM. (h) Quantification of mouse endometrial stromal cell migration distance. (i) Quantification of Ki67-positive cells. (j) Quantitative assessment of branch points in hUVEC tube networks Scale bar, 100 μ m. Data are presented as mean \pm s.d. (n = 3 biologically independent samples). *P < 0.05, **P < 0.01, ***P < 0.001, compared to the Model group (one-way ANOVA with Tukey's post hoc test).

Reviewer #2, Major Comment #7:

Figure 6, a schematic illustration showing how endometrial thickness was measured (e.g., anatomical landmarks, measurement direction, number of sections) would improve reproducibility.

Our Response:

We wholeheartedly agree with the reviewer and thank them for this excellent suggestion, which prompted us to further refine and more transparently present our analytical methodology. A clear, standardized procedure for measuring endometrial thickness is indeed crucial for reproducibility.

In response, we have taken comprehensive action. First, as requested, we have created a new supplementary figure (**Figure S4**) to schematically illustrate our measurement protocol. More importantly, the process of creating this schematic led us to a more rigorous and scientifically honest way of analyzing our data.

We recognized that a meaningful measurement of "endometrial thickness" is predicated on the successful regeneration of a basic, stratified endometrial structure with a patent uterine lumen. Upon close re-examination of our histological sections (**Figure 6c**), we confirmed that this prerequisite was met in the Sham and all hydrogel-treated groups. However, in the Model and T β 4@PLGA-only groups, the severe injury resulted in luminal obliteration and a complete lack of a discernible endometrial structure.

Consequently, a meaningful thickness measurement was biologically impossible in these two groups, as the primary anatomical landmark (the luminal surface) was absent. To reflect this critical biological outcome—the complete failure of structural regeneration—these samples were assigned a thickness value of zero in our quantitative

analysis. This approach, we believe, is the most accurate representation of the profound regenerative failure observed.

To visually aid the reader, we have now annotated the representative H&E images in **Figure 6c**, using green dashed lines to demarcate the myometrial-endometrial border and blue double-headed arrows to indicate the thickness measurement in the successfully regenerated groups. We are confident that this multi-faceted revision—providing a schematic, annotating the actual data, and clarifying the scientific rationale for our quantification—not only fully addresses the reviewer's concern but also adds significant scientific depth to our analysis.

Action Taken:

- A new supplementary figure (**Figure S4**) has been created to schematically illustrate the methodology for endometrial thickness measurement in both regenerated and severely damaged uteri.
- The main figure (**Figure 6c**) has been updated with annotations (green dashed lines and blue double-headed arrows) to visually demonstrate the measurement on representative H&E images.
- The Methods section has been substantially revised to detail this nuanced methodology, clearly defining what constitutes a measurable endometrium and explaining the rationale for assigning a "zero" value to non-regenerated samples.
- The legend for **Figure 6** has been amended to explain the new annotations and the "zero value" assignment.

Revisions in the Manuscript (highlighted in blue):**1. In the Supplementary Information:**

(Figure S4 and its legend) A new figure and legend have been added:

Figure S4 | Schematic illustration of the methodology for endometrial thickness measurement. (a) In a regenerated uterus exhibiting a patent lumen, endometrial thickness was quantified by averaging four measurements (indicated by dashed lines, S1-S4) taken at equidistant points. Each measurement represents the perpendicular distance from the myometrial-endometrial junction to the luminal surface. (b) In a severely damaged uterus, characteristic of the Model group, the uterine cavity was obliterated and lacked a discernible endometrial structure. Consequently, a meaningful thickness measurement was biologically impossible, and these samples were assigned a value of zero to represent a complete failure of structural regeneration.

2. In the Main Manuscript (Figure 6):

(Visual Change in Figure 6c): Green dashed lines have been overlaid on representative images to indicate the myometrial-endometrial border. Blue double-headed arrows have been added to show the direction and extent of thickness measurement. (Revision to Legend of Figure 6): ...The legend has been amended to include the statement: "The green dashed line indicates the myometrial-endometrial border, and the blue double-headed arrows illustrate the thickness measurement. Endometrial thickness was recorded as zero for the Model and T β 4@PLGA groups, reflecting a complete failure to regenerate a structured endometrium."

Figure 6 | DEndo-UdECM/Tβ4@PLGA hydrogel promotes functional endometrial regeneration and suppresses fibrosis in a murine model of endometrial injury. (c) Representative H&E-stained uterine cross-sections. The DEndo-UdECM/Tβ4@PLGA group shows significant restoration of the endometrial architecture, including a patent lumen and numerous glands (indicated by red asterisks). The green dashed line indicates the myometrial-endometrial border, and the blue double-headed arrows illustrate the thickness measurement. Scale bar, 100 μm. Data are presented as mean ± s.d. (n = 6 biologically independent samples). *P < 0.05, **P < 0.01, ***P < 0.001, compared to the Model group. Statistical significance was determined by one-way ANOVA followed by Tukey's post hoc test.

3. In the Supplementary Materials and Methods Section (under "4.2 Histological and Morphometric Analysis"):

Fourteen days post-treatment, mice from each group (n=6/group) were euthanized, and uteri were harvested for analysis. A portion of the uterus was snap-frozen for Western blotting, while the remainder was fixed in 4% PFA for 24 hours. Fixed tissues were processed through graded ethanol dehydration, xylene clearing, and paraffin embedding. Serial transverse sections (5 μm thick) were prepared for subsequent analysis. **H&E staining** was performed to evaluate overall tissue architecture, including endometrial gland number, morphology, and inflammatory cell infiltration. Endometrial thickness was quantified from H&E-stained sections using ImageJ software. A prerequisite for measurement was the presence of a regenerated, patent uterine lumen lined by a continuous epithelium. For groups meeting this criterion, thickness was defined as the average of four equidistant measurements taken perpendicularly from the myometrial-endometrial junction to the luminal surface. In the Model and Tβ4@PLGA groups, where severe injury resulted in luminal obliteration and a lack of a structured endometrium, thickness was deemed biologically unmeasurable and assigned a value of zero to reflect complete regenerative failure, as illustrated in Supplementary Figure S4.

Reviewer #2, Major Comment #8:

Line 609, the statement that “functional restoration of glands was evidenced by the near-complete recovery of the key transcription factor FOXA2” should be elaborated. Please explain the physiological relevance of FOXA2 and how it reflects glandular function.

Our Response: We thank the reviewer for this insightful question and for prompting us to elaborate on the critical role of FOXA2. The reviewer is correct that simply stating its recovery is insufficient; explaining its physiological significance is essential to substantiate our claim of “functional restoration.”

FOXA2 (Forkhead box a2) is not merely a marker, but a **master transcriptional regulator** indispensable for uterine gland development (glandulogenesis), differentiation, and secretory function. A wealth of literature, particularly from pioneering work by Spencer and colleagues, has unequivocally established its central role:

1. **Essential for Gland Formation and Maintenance:** FOXA2 is required for the neonatal development of uterine glands. Its conditional knockout in the murine uterus results in a complete absence of glands, leading to infertility (Kelleher et al., *PNAS*, 2017). Therefore, the re-emergence of robust FOXA2 expression in our treated group is strong evidence of the regeneration of a glandular cell population with the correct developmental identity.
2. **Driver of Glandular Secretions (Histotroph):** Functioning glands must secrete essential factors (the “histotroph”) that support embryo survival and implantation. FOXA2 directly regulates the expression of many of these critical secretory products, such as secreted phosphoprotein 1 (SPP1) and leukemia inhibitory factor (LIF) (Kelleher et al., *Endocr Rev*, 2019; Jia et al., *Nat Commun*, 2025). Thus, restoring FOXA2 expression strongly implies the restoration of the gland’s capacity to produce a supportive luminal fluid.
3. **Definitive Marker of Functional Glandular Epithelium:** In both human and mouse models, FOXA2 is a definitive marker of the glandular epithelium. Its presence is synonymous with a differentiated, functional glandular phenotype,

as demonstrated in studies utilizing advanced models like uterine organoids (Fitzgerald et al., *PNAS*, 2019) and in complex physiological processes like decidualization (Dhakal et al., *PNAS*, 2020).

Based on this robust evidence, we have expanded our Discussion to provide a solid mechanistic basis for our claim that our hydrogel treatment leads to true functional restoration of the uterine glands.

Reference:

Fitzgerald HC, Dhakal P, Behura SK, Schust DJ, Spencer TE. Self-renewing endometrial epithelial organoids of the human uterus. *Proc Natl Acad Sci U S A*. 2019 Nov 12;116(46):23132-23142.

Kelleher AM, DeMayo FJ, Spencer TE. Uterine Glands: Developmental Biology and Functional Roles in Pregnancy. *Endocr Rev*. 2019 Oct 1;40(5):1424-1445.

Kelleher AM, Peng W, Pru JK, Pru CA, DeMayo FJ, Spencer TE. Forkhead box a2 (FOXA2) is essential for uterine function and fertility. *Proc Natl Acad Sci U S A*. 2017 Feb 7;114(6):E1018-E1026.

Jia Z, Li B, Matsuo M, Dewar A, Mustafaraj A, Dey SK, Yuan J, Sun X. Foxa2-dependent uterine glandular cell differentiation is essential for successful implantation. *Nat Commun*. 2025 Mar 12;16(1):2465.

Dhakal P, Kelleher AM, Behura SK, Spencer TE. Sexually dimorphic effects of forkhead box a2 (FOXA2) and uterine glands on decidualization and fetoplacental development. *Proc Natl Acad Sci U S A*. 2020 Sep 22;117(38):23952-23959.

Action Taken:

- We have substantially expanded the Discussion section to incorporate a detailed explanation of FOXA2's physiological relevance.
 - This new text explicitly links the observed FOXA2 recovery in our study to the restoration of glandular identity, secretory potential, and overall uterine function.
 - We have cited the key literature (e.g., Kelleher et al., 2017; Jia et al., 2025) to support these claims.
-

Revisions in the Manuscript (highlighted in blue):

1. In the Discussion Section:

The original sentence has been replaced with the following expanded text to provide a more thorough explanation: " Furthermore, the robust re-expression of the master transcription factor FOXA2 (Fig. 6f, g) signifies more than mere architectural rebuilding [54]; it indicates the re-establishment of glandular functional competence, which is indispensable for producing the histotroph required for embryo survival and implantation [41, 55]."

2. In the Reference List:

The following references have been added to the manuscript's reference list:

Reviewer #2, Major Comment #9:

Figure 7, Panel order needs reorganization for narrative consistency. The blue channel indicating DNA (DAPI) staining is not labeled. A large discrepancy is observed between fibrotic marker expression in IHC and immunofluorescence—please explain potential methodological or interpretive reasons.

Our Response: We are very grateful to the reviewer for their meticulous evaluation of this figure (now renumbered as **Figure 6**). These points are well-taken, and addressing them has significantly enhanced the figure's clarity and the scientific narrative of our anti-fibrosis results. We address each point in turn.

(a) Regarding Panel Order and DAPI Labeling:

We agree completely. The original panel order did not follow the most logical narrative flow, and the DAPI label was an unfortunate omission. We sincerely apologize for these oversights and have corrected them in the revised manuscript.

(b) Regarding the Perceived Discrepancy Between Global Fibrosis (Masson's) and Specific Marker (IF) Expression:

This is a very insightful observation, and we thank the reviewer for giving us the opportunity to clarify our multi-faceted strategy for assessing fibrosis. The perceived difference is not a contradiction, but rather a reflection of the **complementary biological information** captured by these two distinct techniques. Our experimental

design deliberately leveraged both methods to construct a comprehensive picture of fibrosis attenuation:

- **Masson’s Trichrome Staining (The ‘Macro’ View):** Masson’s staining provides a **holistic, architectural assessment** of fibrosis by visualizing total collagen deposition (stained blue). It effectively serves as our primary outcome measure for overall fibrotic burden, demonstrating that our treatment significantly reduces the global extent of scar tissue.
- **Immunofluorescence for Specific Markers (The ‘Micro’ View):** In contrast, IF provides **mechanistic, molecular-level insight**. By using highly specific antibodies (e.g., against α -SMA, Collagen I), we can dissect the composition of the fibrotic matrix and pinpoint the key pro-fibrotic molecules targeted by our therapy.

Therefore, the two datasets are synergistic and mutually reinforcing. The Masson’s data confirm the **‘what’**—that overall fibrosis is reduced. The IF data explain the **‘how’**—by showing that this reduction is achieved, at least in part, by suppressing the expression and deposition of specific, key pro-fibrotic proteins. The visual differences are thus expected and informative, representing the distinction between a landscape-level view of scarring and a molecular-level close-up.

Action Taken:

- The panels in the original Figure 7 (now **Figure 6**) have been completely reorganized to align with the progression of our textual description, ensuring a more intuitive presentation.
- The figure and its legend have been updated to explicitly identify the blue channel as DAPI staining for cell nuclei.
- We have refined the Results section text corresponding to this figure. We now explicitly introduce Masson’s staining as our metric for “overall fibrosis” and IF as our tool for investigating “specific molecular components of the fibrotic matrix.”

Revisions in the Manuscript (highlighted in blue):

1. In the Main Figures:

The original Figure 7 has been removed. Its components have been integrated with the original Figure 6 to create a new, consolidated Figure 6. This new figure has been completely reorganized for improved narrative flow, and all immunofluorescence panels now clearly label the DAPI channel.

2. In the Results Section 2.6:

The text introducing the anti-fibrosis results has been revised to clarify the methodological rationale: "To comprehensively evaluate the anti-fibrotic efficacy of our hydrogel system, we employed a dual-assessment strategy. Masson's trichrome staining was used to assess the overall fibrotic burden by visualizing total collagen deposition (Fig. 6h, i), providing an architectural overview of scar tissue reduction. Concurrently, immunofluorescence staining for key pro-fibrotic markers, such as Collagen I and α -SMA, was performed to gain mechanistic insight into the specific molecular components being modulated by our therapy (Fig. 6j, k, l)."

3. In the Figure Legends:

(Legend for Figure 6): ...For immunofluorescence images, cell nuclei were counterstained with DAPI (blue)...

Reviewer #2, Major Comment #10:

Figure 8, only the postnatal development of pups from the DEndo-UdECM/T β 4@PLGA group is shown. Were offspring from other treatment groups monitored? Comparative data should be provided to validate the functional benefit.

Our Response:

We thank the reviewer for this crucial question, which allows us to clarify our reporting strategy for this ultimate functional endpoint. The reviewer is absolutely correct to ask for comparative data and a clear justification for our presentation choices.

To be fully transparent: **we monitored all live-born pups from all experimental groups until weaning.** The key finding was that the **postnatal survival rate was 100% for all live-born offspring, regardless of the treatment group,** with no gross developmental abnormalities observed in any pup.

Our decision to visually present the developmental timeline for *only* the DEndo-UdECM/T β 4@PLGA group was a deliberate choice based on two core principles of effective scientific communication:

1. **Narrative Focus on the Primary Therapeutic Success:** The primary goal of our therapy was to restore the capacity to carry a pregnancy to term and deliver a healthy-sized litter. As shown in our quantitative data (now in **Figure 7e**), the DEndo-UdECM/T β 4@PLGA group was the **only group to restore the number of live offspring to a level statistically comparable to the healthy Sham controls**. It was, therefore, the only treatment to achieve our primary definition of “complete functional restoration.” To maintain a clear and focused narrative, we chose to illustrate the full successful trajectory (from birth to weaning) for this “lead group” as it represents the complete therapeutic outcome our study aimed to achieve.
2. **Avoiding Redundancy and Maximizing Clarity:** Since the postnatal development of pups was uniformly healthy and visually indistinguishable across all groups that produced live offspring, presenting multiple panels showing identical-looking, healthy pups would be highly redundant. Such a presentation would consume significant figure space without providing any new comparative information beyond what the “100% survival rate” statistic already conveys. We therefore selected the most representative case—the one that successfully restored litter size to normal levels—to illustrate this positive, but uniform, outcome.

We are confident that this approach—transparently reporting the uniform 100% survival rate for all while using the main figure to focus on the narrative of the single group that fully restored fertility—is the most scientifically sound and clear way to present our findings.

Action Taken:

- To ensure full transparency, we have now explicitly stated in the Results section that the postnatal survival rate was 100% across all pups born in any group.

- We have added a new column for “**Postnatal Survival Rate (%)**” to our new Supplementary Table (**Table S1**), where every group with live births is listed with a value of “100%”.
 - We have refined the legend for the representative image (now as **Figure 7f**) to explicitly mention that pups from other groups also developed normally.
-

Revisions in the Manuscript (highlighted in blue):

1. In the Results Section (Section 2.7):

This result signifies a near-complete recovery of the uterus’ s capacity to support a full-term pregnancy. Importantly, we monitored the postnatal development of all live-born offspring. Pups from all groups, including the DEndo-UdECM/Tβ4@PLGA group, exhibited a 100% survival rate to weaning and normal growth patterns, with no observable deformities (Table S1). As the developmental trajectory was uniformly healthy across all viable offspring, we present the postnatal development of the DEndo-UdECM/Tβ4@PLGA group as a representative illustration of this successful outcome (**Fig. 7f**).

2. In the Supplementary Information:

A new column titled 'Postnatal Survival Rate (%)' has been added to Table S1. For all groups that had live births, this column now shows a value of '100'.

3. In the Figure Legends:

(Legend for Figure 7f): Representative images showing the postnatal development of pups from the DEndo-UdECM/Tβ4@PLGA group at different time points. Pups from all other groups that produced live offspring also exhibited a 100% survival rate and normal development (see Table S1 for details).

Reviewer #2, Major Comment #11:

Figure 9, consider supplementing M2 macrophage polarization data with flow cytometry to enhance rigor and quantification.

Response:

We are profoundly grateful to the reviewer for this excellent suggestion, which prompted us to significantly strengthen our analysis of the immune microenvironment. We fully agree that a rigorous, multi-marker approach is the gold standard for substantiating claims of immunomodulation.

In our original submission, we had already initiated this investigation by demonstrating a significant downregulation of the canonical M1 marker **CD86** and an upregulation of the M2 marker **Arginase-1 (Arg-1)**. However, inspired by the reviewer's call for enhanced rigor, we have now **substantially expanded this analysis** to provide a more comprehensive and powerful picture.

To fortify our initial findings, we have now incorporated **two additional canonical markers**: the pro-inflammatory M1 marker **Inducible Nitric Oxide Synthase (iNOS)** and the pro-regenerative M2 marker **CD206**.

The updated Figure 9 (now renumbered as **Figure 8**) presents this comprehensive four-marker panel. The new data for iNOS and CD206 **powerfully corroborate our initial findings**, now unequivocally demonstrating a **dual immunomodulatory effect** of our hydrogel. The results show a consistent and marked downregulation of **both** M1 markers (CD86 and iNOS) and a significant upregulation of **both** M2 markers (Arg-1 and CD206).

This multi-faceted evidence, showing consistent trends across two independent markers for each phenotype, provides an exceptionally strong foundation for our claim that the hydrogel actively rebalances the local immune microenvironment. We believe this spatially-resolved, now four-marker analysis, offers a definitive and mechanistically insightful conclusion. We are sincerely thankful to the reviewer, as their suggestion has directly led to a much more robust and conclusive immunological component of our manuscript.

Action Taken:

- Acknowledged the reviewer's point and embraced the call for enhanced rigor.
- **Expanded our existing M1/M2 analysis** by adding immunofluorescence staining and quantification for two new markers: the M1 marker **iNOS** and the M2 marker **CD206**.

- This complements our original analysis of **CD86** and **Arg-1**.
 - Completely updated the old Figure 9 into a new, comprehensive **Figure 8** that visually and quantitatively presents the full **four-marker** panel.
 - Thoroughly revised the **Methods, Results, and Figure Legend** sections to reflect this expanded and more powerful analysis.
-

Revisions in the Manuscript (highlighted in blue):

1. In the Main Figures:

The entirety of the previous Figure 9 has been replaced with the new, expanded Figure 8, featuring the comprehensive four-marker M1/M2 balance analysis.

2. In the Results Section (Section 2.9):

"2.9 DEndo-UdECM/Tβ4@PLGA Treatment Rebalances the Immune Microenvironment towards a Pro-Regenerative M2 Phenotype

Given that macrophage polarization is a critical determinant of the inflammatory-reparative balance during tissue repair[3], we performed a **comprehensive four-marker immunofluorescence analysis** to rigorously investigate the influence of our composite hydrogel on macrophage phenotype at the injury site. We assessed two canonical M1 markers (**CD86** and **iNOS**) and two canonical M2 markers (**Arginase-1** and **CD206**).

First, we examined the pro-inflammatory M1 phenotype. Expression of **CD86** was significantly upregulated in the Model group compared to the Sham group ($P < 0.001$) (**Fig. 8a, e**). Tβ4@PLGA treatment significantly reduced CD86+ cell density ($P < 0.001$ vs. Model). Both CEndo-UdECM and DEndo-UdECM treatments further decreased CD86+ cell density ($P < 0.001$ vs. Model). The DEndo-UdECM/Tβ4@PLGA group exhibited the lowest CD86+ cell density, comparable to the Sham group ($P > 0.05$ vs. Sham). **Consistent with these findings**, the expression of the M1 marker **iNOS** followed a nearly identical pattern, with the DEndo-UdECM/Tβ4@PLGA treatment most effectively suppressing its expression to levels approaching the Sham group (**Fig. 8b, f**).

Conversely, we assessed the pro-reparative M2 phenotype. Expression of **Arginase-1** (**Arg-1**) was significantly diminished in the Model group compared to Sham (**Fig. 8c**,

g). Tβ4@PLGA treatment alone did not significantly alter Arg-1+ cell density compared to the Model group ($P > 0.05$). Both CEndo-UdECM and DEndo-UdECM treatments significantly increased Arg-1+ cell density compared to the Model group. The DEndo-UdECM/Tβ4@PLGA group displayed the highest Arg-1+ cell density, similar to the Sham group. **This pro-regenerative shift was powerfully corroborated by the analysis of CD206**, another key M2 marker. The DEndo-UdECM/Tβ4@PLGA group demonstrated the most profound upregulation of **CD206** expression, significantly higher than all other treatment and control groups (**Fig. 8d, h**).

Taken together, this comprehensive four-marker analysis unequivocally demonstrates that the DEndo-UdECM/Tβ4@PLGA treatment orchestrates a profound shift in the local immune microenvironment. It does so by **concurrently suppressing the pro-inflammatory M1 phenotype** (evidenced by reduced CD86 and iNOS) and **promoting the pro-regenerative M2 phenotype** (evidenced by elevated Arg-1 and CD206), thereby creating optimal conditions for endometrial repair."

3. In the Figure Legends:

The legend for Figure 8 has been completely rewritten to accurately describe the new, comprehensive four-marker M1/M2 balance analysis (CD86, iNOS, Arg-1, CD206), the quantification method, and the corresponding results.

Figure 8 | The DEndo-UdECM/Tβ4@PLGA hydrogel promotes a pro-regenerative M2 macrophage polarization in the injured endometrium. Immunofluorescence analysis of macrophage infiltration and polarization in regenerated endometrial tissue 14 days post-treatment. **(a, b)** Representative immunofluorescence images **(a)** and corresponding quantification **(b)** of the pan-macrophage marker CD68 (red). **(c, d)** Representative immunofluorescence images **(c)** and corresponding quantification **(d)** of the M2 macrophage marker CD163 (red). **(e, f)** Representative immunofluorescence images **(e)** and corresponding quantification **(f)** of the M2-associated enzyme Arginase-1 (Arg-1, red). **(g, h)** Representative immunofluorescence images **(g)** and corresponding quantification **(h)** of the M2 macrophage marker CD206 (red). For all immunofluorescence panels, cell nuclei

were counterstained with DAPI (blue). The top row shows zoomed-out fields of view, while the second row provides high-magnification insets of the areas indicated by the red dashed boxes. Scale bar, 100 μ m. Data are presented as mean \pm s.d. (n = 6 biologically independent samples). *P < 0.05, **P < 0.01, ***P < 0.001, compared to the Model group. Statistical significance was determined by one-way ANOVA followed by Tukey's post hoc test.

Reviewer #2, Major Comment #12:

Figure 10, the single-channel image for CD34 lacks visible fluorescence signal. Verify if signal acquisition or channel intensity was appropriate. If necessary, adjust exposure or contrast.

Response:

We sincerely thank the reviewer for pointing out this issue with the CD34 image. We agree that the fluorescence signal in the single-channel view was indeed too faint in the originally submitted manuscript, which could lead to misinterpretation.

We have confirmed that this was not an issue with the experimental staining or signal acquisition itself, but rather an oversight during the final figure preparation, where the brightness and contrast settings were suboptimal for the single-channel display.

Action Taken:

- We have returned to the original, raw image data for this figure.
- We have carefully adjusted the brightness and contrast of the CD34 channel to make the positive signal clearly visible, while ensuring that the adjustments do not create artificial signals or obscure the background.
- The improved, clearer version of the figure (now renumbered as **Figure 9**), including the enhanced single-channel CD34 image, has replaced the old one in the revised manuscript.

We appreciate the reviewer's sharp eye, which has helped us improve the clarity and overall quality of our figures.

Revisions in the Manuscript (highlighted in blue):

In the Main Figures:

The entire image panel for Figure 9 (previously Figure 10) has been replaced with the re-processed version that has optimized brightness and contrast for the CD34 channel. This is a visual change to improve data presentation and does not alter the scientific conclusions.

Figure 9 | DEndo-UdECM/Tβ4@PLGA hydrogel promotes multi-lineage endometrial regeneration. Uterine tissues were analysed 14 days post-treatment across all seven experimental groups. **d**, Representative immunofluorescence for the angiogenesis marker CD34 (red).

Reviewer #2, Major Comment #13:

Figure 11, a schematic depiction of the TGFβ1/Smad signaling pathway and its activation by the treatment would help contextualize the molecular findings.

Our Response:

We are sincerely grateful to the reviewer for this excellent suggestion, which has significantly enhanced the clarity and impact of our mechanistic findings. We completely agree that a schematic diagram serves as a powerful visual summary, effectively contextualizing our Western Blot data and making our proposed anti-fibrotic mechanism more accessible and memorable for the reader. We appreciate this constructive advice.

Action Taken:

- A new schematic diagram has been designed and incorporated into the manuscript as **Figure 10a** (the manuscript's figure numbering has been updated).
 - This diagram visually contrasts two scenarios:
 - i. **The Pathological State (IUA):** It depicts how excessive TGF- β 1 binds to its receptor, triggering the phosphorylation of Smad2/3. The activated p-Smad2/3 complex then translocates to the nucleus to promote the transcription of pro-fibrotic genes (e.g., *Colla1*, *Acta2*), leading to tissue fibrosis.
 - ii. **The Therapeutic Intervention:** It clearly illustrates how our DEndo-UdECM/T β 4@PLGA hydrogel intervenes by inhibiting the activation of the TGF- β 1/Smad cascade. This blockade reduces Smad2/3 phosphorylation, thereby attenuating downstream fibrotic gene expression and facilitating endometrial regeneration.
 - This new schematic directly corresponds to and visually summarizes the quantitative Western Blot data presented in the subsequent panels of **Figure 10**.
-

Revisions in the Manuscript (highlighted in blue):

1. In the Main Figures:

A new panel (Figure 10a) containing the schematic diagram has been added to the beginning of Figure 10.

2. In the Figure Legends:

The legend for Figure 10 has been updated to include a description of the new schematic in panel a.

3. In the Main Text (Results Section):

The text has been updated to refer to the new schematic when introducing the molecular mechanism results (e.g., “As illustrated in the schematic in Fig. 10a, our hydrogel was hypothesized to intervene in the canonical TGF- β 1/Smad signaling cascade...”).

Fig. 10 | The DEndo-UdECM/Tβ4@PLGA hydrogel simultaneously suppresses pyroptosis and canonical TGF-β/Smad signaling.

a, Schematic illustrating the dual mechanism of action, where the composite hydrogel inhibits both the TGF-β-driven fibrotic pathway and the pyroptotic inflammatory pathway.

Reviewer #2, Major Comment #14:

Figure 12, clarify the rationale for choosing IHC versus immunofluorescence for each marker. Is it based on antibody availability, signal specificity, or detection sensitivity?

Our Response:

We are deeply grateful to the reviewer for this highly incisive question, which allows us to elaborate on the rigorous methodological validation process that underpins the data presented in Figure 11. The reviewer’s intuition is correct; our choices were neither arbitrary nor based on antibody availability, but were the result of a deliberate, evidence-based optimization strategy for each individual marker.

Our guiding principle was to ensure the **highest possible data fidelity for each specific antibody-target pair**. To achieve this, for every key marker shown in Figure 11, we conducted a **rigorous head-to-head comparison** of both Immunohistochemistry (IHC) and Immunofluorescence (IF) techniques during our initial experimental setup.

The final technique presented in the manuscript was the one that **empirically demonstrated superior performance** based on a stringent set of pre-defined criteria:

1. **Signal Specificity & Localization:** The signal must be cleanly localized to the correct subcellular compartment (e.g., nuclear for PR, apical membrane for MUC1) without off-target staining.

2. **Signal-to-Noise Ratio:** The method must produce a strong, specific signal with minimal non-specific background, ensuring unambiguous interpretation.
3. **Preservation of Morphological Context:** The protocol must maintain the integrity of the tissue architecture, which is crucial for interpreting expression patterns within the endometrial glands and stroma.

Here is a specific breakdown of how this validation process informed our choices for key markers in Figure 11:

- **Example 1: Progesterone Receptor (PR, Fig. 11a).** With the specific PR antibody clone we used, while both techniques produced a nuclear signal, the **IF protocol** yielded an exceptionally clean signal with virtually zero cytoplasmic background. This "cleanliness" was superior to our IHC results for this antibody, allowing for highly robust, reproducible, and unbiased ratiometric quantification (counting PR-positive nuclei against total DAPI-stained nuclei). Therefore, IF was selected as the more rigorous method for presenting the PR data.
- **Example 2: Integrin β 3 (Fig. 11h).** Conversely, for our Integrin β 3 antibody, the **IHC protocol** produced an outstandingly crisp and intense signal precisely on the luminal epithelial cell membrane, clearly delineated against the detailed hematoxylin-stained tissue architecture. In our hands, the corresponding IF protocol, while specific, resulted in a more diffuse signal that did not offer the same level of sharp morphological detail. Thus, IHC was unequivocally the superior method for visualizing this marker's critical localization.

In summary, our approach was not a pre-determined "one-size-fits-all" strategy, but rather an **evidence-based, marker-by-marker optimization process**. This ensures that the data presented for each target in Figure 11 represents the highest quality, most specific, and most biologically accurate result we could achieve in our experimental system.

We trust this detailed explanation clarifies our methodological strategy and demonstrates the rigor behind our experimental choices. We thank the reviewer for prompting us to make this important aspect of our work more explicit.

Action Taken:

- To ensure this rationale is clear to all readers, we have added a concise explanatory sentence to the **Materials and Methods** section, under the “Immunohistochemistry and Immunofluorescence” subsection.
-

Revisions in the Manuscript (highlighted in blue):**In the Materials and Methods Section (under the “Immunohistochemistry and Immunofluorescence” subsection):**

The choice between IHC and IF for each marker was determined through a rigorous, head-to-head validation process. For each antibody, both techniques were initially tested. The method ultimately presented was selected based on its empirically demonstrated superiority in providing the optimal combination of signal specificity, signal-to-noise ratio, and preservation of morphological context for that specific target in our experimental model.

Reviewer #2, Major Comment #15:

Figure S3, the toxicity assay employed is insufficient to conclude biocompatibility. Standard in vivo biocompatibility evaluation should be conducted—e.g., subcutaneous implantation followed by histology and IHC at defined time points.

Response: We are profoundly grateful to the reviewer for this insightful and critical point. We completely agree that a robust biocompatibility assessment is paramount and requires a multi-faceted approach. The reviewer's suggestion has pushed us to construct a truly comprehensive validation of our biomaterial's safety, which we now believe is a significant strength of the manuscript.

To address this, we have moved beyond the initial systemic safety screen and now present a robust, **two-tiered evaluation** that systematically differentiates between systemic safety and intrinsic local biocompatibility:

- **Tier 1: Confirmation of Systemic Safety.** Our original data (now **Figure S3a**) established the systemic safety of our hydrogel. H&E analysis of all major organs following intrauterine administration revealed no off-target pathological changes, confirming that the material and its degradation products are not systemically toxic.
- **Tier 2: Gold-Standard Assessment of Intrinsic Local Biocompatibility.** While the systemic data was reassuring, we concur with the reviewer that it does not address the material's inherent interaction with host tissue. Therefore, in direct response to this critical point, we performed the **recommended subcutaneous implantation study** in a neutral, non-diseased tissue environment. The results (new **Figure S3b-c**) provide a definitive assessment of local biocompatibility over 21 days:
 - **H&E staining (Fig. S3b)** revealed only a mild and transient cellular infiltrate at day 7, representing a typical acute host response. Critically, this resolved significantly by day 14 and was indistinguishable from the PBS control by day 21, demonstrating the **absence of a chronic inflammatory reaction**.
 - Furthermore, **Masson's trichrome staining (Fig. S3c)** specifically addressed the potential for a fibrotic foreign body response. This analysis revealed a normal, well-organized collagen architecture, comparable to the PBS control at all time points. Crucially, we observed **no signs of excessive collagen deposition or the formation of a dense fibrous capsule**, which is the hallmark of a negative foreign body reaction.

In summary, we have now presented a comprehensive safety narrative that flows from systemic safety (Tier 1) to fundamental local biocompatibility (Tier 2). This robust evidence confirms that the therapeutic effects observed in our IUA model are the result of a biocompatible, bio-functional material, not a confounding inflammatory or fibrotic artifact. We are sincerely indebted to the reviewer for this suggestion, as it has allowed us to present a much more rigorous and compelling case for the translational potential of our technology.

Action Taken:

1. **New Experiment Performed:** We conducted a comprehensive *in vivo* biocompatibility study via subcutaneous implantation of our hydrogels in healthy mice, as recommended by the reviewer.
2. **Multi-Time Point Analysis:** Tissues were harvested at 7, 14, and 21 days post-implantation to assess both the acute and chronic host response.
3. **Gold-Standard Histological Evaluation:** We performed both H&E staining (for cellular infiltration and inflammation) and Masson's trichrome staining (for fibrotic capsule formation) on the explanted tissues.
4. **Revised Supplementary Figure:** The original systemic toxicity data has been integrated with the new local biocompatibility data into a new, comprehensive **Figure S3**. A new, detailed figure legend has been written.
5. **Updated Methods Section:** A new sub-section detailing the subcutaneous implantation protocol has been added to the Supplementary Materials and Methods.

Revisions in the Manuscript (highlighted in blue):**1. In the Supplementary Figures:**

The original Figure S3 has been replaced with a new, multi-panel Figure S3 that includes both the original systemic toxicity data (now panel a) and the new local biocompatibility data from the subcutaneous implantation study (new panels b and c). The figure legend has been completely rewritten to reflect this comprehensive assessment.

2. In the Supplementary Materials and Methods:**(4.5 In Vivo Biocompatibility Assessment)**

To evaluate local tissue biocompatibility, healthy female ICR mice (8 weeks old) were subcutaneously injected in the dorsal region with 100 μ L of PBS (n=9), T β 4@PLGA@CEndo-UdECM hydrogel (n=9), or T β 4@PLGA@DEndo-UdECM hydrogel (n=9). At 7, 14, and 21 days post-injection, 3 mice were euthanized, and the skin tissue surrounding the injection site was harvested. The tissue samples were

fixed in 4% paraformaldehyde, embedded in paraffin, sectioned into 5 μm slices, and subsequently stained with Hematoxylin and Eosin (H&E) and Masson's trichrome staining for histological evaluation. were fixed in 4% paraformaldehyde, embedded in paraffin, sectioned into 5 μm slices, and subsequently stained with Hematoxylin and Eosin (H&E) and Masson's trichrome staining for histological evaluation.

Fig. S3 | Systemic and local biocompatibility assessment of the hydrogel formulations.

a, To evaluate systemic safety, major organs (heart, liver, spleen, lung, kidney) were collected 14 days post-treatment and subjected to haematoxylin and eosin (H&E) staining. Representative images from all experimental groups show normal tissue histology, with no evidence of pathological damage, inflammation, or cytotoxicity. **b**, **c**, To assess local tissue response and material degradation, the composite hydrogels were implanted subcutaneously in mice. The tissue surrounding the implants was harvested at 7, 14, and 21 days post-implantation. **b**, H&E staining reveals a mild and progressively resolving inflammatory response over the 21-day period for all hydrogel groups. **c**, Masson's trichrome staining, which visualizes collagen in blue, indicates the

formation of only a thin fibrotic capsule around the implants, suggesting excellent local tissue integration and biocompatibility. Scale bars, 100 μm .

Point-by-Point Responses to Reviewer #3:

Reviewer #3 (Remarks to the Author):

This manuscript presents the development of a decellularized extracellular matrix hydrogel incorporating T β 4-loaded PLGA microspheres. Notably, the study introduces an innovative approach by utilizing decidualized mouse uterus for the preparation of dECM hydrogel, demonstrating favorable regenerative efficacy. However, the scientific evidence remains insufficiently robust and the experimental design requires refinement. Given the current scope and depth of the work, I conclude that this study does not meet the standards for publication in Nature Communications and therefore recommend rejection. Detailed comments are provided below.

Dear Reviewer #3,

We are sincerely grateful for your thorough and critical evaluation of our manuscript. We fully accept your initial assessment that our original submission required more robust scientific evidence and experimental refinement to meet the exceptionally high standards of *Nature Communications*. Your rigorous critique provided a clear, albeit challenging, roadmap for the substantial improvements we needed to make.

We were, however, particularly encouraged that you recognized the core of our work as an **“innovative approach”** with **“favorable regenerative efficacy.”** This acknowledgment of our central concept motivated us to undertake the comprehensive revisions necessary to fully support our claims with the level of evidence this journal demands.

Taking your feedback to heart, we have performed substantial new experiments—including the gold-standard *in vivo* biocompatibility study and the addition of key molecular markers—and have conducted more rigorous quantitative analyses throughout the manuscript. We believe these comprehensive revisions have **transformed the manuscript**, directly resolving the concerns you raised and elevating the study to the high standard of scientific rigor we are confident you will now find compelling.

We have carefully considered all of your suggestions and have addressed them point-by-point below, outlining the specific actions taken. We thank you again for your invaluable contribution to improving our study and respectfully invite you to reappraise our substantially revised work.

Major remarks:

Reviewer #3, Major Comment #1:

Section 3.1 mentions that the dECM hydrogel exhibits thermosensitive gelation properties. Experimental validation of this property, including quantification of the sol-gel transition time, should be provided.

Response: We are sincerely grateful for this essential critique. The reviewer is absolutely correct that our original manuscript lacked direct, quantitative evidence for the hydrogel's gelation kinetics. To rectify this, we have now performed a comprehensive rheological analysis.

Our new data now demonstrates two critical and complementary properties. **First**, a new time-sweep analysis confirms the hydrogel's rapid thermosensitive gelation, with the sol-gel transition occurring at **approximately 52 seconds** at 37°C (**New Figure 4d**). **Second**, our frequency-sweep data (**Figure 4c**) characterizes the nature of the resulting scaffold. Crucially, the storage modulus ($G' \approx 1-1.3$ kPa) is **consistently and significantly higher than the loss modulus ($G'' \approx 0.6-0.9$ kPa)** across the entire frequency range. This unequivocally confirms the formation of a stable, solid-like gel. More importantly, this specific mechanical profile—a **soft, viscoelastic gel** rather than a rigid one—is a key design feature. It is well-established that hydrogels with mechanical properties that mimic the native target tissue (the endometrium being a soft tissue) provide a more favorable microenvironment for cell infiltration, remodeling, and regeneration. Thus, these complementary analyses now provide a complete and rigorous picture: our hydrogel not only gels rapidly at body temperature but also forms a **mechanically biomimetic and stable scaffold** ideally suited for endometrial repair. We are confident these additions fully resolve the reviewer's concern and substantially strengthen the scientific rationale for our material's design.

Action Taken:

- **Quantification of Gelation Kinetics (New Data):** We performed a time-sweep rheological analysis at 37°C. As shown in the new **Figure 4d**, the G'/G'' crossover point occurred at approximately 52 seconds, quantitatively confirming rapid gelation.
- **Characterization of Final Gel Properties (Re-contextualized Data):** We have re-analyzed our frequency-sweep data on the fully formed hydrogel (**Figure 4c**). The results clearly show that G' is consistently higher than G'' , confirming the formation of a stable gel. We have revised the text to accurately describe this as a **soft, viscoelastic gel** and have added discussion regarding the biological significance of this property for endometrial tissue engineering.
- **Manuscript Revision:** The Results and Methods sections have been substantially revised to incorporate these new findings and to clearly articulate the complementary nature of these two essential rheological analyses.

Revisions in the Manuscript (highlighted in blue):

1. In the Main Figures:

A new panel, Figure 4d, has been added, presenting the time-sweep rheological data. The legend for Figure 4 has been updated.

2. In the Results Section (under subsection 2.3):

“To assess its gelation kinetics at physiological temperature, time-sweep rheology was performed. This revealed a rapid sol-gel transition, with the crossover of the storage modulus (G') and loss modulus (G'') occurring at approximately 52 seconds (**Fig. 4c**), confirming its ability to form a stable scaffold quickly upon injection.”

Figure 4c, Time sweep rheology tracking the sol-gel transition at 37 °C, with the crossover point indicating gelation time.

3. In the Supplementary Materials and Methods 2.4 Section (under "Rheological Properties"):

- **Rheological Properties:** Rheological measurements were performed using a rotational rheometer (Anton Paar MCR 302, Austria) equipped with a parallel plate geometry (25 mm diameter, 1 mm gap).
 - ✧ Gelation Kinetics (Time-Sweep Analysis): To quantitatively determine the thermo-responsive gelation behavior, 500 μL of ice-cold pre-gel solution was loaded onto the lower plate pre-cooled to 4°C. The temperature was then rapidly increased to and maintained at 37°C. A time sweep was immediately conducted at a constant frequency (1 Hz) and constant strain (1%, within the linear viscoelastic region) to monitor the evolution of the storage modulus (G') and loss modulus (G'') over time. The gelation time was defined as the time at the crossover point where G' equaled G'' .
 - ✧ Mechanical Properties of the Final Gel (Frequency-Sweep Analysis): After the time-sweep analysis, once the G' and G'' moduli reached a stable plateau, a frequency sweep was performed at a constant strain (1%) over a frequency range of 0.1-10 Hz. This analysis was used to characterize the frequency-dependent viscoelastic properties of the fully formed hydrogel scaffold.

Reviewer #3, Major Comment #2:

The manuscript only presents the release curve of T β 4 from microspheres. Please also show the release behavior of T β 4 from the hydrogel.

Our Response:

We thank the reviewer for this crucial observation. The reviewer is absolutely correct that understanding the T β 4 release profile from the final composite hydrogel system is

essential for correlating our material's properties with its therapeutic efficacy. The release from the microspheres alone does not capture the full picture. We have therefore conducted a new, comprehensive *in vitro* release study to address this point directly, and the results have revealed a key design advantage of our system.

The results, now presented in **Figure 4j**, reveal a critical finding: the DEndo-UdECM matrix significantly **retards** the release of T β 4 from the embedded microspheres, creating a more sustained and linear release profile. Specifically, at 24 hours, only approximately **12%** of T β 4 was released from the composite hydrogel, compared to **~22%** from the microspheres alone. This trend continues, with the composite system consistently exhibiting a more gradual and prolonged release throughout the critical first week of regeneration.

This result directly demonstrates that the hydrogel does not act as a passive carrier but as an **active secondary diffusion barrier**. This is a highly desirable property for regenerative medicine, as it effectively **prevents a premature burst release**, ensuring a more sustained, therapeutically optimal delivery of T β 4 over the critical window of endometrial repair. We are sincerely grateful to the reviewer for pushing us to perform this experiment, as it has allowed us to uncover and highlight a key design feature that significantly strengthens the rationale for our composite system's success.

Action Taken:

- Conducted a new *in vitro* release study to characterize and compare the release kinetics of T β 4 from the microspheres alone versus from the complete DEndo-UdECM/T β 4@PLGA composite hydrogel.
- Generated a new data panel (**Figure 4j**) that clearly illustrates the release-modulating effect of the hydrogel matrix.
- Updated the **Results, Methods, and Figure 4 legend** to incorporate this new data and its powerful interpretation.

Revisions in the Manuscript (highlighted in blue):**1. In the Results Section 2.3:**

“Critically, the drug release profile was evaluated both for the microspheres alone and for the composite hydrogel system. While T β 4-loaded microspheres alone exhibited a burst release with nearly 70% released within 3 days, their incorporation into the DEndo-UdECM hydrogel resulted in a significantly more sustained and controlled release profile. This dual-component system effectively mitigated the initial burst and prolonged delivery over the 10-day period, ensuring prolonged local availability of the therapeutic peptide (Fig. 4j).”

2. In the Supplementary Materials and Methods Section 2.4 (under "Characterization "):

“In Vitro T β 4 Release from Composite Hydrogel: To measure T β 4 release from the complete system, T β 4@PLGA microspheres were first uniformly suspended in the ice-cold DEndo-UdECM pre-gel solution. Aliquots of 100 μ L of the composite pre-gel were then cast into a 96-well plate and allowed to gel at 37°C for 30 minutes. Following gelation, 200 μ L of PBS (pH 7.4) was added to each well. The plate was incubated at 37°C with gentle shaking. At predetermined time points, the entire supernatant was collected and replaced with fresh PBS. The concentration of T β 4 in the collected supernatant was quantified using a commercial ELISA kit according to the manufacturer's instructions. The cumulative release percentage was calculated based on the total encapsulated T β 4 content.”

3. In the Figure Legends:

Fig. 4j, Cumulative *in vitro* T β 4 release profiles from T β 4@PLGA microspheres alone (blue) and from the composite hydrogel (red). data are presented as mean \pm SD from n = 3 independent experiments.

Reviewer #3, Major Comment #3:

The in vitro and in vivo degradation properties of the composite hydrogel should be investigated.

Our Response:

We sincerely thank the reviewer for this insightful comment. We agree completely that the degradation profile is a critical attribute for any biomaterial designed for tissue regeneration.

To address this point with the scientific rigor it deserves, we have adopted a comprehensive, two-pronged approach in our revised manuscript:

1. **Robust Quantitative *In Vitro* Data as a Foundation:** We present our clear and quantitative *in vitro* degradation data in **Figure S6c**. This analysis shows a controlled, steady degradation profile that is highly congruous with the physiological timeframe required for endometrial healing (approx. 1-2 weeks). This provides a strong, quantitative basis for the material's intrinsic stability and its suitability for the intended application.
2. ***In Vivo* Functional Outcomes as the Most Compelling Evidence:** We argue that for a complex, bioactive scaffold like our dECM hydrogel, the most compelling and functionally relevant evidence of an appropriate *in vivo* degradation profile is its demonstrated therapeutic success.
 - We have explicitly addressed the significant technical challenges and risks of artifact associated with direct fluorescent labeling of a complex dECM scaffold, where such modifications could alter its native bioactivity or lead to misleading results.
 - Instead, we present a strong mechanistic argument: a hydrogel that degrades too quickly would fail to provide a supportive scaffold, while one that persists too long would impede tissue integration. The fact that we observed robust endometrial regeneration, neovascularization, and ultimately, the complete restoration of fertility (**Figures 6, 7**), serves as

powerful, albeit indirect, evidence that the hydrogel persists for the "correct" duration *in vivo* before being safely resorbed.

We believe this combination of direct quantitative *in vitro* data and a strong, function-based argument for its *in vivo* behavior provides a more scientifically insightful and relevant assessment of our material's degradation profile than a potentially artifact-prone labeling study.

Action Taken:

- A comprehensive paragraph has been added to the **Discussion section** of the manuscript.
 - This new text explicitly links the quantitative *in vitro* degradation profile (**Figure S6c**) to the physiological requirements for endometrial repair.
 - It transparently discusses the technical rationale for not performing direct *in vivo* labeling of a complex, bioactive dECM scaffold, highlighting the risk of altering its therapeutic properties.
 - It formally presents the argument that the robust therapeutic outcomes observed in our study serve as the most powerful evidence for the hydrogel's appropriate *in vivo* residence time and degradation rate.
-

Revisions in the Manuscript (highlighted in blue):

1. In the Supplementary Information:

Figure S6c Comparative *in vitro* degradation profiles of hydrogels derived from non-decidualized (CEndo-UdECM) and decidualized (DEndo-UdECM) uterine ECM over 14 days. Data are presented as mean \pm s.d. ($n = 3$ biologically independent samples).

2. In the Discussion section 3.1, the following paragraph has been added:

“This design provided a controlled release of Tβ4 over approximately 10 days, a profile deliberately synchronized with the hydrogel's 14-day degradation window (Fig. 4j, Fig. S6c) to ensure therapeutic action throughout the critical phase of tissue remodeling.

While direct *in vivo* tracking via fluorescent labeling is a common evaluation method, we recognized the significant risk that covalently modifying our complex, bioactive dECM scaffold could alter its native therapeutic properties or introduce labeling artifacts. Therefore, we contend that the most compelling evidence for the hydrogel's appropriate *in vivo* residence time is its demonstrated therapeutic efficacy. The significant improvements in endometrial regeneration, neo-vascularization, and ultimately, the restoration of fertility (figures 6, 7, 9), would be mechanistically implausible if the scaffold's degradation profile were unsuitable. A scaffold that degrades too rapidly would fail to guide tissue repair, while one that persists too long would impede tissue integration and likely elicit a chronic inflammatory response. Thus, the robust functional success observed *in vivo* strongly corroborates our *in vitro* findings, confirming that the hydrogel possesses a degradation rate finely tuned for its intended therapeutic application.”

Reviewer #3, Major Comment #4:

Figure 5, the authors only evaluated in vitro regeneration effects of CEndo-UdECM and DEndo-UdECM. Since the introduction highlights Tβ4's role in enhancing tissue repair and promoting angiogenesis, DEndo-UdECM/Tβ4@PLGA group should be added

Our Response:

We thank the reviewer for this excellent and insightful question, which touches upon the core rationale of our composite hydrogel's design. We agree that, on the surface, including a Tβ4 group in the *in vitro* regeneration assays seems logical. However, our decision to omit it was a **deliberate and scientifically-driven choice** based on the specific roles assigned to each component of our system and the temporal limitations

of the assays themselves. We appreciate this opportunity to clarify our experimental strategy.

Our rationale is twofold:

1. Mismatch in Functional Timescales: Short-Term Assay vs. Long-Term Release:

- The *in vitro* regeneration assays shown in **Figure 5** (cell proliferation, migration, and tube formation) are inherently **short-term experiments**, typically conducted over **6–24 hours**. This timeframe is dictated by the rapid dynamics of these cellular processes *in vitro*.
- In contrast, our T β 4 delivery system (T β 4@PLGA microspheres) is explicitly designed for **long-term, sustained release over approximately 10 days** (as demonstrated in our release studies, **Figure 4j**). In the first 24 hours, only a small fraction of the total T β 4 payload is released.
- Therefore, adding the DEndo-UdECM/T β 4@PLGA group to these short-term assays would not accurately reflect its intended therapeutic function *in vivo*. The minimal amount of T β 4 released in such a short window would likely yield negligible or misleading results, failing to capture the true potential of the sustained delivery system.

2. Strategic "Division of Labor" and Targeted Validation:

- Our central hypothesis is that uterine regeneration requires a synergistic, dual-pronged approach. We designed our composite hydrogel based on a strategic “**division of labor**”:
 - **The DEndo-UdECM hydrogel acts as the primary pro-regenerative and pro-angiogenic scaffold.** Its function is to provide the immediate, niche-mimicking signals to promote rapid cell recruitment, proliferation, and vascular network formation. We confirmed this role in the short-term assays in **Figure 5**.
 - **The T β 4@PLGA microspheres act as the long-term anti-fibrotic modulator.** While T β 4 has pleiotropic effects, its most critical role in the context of severe IUA is to suppress the chronic, pro-fibrotic signaling cascades (like TGF- β 1/Smad) that impede functional repair over weeks.

- To specifically validate this designated anti-fibrotic role of T β 4, we conducted a dedicated and **comprehensive anti-fibrosis assay**. As now presented in the expanded **Fig. S2**, we demonstrate that T β 4 potently suppresses TGF- β 1-induced upregulation of a wide array of key fibrotic markers, including the myofibroblast marker **α -SMA** and multiple extracellular matrix proteins (**COL1A1, Fibronectin, COL4, LAMININ**).
-

Action Taken:

In summary, our experimental design intentionally decouples the assessment of the two key components to reflect their distinct primary functions and temporal windows of action. The DEndo-UdECM provides the “Go” signal for regeneration, while T β 4 provides the “Stop” signal for fibrosis.

To make this rationale clearer in the manuscript, we have revised the Introduction, Results, and Discussion sections to more explicitly state this “division of labor” hypothesis and the specific, complementary roles of the DEndo-UdECM and T β 4. We now more clearly articulate that the synergy arises from the combination of a rapid, pro-regenerative scaffold with a long-term, anti-fibrotic agent, which together create a favorable environment for functional uterine repair. We are grateful to the reviewer for prompting us to elaborate on this critical aspect of our work.

Revisions in the Manuscript (highlighted in blue):

1. In the Introduction Section:

When introducing the composite hydrogel, a sentence has been added to clarify its design principle: " **This creates a dual-action system designed for a synergistic 'division of labor': the DEndo-UdECM scaffold provides the immediate pro-regenerative blueprint, while T β 4@PLGA acts as a sustained brake on fibrosis.** "

2. In the Figure and legend Section

Fig. S2 | Tβ4 suppresses TGF-β1-induced fibrotic activation in human endometrial stromal cells (hESCs) *in vitro*.

Human endometrial stromal cells were stimulated with TGF-β1 (10 ng/mL) for 48 h to induce a fibrotic phenotype, with or without co-treatment with Tβ4. **a**, Representative immunofluorescence images co-staining for the myofibroblast marker α-Smooth Muscle Actin (αSMA; green) and the key fibrosis marker Collagen Type I Alpha 1 (COL1A1; red). **b, c**, Quantification of the mean fluorescence intensity for αSMA (**b**) and COL1A1 (**c**). **d, f, h**, Representative immunofluorescence images for additional extracellular matrix (ECM) proteins: Fibronectin (FN1; red) (**d**), Collagen Type IV (COL4; red) (**f**), and Laminin (LN; red) (**h**). In these panels, the actin cytoskeleton is visualized with phalloidin (F-actin; green). **e, g, i**, Corresponding quantification of mean fluorescence intensity for FN1 (**e**), COL4 (**g**), and Laminin (**i**). **j**, To provide orthogonal validation, protein levels of key fibrotic markers were assessed by western blot. Representative western blot bands for COL1A1 and α-SMA are shown, with GAPDH serving as a loading control. **k, l**, Corresponding densitometric quantification for COL1A1 (**k**) and α-SMA (**l**). In all immunofluorescence images, cell nuclei are counterstained with DAPI (blue). Scale bars, 100 μm. In all quantitative plots, data are presented as scatter plots with mean ± s.d. from n = 6 biologically independent experiments. Statistical significance was determined by one-way ANOVA with Tukey's post hoc test; **P* < 0.05, ***P* < 0.01, ****P* < 0.001 vs. the TGF-β1 group.

3. In the Discussion Section 3.2:

A paragraph has been added to further elaborate on this strategy:

"3.2 A Strategic 'Division of Labor': Validating the Dual-Action Mechanism *In Vitro*

“A key strength of our therapeutic design is the strategic 'division of labor' between the hydrogel scaffold and the drug delivery microspheres, and our in vitro assays were meticulously designed to validate these distinct roles.

First, we confirmed the potent, immediate pro-regenerative capacity of the DEndo-UdECM scaffold itself—the system's 'pro-regenerative engine.' As hypothesized, DEndo-UdECM significantly outperformed CEndo-UdECM in promoting endometrial epithelial cell proliferation, stromal cell migration, and endothelial cell (HUVEC) tube formation (Fig. 5). At first glance, these potent pro-regenerative effects might seem paradoxical, given the terminally differentiated nature of the mature decidua. However, this apparent conflict is resolved when decidualization is viewed not as a static endpoint, but as a dynamic, multi-phase process fundamentally analogous to controlled, scar-free wound healing [48, 50]. Authoritative reviews highlight that the initial phase of decidualization is inherently proliferative and involves extensive tissue remodeling, with stromal cells first undergoing “extensive proliferation and differentiation” to build the necessary architecture[48]. Therefore, our findings do not contradict the biomimetic premise; on the contrary, they directly confirm that DEndo-UdECM successfully recapitulates the crucial, pro-regenerative signals of this early decidualization process, making it an ideal 'engine' for therapeutic wound repair[51]. Conversely, we validated the critical role of T β 4 as the system's long-term 'anti-fibrotic brake.' In a dedicated assay mimicking a sustained pro-fibrotic challenge, the controlled release of T β 4 effectively suppressed TGF- β 1-induced fibrotic activation in endometrial stromal cells in vitro (Fig. S2), consistent with its well-established anti-fibrotic functions[52]. This strategic design, combining a rapid pro-regenerative 'engine' with a sustained anti-fibrotic 'brake,' provides a clear mechanistic rationale for the profound synergistic effects observed in our in vivo model, where both rapid healing and prevention of chronic scarring are essential for functional recovery.”

Reviewer #3, Major Comment #5:

A large number of immunofluorescence and histological experiments were used in in vitro and in vivo experiments to observe the phenotype. Please add Western blotting experiments for in vitro experiments to increase credibility

Our Response:

We sincerely thank the reviewer for this excellent and constructive suggestion. We completely agree that supplementing our imaging-based phenotypic analysis with robust, quantitative biochemical data is crucial for increasing the credibility and mechanistic depth of our findings.

Acting on this valuable advice, we prioritized providing direct biochemical validation for the **most critical molecular mechanism** underpinning our therapeutic strategy: the anti-fibrotic efficacy of T β 4. While our manuscript presents multiple *in vitro* phenotypes, we reasoned that the suppression of key fibrotic protein expression is the cornerstone of our hydrogel's function in combating IUA pathology. Therefore, we focused our efforts on this key aspect to provide the most impactful new data.

Action Taken:

- **New Experiment Performed: Biochemical Validation of Anti-Fibrotic Efficacy:** We have performed a new Western Blotting experiment to validate the anti-fibrotic bioactivity of T β 4 at the molecular level. We revisited our TGF- β 1-induced fibroblast model (as presented in Supplementary Figure S2) and quantified the expression of the two canonical fibrosis markers: **α -smooth muscle actin (α -SMA)**, a marker of myofibroblast activation, and **Collagen Type I Alpha 1 (COL1A1)**, a primary component of scar tissue.
- **New Figure Panel Added:** The results, including representative blots and full quantification, are now presented in the new **Figure S2j-1**. As can be clearly seen from the provided image, this Western Blot unequivocally demonstrates that T β 4 treatment **markedly suppresses** the TGF- β 1-induced upregulation of both α -SMA and COL1A1 protein levels.
- **Strengthened Conclusion:** This new biochemical evidence provides powerful, quantitative corroboration for our immunofluorescence findings (Figure S2a-c).

It moves beyond phenotypic observation to offer **irrefutable molecular proof** of T β 4's mechanism of action in mitigating the core pathological drivers of fibrosis.

In summary, by supplementing our study with this targeted Western Blot analysis, we have directly addressed the reviewer's concern and provided robust, quantitative support for our key *in vitro* claims. We have incorporated this new figure and the corresponding descriptions into the revised manuscript. We are very grateful to the reviewer for prompting us to significantly strengthen the mechanistic foundation of our work.

Revisions in the Manuscript (highlighted in blue):

1. In the Results Section (describing the *in vitro* anti-fibrotic effects of T β 4):

“...To provide quantitative biochemical validation for these phenotypic observations, we performed Western Blot analysis. The results confirmed that T β 4 treatment significantly suppressed the TGF- β 1-induced upregulation of both α -SMA and COL1A1 protein levels (Fig. S2j-l), offering direct molecular evidence of its anti-fibrotic activity.”

2. In the Supplementary Figures:

Fig. S2 | T β 4 suppresses TGF- β 1-induced fibrotic activation in human endometrial stromal cells (hESCs) *in vitro*.

j, To provide orthogonal validation, protein levels of key fibrotic markers were assessed by western blot. Representative western blot bands for COL1A1 and α -SMA are shown, with GAPDH serving as a loading control. **k**, **l**, Corresponding densitometric quantification for COL1A1 (**k**) and α -SMA (**l**). In all quantitative plots, data are presented as scatter plots with mean \pm s.d. from $n = 6$ biologically independent experiments. Statistical significance was determined by one-way ANOVA with Tukey's post hoc test; * $P < 0.05$, ** $P < 0.01$, *** $P < 0.001$ vs. the TGF- β 1 group.

Reviewer #3, Major Comment #6:

The in vitro anti-fibrosis experiments are insufficient. Additional studies are needed to evaluate the overall effect of the hydrogel.

Our Response:

We sincerely thank the reviewer for this exceptionally insightful and critical point. It touches upon the very core of our therapeutic strategy and has prompted us to more clearly articulate the rationale behind our *in vitro* experimental design in the revised manuscript.

The reviewer correctly observes that our *in vitro* anti-fibrosis assay (Figure S2) was designed to specifically validate the efficacy of the therapeutic agent (T β 4), rather than the composite hydrogel system. We respectfully submit that this was a **deliberate and strategic choice**, rooted in a “problem-oriented” approach to therapeutic development. We believe that testing the composite hydrogel *in vitro* for anti-fibrosis would introduce confounding variables, leading to results that are difficult to interpret cleanly.

Our Rationale:

- **1. Foundational Principle: Isolate and Validate the Active Pharmaceutical Ingredient (API).** In any combination therapy, it is a foundational principle to first demonstrate that the API (in our case, T β 4) is effective against the specific pathological process it is intended to treat (TGF- β 1-induced fibrosis). Our Figure S2, now strengthened with Western Blot data (Fig. S2j-l), achieves precisely this. It provides clear, direct evidence that T β 4 potently counteracts the fibrotic cascade at the cellular level, justifying its inclusion in the delivery system.
- **2. Scientific Rigor: Avoiding Confounding Variables and Misinterpretation.** The DEndo-UdECM is a complex biological scaffold, rich in a multitude of pro-regenerative and pro-angiogenic factors (e.g., FGF, VEGF, Collagens). Its primary role is to create a pro-regenerative microenvironment, as demonstrated in our experiments with endometrial cells and HUVECs (Figure 5). Testing this pro-regenerative scaffold directly in a high-dose, pro-

fibrotic stimulation model *in vitro* would create a “push-pull” scenario. The pro-regenerative signals from the ECM could mask, interfere with, or even paradoxically interact with the specific anti-fibrotic signal of T β 4. The resulting data would be an un-deconvolvable mixture of two different biological processes, making clean mechanistic interpretation impossible.

- **3. Biological Relevance: The In Vivo Arena is the True Test for the Composite System.** The ultimate validation for a composite therapeutic system lies in the *in vivo* setting, where all complex biological processes—regeneration, angiogenesis, inflammation, and fibrosis—occur simultaneously. Our extensive *in vivo* data (Figures 6, 7, 8, 9, 10, 11) is precisely the experiment that evaluates the “overall effect of the hydrogel.” These results demonstrate unequivocally that when the **pro-regenerative scaffold (DEndo-UdECM)** and the **anti-fibrotic agent (T β 4)** are combined in the complex milieu of a damaged uterus, they work in synergy to produce superior functional outcomes. The *in vivo* success is the ultimate proof that our design works as intended.

In summary, we respectfully argue that our *in vitro* study design was not an oversight, but a deliberate strategy to isolate and validate the function of each component before demonstrating their combined power in the most relevant setting—the living organism.

Action Taken:

While we stand firmly by our experimental design for the reasons outlined above, we recognize that our initial manuscript did not sufficiently explain the logic behind it. In response to the reviewer’s valuable feedback, we have made the following revisions to enhance clarity.

Revisions in the Manuscript (highlighted in blue):

1. In the Introduction:

“This creates a dual-action system designed for a synergistic 'division of labor': the DEndo-UdECM scaffold provides the immediate pro-regenerative blueprint, while T β 4@PLGA acts as a sustained brake on fibrosis

2. In the Discussion Section (specifically in the newly revised section 3.2):

“A key strength of our therapeutic design is the strategic 'division of labor' between the hydrogel scaffold and the drug delivery microspheres, and our in vitro assays were meticulously designed to validate these distinct roles. ”

“First, we confirmed the potent, immediate pro-regenerative capacity of the DEndo-UdECM scaffold itself—the system's 'pro-regenerative engine.' As hypothesized, DEndo-UdECM significantly outperformed CEndo-UdECM in promoting endometrial epithelial cell proliferation, stromal cell migration, and endothelial cell (HUVEC) tube formation (Fig. 5).”

“Conversely, we validated the critical role of T β 4 as the system's long-term 'anti-fibrotic brake.' In a dedicated assay mimicking a sustained pro-fibrotic challenge, the controlled release of T β 4 effectively suppressed TGF- β 1-induced fibrotic activation in endometrial stromal cells in vitro (Fig. S2), consistent with its well-established anti-fibrotic functions[52]. This strategic design, combining a rapid pro-regenerative 'engine' with a sustained anti-fibrotic 'brake,' provides a clear mechanistic rationale for the profound synergistic effects observed in our in vivo model, where both rapid healing and prevention of chronic scarring are essential for functional recovery.”

We trust that this detailed explanation, coupled with the substantial revisions to the manuscript, clarifies our rigorous scientific strategy. We are grateful to the reviewer for pushing us to make this crucial aspect of our work more transparent and impactful.

Reviewer #3, Major Comment #7:

All immunofluorescence images suffer from poor clarity and overexposure. Please improve the image quality. Non-specific fluorescence signals are evident in multiple immunofluorescence images, particularly in Figures 9A, 10A, and 11H.

Our Response:

We sincerely thank the reviewer for their meticulous eye and for raising this absolutely critical point about image quality. We have conducted a rigorous re-evaluation of all our immunofluorescence data presentation and agree that the images in the previous version did not meet the high standards required. The reviewer's feedback has prompted

us to undertake a comprehensive overhaul of our image processing and presentation workflow.

Upon investigating the root cause, we identified that a significant contributor to the perceived issues was an unfortunate technical artifact introduced during image processing and file compression for manuscript preparation. This process led to a loss of fine detail, artificially brightened signals that appeared overexposed, and a muddling of the background.

Action Taken:

We have taken immediate and comprehensive action to rectify this. Our goal was not to artificially "beautify" the images, but to present the original data with the highest possible fidelity.

- **1. Return to Source Data:** We have gone back to the original, high-resolution, uncompressed raw data files (e.g., .tiff, .czi) for every immunofluorescence experiment presented in the manuscript.
- **2. Optimized Re-processing:** We have meticulously re-processed and re-exported every image using a lossless format (TIFF). This involved careful, standardized adjustments to brightness and contrast *across all groups within an experiment* to ensure the specific signal is clearly visible without saturating the detector or altering the underlying data integrity.
- **3. Comprehensive Image Replacement:** As a result, we have **replaced ALL immunofluorescence images** throughout the manuscript—including but not limited to the highlighted Figures 9, 10, and 11—with these substantially improved versions. We believe the revised images now offer a much clearer and more faithful representation of our findings.

Clarification on Signal Specificity (Addressing Non-Specific Signals):

The reviewer's point about non-specific signals is particularly insightful, and we wish to address it with full transparency.

- In our re-evaluation, we focused on distinguishing true non-specific staining from compression artifacts. In the revised images, the signal-to-noise ratio is markedly improved.

- Regarding the specific example of ALDH1A1 staining (now Figure 9a), we acknowledge that upon close inspection, some low-level background puncta may still be visible in the stromal compartment. **However, this is where our biological understanding becomes critical.** The specific and biologically relevant signal for ALDH1A1 is known to be strongly and cleanly localized to the **glandular epithelium**. As the revised Figure 9a clearly shows, the signal in the glands is intense and specific, while the stromal background puncta are sporadic and do not interfere with the clear identification, interpretation, or quantification of the positive signal in the target tissue structure.
- This principle of contextual interpretation applies to our other images as well. Our re-processing aimed to present the specific signal as clearly as possible while honestly representing the original data, rather than digitally manipulating the images to artificially remove every trace of background.

In summary, while we have made significant efforts to improve the technical presentation of our images, we also stand by our scientific interpretation, which is always based on the specific signal within its known biological context.

Revisions in the Manuscript (highlighted in blue):

In the Main Figures and Supplementary Figures:

All immunofluorescence images throughout the manuscript and its supplement have been replaced with new, higher-quality versions derived from original raw data files to enhance clarity and more accurately represent the experimental results.

We trust that these substantial efforts, combined with our transparent explanation, have led to a marked improvement in image quality that now meets the reviewer's standards.

We believe the revised images, while honestly reflecting the realities of biological imaging, now provide a much clearer and more reliable foundation for our conclusions.

We are deeply grateful for this critical feedback, as it has pushed us to significantly elevate the presentation and professionalism of our study.

Reviewer #3, Major Comment #8:

As the manuscript described, the established animal model is in fact an endometrium injured-model, other than an IUA model which should present thin endometrium and severe fibrosis.

Response: We sincerely thank the reviewer for raising this critical and highly relevant point regarding the classification and validity of our animal model. This question touches upon the very foundation of our in vivo investigation, and we appreciate the opportunity to provide a more thorough and explicit justification in the revised manuscript.

The reviewer is entirely correct in stating that a clinically relevant model of Intrauterine Adhesions (IUA) must faithfully recapitulate the key pathological hallmarks of the human condition, which are primarily characterized by a **thinned, atrophic endometrium** and **excessive, pathological fibrosis**. We completely agree with this rigorous standard.

We respectfully posit that our model, induced by intrauterine ethanol infusion, does indeed robustly meet these criteria, making it a valid and appropriate model for studying IUA. Our justification is grounded in both our own comprehensive data and the established consensus in the field.

- **1. Evidence of IUA Pathological Hallmarks in Our Data:**

Our results clearly demonstrate the successful establishment of both key IUA features in the “Model” group:

- **Endometrial Thinning:** Our morphometric analysis, presented in **Figure 6E**, shows a statistically significant reduction in endometrial thickness in the Model group compared to the sham-operated animals. This directly confirms the atrophic phenotype characteristic of IUA.
- **Severe Fibrosis:** We provided extensive, multi-faceted evidence of severe fibrosis. The Masson’s trichrome staining in **Figure 7A** vividly illustrates massive collagen deposition (dense blue staining) and near-total luminal obliteration, which is the anatomical definition of an adhesion. This qualitative observation is quantitatively supported by the significant upregulation of key fibrosis markers, including Collagen I

and α -SMA, in both our IHC and immunofluorescence analyses (Figures 7B-E and 11H).

Taken together, our own data provides compelling evidence that our model is far more than a simple “injury model”; it is a model of chronic, pathological repair that results in a phenotype fully consistent with clinical IUA.

- **2. Validation by Established, High-Impact Literature:**

Crucially, the ethanol infusion method is a well-established and widely accepted protocol for inducing a severe IUA phenotype in rodents. This model has been utilized and validated in numerous high-impact studies published in top-tier journals, where it is consistently referred to as an “IUA model.” For instance:

- *e.g., Qi et al., **Biomaterials**, 2024, 309, 122615.*
- *e.g., Cao et al., **Advanced Science**, 2024, 20(11), e2306507.*
- *e.g., Jungho et al., **Advanced Functional Materials**, 2023, 34(33), 2214291.*
- *e.g., Yi et al., **Advanced Materials**, 2022, 8(34), e2106510.*

The consistent use and validation of this model in the leading literature underscores its robustness and relevance for investigating IUA pathophysiology and evaluating novel therapeutics like ours.

Action Taken:

While we are confident in the model’s validity, we acknowledge the reviewer’s point that our initial description could have been more precise and better justified. To address this, we have expanded the Methods section to include a brief statement justifying the choice of this model, explicitly mentioning that it recapitulates the key features of endometrial thinning and fibrosis, and citing the key validation literature.

Revisions in the Manuscript (highlighted in blue):

1. In the Methods Section (Sub-heading: *IUA Model Establishment and Treatment Groups*):

To address the reviewer's valuable feedback and enhance clarity, we have revised the description in the Methods section to more explicitly justify our model selection. The updated text now reads:

“Ethanol-Induced IUA Model Establishment and Treatment Groups: A severe murine unilateral model of Intrauterine Adhesions (IUA) was established in the left uterine horn via intrauterine infusion of 95% ethanol. This method is a widely validated approach known to faithfully recapitulate the key pathological hallmarks of clinical IUA, including significant endometrial thinning and severe fibrosis [10, 21, 68, 69]. Following our previously published methodology[10], mice were subsequently randomly assigned to seven groups (n=24/group): Sham, Model (PBS control), Tβ4@PLGA, CEndo-UdECM, DEndo-UdECM, CEndo-UdECM/Tβ4@PLGA, and DEndo-UdECM/Tβ4@PLGA. Each treatment group received a single intrauterine administration of the corresponding therapeutic material.”

We hope this detailed justification, supported by our own data and the weight of existing literature, fully addresses the reviewer's concerns. We are grateful for the opportunity to clarify this crucial aspect of our study, which has undoubtedly strengthened the manuscript.

Reviewer #3, Minor Comment #1:

Caption of Figure 2, “Characterization of UdECM scaffolds before and after decidualization”, herein “decidualization” should be “decellularization”.

Response: We sincerely thank the reviewer for catching this embarrassing typographical error. We have corrected “decidualization” to “**decellularization**” in the caption of Figure 2. We apologize for this oversight and appreciate the reviewer's careful reading.

Reviewer #3, Minor Comment #2:

Figure 11A, please add the number of samples in each group on the western blot bands.

Response: We thank the reviewer for this excellent suggestion to improve clarity. We have revised Figure 11A (now as **figure 10b**) to clearly indicate the number of

biological replicates. The new figure now displays the individual bands for each of the three independent biological replicates (n=6) for every group, making the data presentation more transparent and robust.

Reviewer #3, Minor Comment #3:

Figure 11A, please add western blot bands of Total Smad3, and Fibrosis related proteins (e.g. FN, COL1A1...)

Our Response: We are extremely grateful for this critical and valuable suggestion. The reviewer is absolutely correct that adding Western blots for Total Smad3 and downstream fibrosis-related proteins would provide much stronger and more quantitative evidence for our proposed anti-fibrotic mechanism. We immediately acted upon this excellent recommendation.

Action Taken:

We have performed new Western blot experiments using uterine tissue lysates harvested from all treatment groups. In direct response to the reviewer's request, we have now included data for:

- **1. Total Smad3:** This new blot confirms that the observed decrease in p-Smad3 is indeed due to reduced phosphorylation rather than a change in the total protein level. This critically validates the use of the p-Smad3/Smad3 ratio as a reliable measure of signaling pathway activation.
- **2. Key Fibrosis Markers (α -SMA and COL1A1):** We chose to measure α -SMA and Collagen I (COL1A1) as they are canonical markers of fibrosis. These new blots provide robust, quantitative molecular validation for the anti-fibrotic effects that were previously demonstrated through our histological (Masson's trichrome) and immunofluorescence analyses.

These additions have significantly enhanced the mechanistic depth and rigor of our study.

Revisions in the Manuscript (highlighted in blue):

1. In the Main Figures:

The new Western blot results have been incorporated into the manuscript. Per the reviewer's suggestion regarding the data originally in Figure 11A, these have been integrated into the revised Figure 10 (now presented as Figure 10b) for a more cohesive presentation of the molecular data. The corresponding quantification has also been added. The newly incorporated Figure 10b, showcasing the Western blot analysis for Total Smad3, α -SMA, and COL1A1 alongside other relevant proteins.

2. In the Main Text and Figure Legends:

The Results section and the legend for Figure 10 have been substantially updated to describe and interpret these new, definitive Western blot findings.

Fig. 10 | The DEndo-UdECM/T β 4@PLGA hydrogel simultaneously suppresses pyroptosis and canonical TGF- β /Smad signaling.

b - k, Western blot analysis of key signaling proteins in endometrial tissue lysates 14 days post-treatment. b, Representative western blot images. c - g, Quantification of proteins involved in the pyroptosis pathway: Gasdermin D (GSDMD; c), its N-terminal fragment (GSDMD-NT; d), pro-caspase-1 (e), and the mature inflammatory cytokines IL-1 β (f) and IL-18 (g). h - k, Quantification of proteins involved in the fibrosis pathway: α -smooth muscle actin (α -SMA; h), the ratio of phosphorylated Smad3 (P-Smad3) to total Smad3 (i), Collagen Type I Alpha 1 (COL1A1; j), and TGF- β 1 (k), data are presented as scatter dot plots from n = 6 biologically independent animals, with data shown as mean \pm s.d.

Statistical significance was determined by one-way ANOVA with Tukey' s post hoc test; *P < 0.05, **P < 0.01, ***P < 0.001.

Reviewer #3, Minor Comment #3:

In figure. 5B, C, D, the PI/AM staining results of the gelatin hydrogel were not displayed.

Response: We sincerely thank the reviewer for pointing out this important omission. A proper control group is indeed essential for a valid comparison, and we apologize for this oversight. We have now performed the requested experiments and have updated the figures to include the Gelatin hydrogel control group.

Action Taken:

We have performed the live/dead (Calcein-AM/PI) staining for cells cultured within the Gelatin hydrogel, serving as a baseline biocompatibility control.

- The new representative images for the Gelatin group have been incorporated into the revised **Figure 5c, d**.

Revisions in the Manuscript:

The revised Figure 5 now includes the essential Gelatin control group for all live/dead staining assays. The updated results confirm high cell viability across all hydrogel groups, as expected. A representative panel from the revised figure is shown below:

Fig. 5 | DEndo-UdECM hydrogel promotes pro-regenerative cellular responses *in vitro*. **c, d,** Biocompatibility assessment using Calcein-AM (live cells, green) and Propidium Iodide (PI; dead cells, red) staining of Ishikawa cells. **c,** Representative fluorescence images. **d,** Quantification of live and dead cell percentages. Scale bars, 100 μ m. Data are presented as mean \pm SD from $n = 3$ biologically independent samples. Statistical significance was determined by one-way ANOVA with Tukey's post hoc test; * $P < 0.05$, ** $P < 0.01$, *** $P < 0.001$.

We appreciate the opportunity to make this necessary correction, which strengthens the validity of our *in vitro* results.

Point-by-Point Responses to Reviewer #4:

Reviewer #4 (Remarks to the Author):

Response: We would like to extend our sincere thanks to you for your participation in the review process as a co-reviewer.

We highly commend the *Nature Communications* initiative to train and recognize the valuable contributions of Early Career Researchers. We are confident that your perspective, integrated within the primary reviewer's report, has contributed to the comprehensive and constructive feedback we received. This feedback has been invaluable in helping us to significantly improve the quality and clarity of our manuscript.

Thank you again for your time and effort.

Response to Reviewers for Manuscript ID NCOMMS-25-36427A

Title: **Decidualization-Empowered ECM Hydrogel integrating sustained T β 4 release Drives Endometrial Regeneration in Intrauterine Adhesions**

Authors: Yuxiang Liang^{1,2}, Zhaowei Yu¹, Shaobo Du¹, Yuqian Guo¹, Jing Li¹, Yujia Yan¹, Shanshan Jin¹, Wenjing Liang¹, Mengyuan Li¹, Jiao Yang¹, Zhiwei Peng¹, Zhaoyang Chen², Zhizhen Liu^{*1}, Qizhi Shuai^{*1}, Liping Li^{*1}, Jun Xie^{*1}

*In this response letter, all revisions made to the manuscript text are highlighted in **blue** for clarity.*

Point-by-Point Responses to Reviewer #1:

Reviewer #1 (Remarks to the Author):

Overall, the revised manuscript presents a coherent and mechanistically substantiated narrative that advances the study from a phenomenological report to a conceptually unified demonstration of a “decidualization-empowered” bioactive matrix. The work now offers clear scientific novelty, methodological rigor, and tangible translational relevance in uterine regenerative medicine. Minor refinements—such as clarifying methodological details related to ECM batch variability—could further enhance reproducibility, but no substantive deficiencies remain.

Dear Reviewer #1,

We are profoundly grateful to the reviewer for their overwhelmingly positive assessment and for their invaluable guidance throughout the entire review process. Their insightful critiques were instrumental in transforming our manuscript into the "coherent and mechanistically substantiated narrative" they so generously describe. We are thrilled that they recognize the study's novelty, rigor, and translational relevance.

The reviewer's final suggestion to address ECM batch variability is exceptionally insightful and touches upon a critical aspect of translating natural biomaterials. We agree completely that making this point explicit will further strengthen the manuscript.

Action Taken:

Following the reviewer's excellent advice, we have added a sentence to the **"Limitations and Future Perspectives"** section of our Discussion. This addition explicitly acknowledges batch variability as a known consideration and briefly describes our strategy to mitigate it, thereby enhancing the manuscript's transparency and rigor.

The revised text now includes the following statement:

"... Finally, while our hydrogel has demonstrated potent efficacy, we acknowledge a common challenge for all natural biomaterials: potential batch-to-batch variability [44]. In anticipation of this, we established a rigorous, standardized production and quality control (QC) protocol. This protocol, including standardized assessments of final protein concentration, residual DNA content, and rheological properties, was implemented to minimize variance and ensure the reproducibility of our findings. Such stringent QC represents a foundational step towards future clinical translation."

We once again thank the reviewer for their meticulous review and constructive feedback, which have been indispensable in elevating our work to its current standard.

Point-by-Point Responses to Reviewer #2:

Reviewer #2 (Remarks to the Author):

The authors have revised some figures and text in response to the previous review and have addressed a few of the questions raised. However, many critical issues remain unresolved, and several responses contain inaccuracies or appear to downplay key concerns. Therefore, I recommend major revision or rejection of the manuscript.

Dear Reviewer #2

We are sincerely grateful to you for your meticulous review and for providing such insightful and constructive feedback. We are particularly encouraged by your positive assessment of our study as **"conceptually innovative"** and your recognition that our **"biological outcomes are thoroughly validated."**

You have precisely identified the areas where the manuscript required significant improvement. We completely agree that robust **materials characterization, enhanced transparency in experimental design, and higher-quality figure presentation** are

absolutely essential for ensuring the "**reproducibility and mechanistic clarity**" that a study of this nature demands.

Guided by your expert recommendations, we have undertaken a comprehensive revision. This involved not only conducting several new experiments to bolster our materials data but also meticulously rewriting relevant methods sections and systematically improving all figures to meet the highest standards of clarity. We have addressed every point you raised in detail below, and we are confident that these substantial revisions have transformed the manuscript into a much stronger, more rigorous, and more reproducible piece of work.

Reviewer #2, Major Comment #1:

Fig. 4i: The image does not clearly demonstrate a homogeneous distribution of T β 4@PLGA microspheres; local aggregation appears evident. Fluorescence labeling combined with confocal microscopy is recommended to more accurately verify the spatial distribution of the microspheres within the hydrogel.

Response:

We are sincerely grateful to the reviewer for this exceptionally insightful and constructive suggestion. The reviewer is absolutely correct that a 2D bright-field image is insufficient to definitively assess 3D spatial distribution, and their recommendation to use confocal microscopy was the ideal solution.

Following the reviewer's excellent advice, we have performed a comprehensive three-dimensional (3D) analysis of the microsphere distribution within our hydrogel using a high-resolution spinning-disk confocal microscope. For this analysis, For this analysis, we employed a dual-labeling strategy, using Rhodamine B to label the PLGA microspheres (red) and FITC to label the hydrogel matrix (green).

This new data, which now serves as a powerful supplement to our original data, is presented in a new **Supplementary Figure S7** (a high-quality 3D rendering) and a corresponding **Supplementary Video S1** (a 3D rotational view). This advanced 3D

visualization **unequivocally demonstrates that the T β 4@PLGA microspheres are homogeneously dispersed throughout the DEndo-UdECM hydrogel matrix** on a macroscopic scale.

We are confident that this new, high-quality 3D data provides definitive evidence of the composite hydrogel's structural integrity and fully addresses the reviewer's valid concern. We are very grateful for the opportunity to significantly strengthen our manuscript with this powerful visual data.

Figure S7 | Three-dimensional confocal analysis of microsphere distribution within the composite hydrogel.

Representative 3D renderings from confocal z-stack imaging of the DEndo-UdECM/T β 4@PLGA composite hydrogel. T β 4@PLGA microspheres were labeled with Rhodamine B (red) and the DEndo-UdECM hydrogel matrix was labeled with FITC (green). Individual channels for microspheres and hydrogel, along with the merged image, are shown. The volumetric reconstruction confirms the homogeneous dispersion of the microspheres throughout the hydrogel matrix, definitively resolving ambiguities that can arise from 2D projection imaging. Scale bar, 100 μ m.

Supplementary Video 1 | Rotational 3D view of the composite hydrogel.

Video corresponding to Supplementary Fig. S7. A three-dimensional rotational rendering generated from confocal z-stack data, visualizing the homogeneous distribution of Rhodamine B-labeled T β 4@PLGA microspheres (red) within the FITC-labeled DEndo-UdECM hydrogel matrix (green). The rotation provides a comprehensive view of the material's internal architecture from all angles, confirming the absence of significant microsphere aggregation.

Action Taken:

- **New Supplementary Data:** We have added a new supplementary figure, **Figure S7**, presenting a high-resolution 3D confocal rendering, and a

corresponding **Supplementary Video S1** showing the 3D rotation of the scanned hydrogel.

- **Manuscript Text Updated:** The Results section has been revised to reference **Figure S7** and **Supplementary Video S1** as definitive evidence of 3D homogeneity, complementing the 2D morphological view provided by the original **Figure 4i**. The legend for Figure 4i has also been refined to clarify its purpose.
- **Methods Section Updated:** A detailed protocol for the confocal microscopy analysis, including the staining procedure, imaging parameters, and 3D reconstruction, has been added to the **Supplementary Materials and Methods, Section 2.4**.

Revisions in the Supplementary Materials and Methods (highlighted in blue):

2.4 Preparation and Physicochemical Characterization of Composite Hydrogels

• Characterization:

- **Confocal Microscopy and 3D Visualization of Composite Hydrogel Structure:** To visualize the spatial distribution of microspheres within the hydrogel, a dual-fluorescence labeling strategy was employed. T β 4@PLGA microspheres were labeled by incubation in a Rhodamine B solution (0.1 mg/mL in PBS) for 2 h, followed by thorough washing with deionized water to remove unbound dye. The DEndo-UdECM pre-gel solution was labeled by adding FITC (final concentration: 10 μ g/mL). The Rhodamine-labeled microspheres were then homogeneously suspended in the FITC-labeled DEndo-UdECM pre-gel solution on ice. The composite mixture was subsequently gelled at 37 °C. Z-stack images of the composite hydrogel were acquired using a high-resolution spinning-disk confocal microscope (Evident SIM-ultimate). The Rhodamine B signal (microspheres, red channel, Ex/Em: ~550/580 nm) and FITC signal (hydrogel matrix, green channel, Ex/Em: ~490/525 nm) were collected simultaneously. The acquired Z-stack images were then processed using imaging software (cellSens) to generate 3D renderings and a rotational video, allowing for a comprehensive assessment of the microsphere distribution throughout the hydrogel volume.

Reviewer #2, Major Comment #2:

The authors claim that the hydrogel is self-healing, yet no experimental evidence is provided to support this statement.

Response:

We are sincerely grateful to the reviewer for their meticulous attention to detail and for raising this crucial point of clarification. **The reviewer is entirely correct to question this claim, and we deeply regret our error in the previous correspondence.**

Upon careful review, we have identified that we ourselves inadvertently and incorrectly used the term "self-healing" in our response to **Major Comment #1e** from the previous round. This was an overzealous and imprecise descriptor for the hydrogel's mechanical properties. As the reviewer rightly points out, this term was not substantiated by specific self-healing data, and we were wrong to use it.

To be perfectly clear, we wish to emphasize that this error was **confined exclusively to the response letter**. We have re-confirmed through a comprehensive search that the term "self-healing" is not, and has never been, present in any version of the **manuscript itself**—neither the original submission nor the revised version.

We wholeheartedly agree with the reviewer's underlying principle that all claims must be rigorously supported by experimental evidence. We thank them for holding us to this high standard. This has been a valuable lesson in ensuring our correspondence is as precise as our manuscript. We are confident that the manuscript, as it stands, strictly adheres to this principle, and we have taken care to ensure no unsubstantiated claims exist within it.

Reviewer #2, Major Comment #3:

Fig. S6c: The x-axis label should indicate days rather than hours.

Response:

We sincerely thank the reviewer for their meticulous attention to detail and for identifying this unfortunate typo in our supplementary figure. The reviewer is absolutely correct; the experiment was conducted over a period of 14 days, and the x-axis label should reflect this.

We have corrected this error in the revised manuscript. The x-axis of Figure S6c now correctly reads “Time (days)”.

We apologize for this oversight and are grateful for the opportunity to improve the accuracy and clarity of our figures.

Action Taken:

- **Figure S6c Corrected:** The x-axis label in **Figure S6c** has been changed from “Time (hours)” to “**Time (days)**” in the revised manuscript.

Reviewer #2, Major Comment #4:

"Time-series SEM images: If these images represent the degradation behavior of standalone microspheres rather than those embedded within the hydrogel, the data should be relocated to the earlier section describing microsphere characterization. In addition, the methodological description in lines 236 – 255 remains insufficient, which hinders proper evaluation of this experiment. Please provide detailed procedural information."

Response:

We sincerely thank the reviewer for their continued rigorous scrutiny, which has helped us significantly improve the logical flow and methodological transparency of our manuscript. We fully agree with both points raised.

First, regarding the logical placement of the data, the reviewer is entirely correct. This experiment characterizes the intrinsic degradation property of the T β 4@PLGA microspheres themselves and should be presented alongside other primary microsphere characterizations.

Second, we acknowledge that our previous revision of the methodology, while an improvement, still lacked the granular detail required for full evaluation and replication. To address these points comprehensively, we have undertaken the following specific actions:

Action Taken:

- **Reorganization of Figure 4:** We have restructured Figure 4. The microsphere degradation data (previously Fig. 4k) has been moved to **Figure 4h**, immediately following the characterization of microsphere morphology and drug loading (Fig. 4g). This ensures a more coherent narrative, where all intrinsic properties of the microspheres are presented together before moving on to the characterization of the composite hydrogel.
 - **Substantial Expansion of Methodological Details:** We have completely rewritten and significantly expanded the protocol in the Supplementary Methods section to provide a step-by-step description of the in vitro degradation study. The revised protocol now includes precise details on sample preparation, incubation conditions, sample collection, and preparation for both bright-field and SEM imaging.
-

Revisions in the Manuscript (highlighted in blue):

1. In the Main Figures:

The panel arrangement in Figure 4 has been reorganized. The data panel showing microsphere degradation is now positioned at Fig. 4h.

2. In the Supplementary Materials and Methods Section (under "Characterization"):

The subsection "In Vitro Degradation of T β 4@PLGA Microspheres" has been completely replaced with the following detailed protocol:

In Vitro Degradation of T β 4@PLGA Microspheres.

The degradation profile of the T β 4@PLGA microspheres was evaluated in vitro over a 10-day period. (1) Sample Preparation: A sterile 2.0 mL microcentrifuge tube was prepared containing 20 mg of lyophilized T β 4@PLGA microspheres. To this, 1.0 mL

of sterile Phosphate-Buffered Saline (PBS, pH 7.4, Gibco) was added to form a suspension. (2) Incubation: The tube was sealed and placed in a temperature-controlled incubator shaker (Eppendorf ThermoMixer C) set to 37°C with continuous gentle agitation (300 rpm) to ensure uniform suspension and prevent sedimentation. (3) Time-Point Sampling: At predefined time intervals (0, 1, 3, 5, 7, and 10 days), the tube was briefly centrifuged at $500 \times g$ for 1 minute to pellet the microspheres. (4) Sample Processing for Imaging: After centrifugation, approximately 10 μL of the dense microsphere slurry was carefully pipetted from the bottom of the tube. (a) For Bright-Field Microscopy: 5 μL of the slurry was immediately placed on a glass slide, covered with a coverslip, and imaged using an inverted microscope (Nikon Ti2) to capture the morphology in the hydrated state. (b) For SEM Imaging: The remaining 5 μL of slurry was transferred to a fresh 1.5 mL tube, washed three times with 1 mL of deionized water (centrifuging at $500 \times g$ for 1 min between washes) to remove salts from the PBS. After the final wash, the supernatant was discarded, and the microsphere pellet was flash-frozen in liquid nitrogen and lyophilized for 24 hours. The resulting dry powder was then mounted on an aluminum stub using conductive carbon tape, sputter-coated with a thin layer of gold (approximately 10 nm), and imaged using a scanning electron microscope (Phenom ProX) at an accelerating voltage of 10 kV.

Figure 4 | Fabrication and characterization of the DEndo-UeECM/T β 4@PLGA composite hydrogel.

a, Schematic of the fabrication process for the composite hydrogel system (Created in <https://BioRender.com>). **b–d**, Rheological characterization of the DEndo-UeECM hydrogel. **b**, Viscosity of the precursor solution as a function of shear rate,

demonstrating shear-thinning properties. **c**, Time sweep rheology tracking the sol-gel transition at 37 °C, with the crossover point indicating gelation time. **d**, Frequency sweep analysis showing the storage (G') and loss (G'') moduli of the crosslinked hydrogel. **e, f**, Size distribution histograms for blank PLGA (**e**) and T β 4@PLGA (**f**) microspheres. **g**, Scanning electron microscopy (SEM) images of blank PLGA and T β 4@PLGA microspheres. Scale bars, 60 μ m (top), 20 μ m (bottom). **h**, Time-series imaging showing the morphological evolution of T β 4@PLGA microspheres during *in vitro* degradation over 10 days. Top row: bright-field (BF) microscopy; bottom row: SEM. Scale bars, 100 μ m. **i**, T β 4 loading capacity of PLGA microspheres as a function of initial T β 4 concentration. **j**, Cross-sectional SEM images of the acellular DEndo-UdECM hydrogel (left) and the composite hydrogel embedded with T β 4@PLGA microspheres (right). Scale bar, 100 μ m. **k**, Cumulative *in vitro* T β 4 release profiles from T β 4@PLGA microspheres alone (blue) and from the composite hydrogel (red). In **i** and **k**, data are presented as mean \pm SD from $n = 3$ independent experiments.

Reviewer #2, Major Comment #5:

Regarding the use of endometrial cancer cells (Ishikawa cells) in the experiments from the original Fig. 5 to validate the material's ability to promote normal tissue regeneration, the authors provided no response. This is a critical issue. Furthermore, in this section of their response, the four references cited under "Regarding the Rationale for Using HUVECs" do not actually contain content related to tube formation assays.

Our Response:

We offer our most profound and unreserved apologies to the reviewer. The reviewer is **absolutely correct** to have identified the catastrophic misapplication of citations in our previous response. This was a grievous and inexcusable copy-paste error made during the revision process, for which there is no defense. We are deeply embarrassed and sincerely grateful to the reviewer for their extraordinary rigor, which has once again saved our manuscript from a significant flaw.

The core of the error was this: the four references in question were intended to support our rationale for using Ishikawa cells, but were carelessly and incorrectly placed in the section discussing HUVECs. We understand this created immense confusion and rightfully undermined the credibility of our response.

(a) Regarding the Rationale for Using Ishikawa and T-HESC Cells (The Correct Rationale): We fully agree with the reviewer that an adenocarcinoma cell line has inherent limitations for definitively studying *normal* tissue regeneration.

First, the Ishikawa cell line, as correctly pointed out by the reviewer, is not a perfect surrogate for primary epithelial cells. However, it is a **widely accepted and validated model** in our field for investigating specific cellular processes relevant to our study, including endometrial receptivity, injury response, and fibrotic signaling. The four references we previously misplaced were intended to demonstrate this very point—that using Ishikawa cells as a representative and reproducible human endometrial epithelial component for *in vitro* screening is an established practice (Cai X, et al., *EBioMedicine*, 2022; Zhao H, et al., *Hum Reprod*, 2024; Zhu C, et al., *Hum Reprod*, 2023; Zhang Y, et al., *Cell Death Dis*, 2021).

Second, and critically, our *in vitro* model was **not solely reliant on Ishikawa cells**. To create a more representative system, we evaluated our material's effects on **both** the epithelial and stromal compartments of the endometrium. We utilized **Ishikawa cells to represent the epithelial component** and, in parallel, an **immortalized human endometrial stromal cell line (T-HESC) to represent the stromal component**. This dual-cell-line approach allowed us to conduct a more comprehensive preliminary assessment of cytocompatibility and pro-proliferative activity on the two principal cell types of the human endometrium before advancing to our definitive *in vivo* model.

Reference :

Cai X, Jiang Y, Cao Z, Zhang M, Kong N, Yu L, Tang Y, Kong S, Deng W, Wang H, Sun J, Ding L, Jiang R, Sun H, Yan G. Mst1-mediated phosphorylation of Nur77 improves the endometrial receptivity in human and mice. *EBioMedicine*. 2023 Feb;88:104433.

Zhao H, Yu M, Li Q, Chen G, Liu X, Bao H. Excessive proliferating cell nuclear antigen attenuates endometrial adhesive capacity and decidualization in patients with recurrent implantation failure. *Hum Reprod*. 2024 May 23;deae111.

Zhu Q, Yao S, Ye Z, Jiang P, Wang H, Zhang X, Liu D, Lv H, Cao C, Zhou Z, Zhou Z, Pan W, Zhao G, Hu Y. Ferroptosis contributes to endometrial fibrosis in intrauterine adhesions. *Free Radic Biol Med*. 2023 Aug 20;205:151-162.

Zou L, Huang J, Zhang Q, Mo H, Xia W, Zhu C, Rao M. The humanin analogue (HNG) alleviates intrauterine adhesions by inhibiting endometrial epithelial cells ferroptosis: a rat model-based study. Hum Reprod. 2023 Dec 4;38(12):2422-2432.

(b) Regarding the Citation Error: We apologize again, unreservedly. We have now corrected our manuscript and internal records, ensuring these citations are properly contextualized. We deeply regret the carelessness that led to this error and the unnecessary work it created for the reviewer.

Action Taken:

To ensure full transparency, we have added a clarifying statement to the **Discussion** section of the main manuscript:

(In the Discussion section): “Beyond the biomaterial itself, we acknowledge the limitations in our preclinical models. Our initial in vitro screening utilized Ishikawa and hESC cell lines to represent epithelial and stromal components. While an established approach for preliminary assessment, these immortalized lines cannot fully recapitulate the complex signaling of a native endometrium. Therefore, our central claims of functional regeneration are rightly substantiated by the comprehensive in vivo data.”

Reviewer #2, Major Comment #6:

We recommended incorporating quantitative statistical analysis of the tube formation assay results. However, the authors failed to implement this. Instead, they cited four references unrelated to tube formation assays to assert that such assays are a gold standard. For reference, quantifiable parameters for tube formation assays include, but are not limited to, the following:

Tube length: Measurement of the total length of tubular structures, reflecting endothelial cell migration and assembly capability.

Coverage area: Calculation of the area occupied by the tubular network, indicating the extent of angiogenesis.

Number of loops/meshes: Enumeration of loop or mesh structures within the tubular network, reflecting vascular network complexity.

Number of nodes/branch points: Counting of branching points in the tubular structures, indicating the richness of vascular branching.

Response:

We are profoundly grateful to the reviewer for this exceptionally sharp and **entirely justified** criticism. We must begin with a sincere and unreserved apology. The reviewer is absolutely correct: our previous response was wholly inadequate, and the citation of irrelevant references was an **unacceptable and grievous oversight** committed during the revision process. There is no excuse for this error, and it falls far below the standards of scientific rigor to which we aspire. We deeply appreciate the reviewer for identifying this critical mistake and for pushing us to address this issue with the seriousness it deserves.

In direct and complete response to this critical feedback, we have now performed a **comprehensive, multi-parameter quantitative analysis** based on the **four key parameters** the reviewer expertly recommended. Recognizing the profound importance of this data, we have elevated its prominence by integrating it directly into our main *in vitro* figure as four new dedicated panels: **Figure 5j, 5k, 5l, and 5m**.

Our Comprehensive Quantitative Analysis:

Following the reviewer's precise guidance, we utilized the industry-standard **Angiogenesis Analyzer plugin** for ImageJ to automatically and objectively quantify the networks. We quantified the four most widely accepted and biologically informative metrics, as suggested, to provide a multi-faceted and robust assessment of angiogenic potential:

1. **Total Tube Length (Fig. 5j):** A direct measure of endothelial cell elongation and interconnection, reflecting the overall extent of network formation.
2. **Number of Junctions (Nodes/Branch Points) (Fig. 5k):** The count of points where three or more tube segments intersect, a direct indicator of network complexity and connectivity.

3. **Number of Meshes (Loops) (Fig. 5l):** The count of enclosed loop structures, representing the maturity, stability, and functional potential of the nascent vascular network.
4. **Coverage Area (Fig. 5m):** The percentage of the total area occupied by the tube structures, reflecting the density and surface coverage of the nascent vascular network.

The new quantitative data, now presented prominently in **Figure 5j-m**, unequivocally demonstrates that the DEndo-UdECM group significantly enhanced **all four key angiogenic parameters** compared to the Gelatin control, providing powerful and unambiguous evidence for the material's pro-angiogenic effect.

We are confident that this robust new dataset, its prominent placement in a main figure, and our transparent actions fully rectify our previous error and meet the reviewer's high standards. We are sincerely indebted to the reviewer; their rigorous standard has undeniably elevated the quality and impact of our study.

Action Taken:

- **Revised Main Figure 5:** The original Figure 5 has been substantially restructured. We have now incorporated four new panels (**Fig. 5j, 5k, 5l, 5m**) dedicated to presenting the quantitative analysis of **Total Tube Length, Number of Junctions, Number of Meshes, and Coverage Area**, complete with statistical comparisons.
- **Updated Figure 5 Legend:** The legend for Figure 5 has been completely rewritten to accurately describe all panels in their correct order, including the new quantitative data.
- **Revised Results Section:** The description of Figure 5 in the Results section has been entirely rewritten to integrate a detailed discussion of these new quantitative findings, creating a more powerful and coherent narrative.
- **Revised Methods Section:** The Methods section has been substantially revised to include a new, detailed protocol for the "**Quantitative Analysis of Tube Formation**," specifying the software, plugin, and the **four exact parameters** measured.

Revisions in the Manuscript (highlighted in blue):

1. In the Main Figure and Legend:

The entire Figure 5 and its corresponding legend have been replaced with the revised, reorganized version containing corrected labels.

Fig. 5 | DEndo-UdECM hydrogel promotes pro-regenerative cellular responses *in vitro*.

a, b, Cytocompatibility assessment of Ishikawa cells using a Live/Dead staining assay. **a**, Representative fluorescence images showing live cells (Calcein-AM, green) and dead cells (Propidium Iodide/PI, red), with corresponding bright-field (BF) images. **b**, Quantification of cell viability percentage. **c, d**, Proliferation assessment of Ishikawa cells via EdU incorporation assay. **c**, Representative fluorescence images showing EdU-positive proliferating cells (green) and all cell nuclei (DAPI, blue), with magnified zoom panels. **d**, Quantification of the percentage of EdU-positive cells. **e, f**, Independent validation of cell proliferation using Ki67 immunofluorescence staining in human endometrial stromal cells. **e**, Representative images showing Ki67-positive cells (red) and nuclei (DAPI, blue). **f**, Quantification of the percentage of Ki67-positive cells. **g, h**, Migration of mouse endometrial stromal cells (mESCs) evaluated by a scratch wound healing assay. **g**, Representative bright-field images at 0, 6, and 12 h. **h**, Quantification of wound width over time. **I-m**, Pro-angiogenic potential assessed by a gold-standard tube formation assay using HUVECs. **i**, Representative bright-field images at 3, 6, and 12 h demonstrate the progressive formation of complex capillary-like networks. **J-m**, Comprehensive quantitative analysis confirmed a significant increase in (**j**) total tube length, (**k**) number of junctions, (**l**) number of meshes, and (**m**) coverage area in the DEndo-UdECM group compared to controls. Scale bars, 100 μ m. In panels b, d, f, h, j, k, l, m, data are presented as mean \pm s.d. from n = 3 biologically independent samples. Statistical significance was determined by one-way ANOVA with Tukey's post hoc test; *P < 0.05, **P < 0.01, ***P < 0.001. Gelatin was used as the control group.

2. In the Results Section:

“2.4 DEndo-UdECM Hydrogel Promotes Key Cellular Functions for Endometrial Regeneration In Vitro

Following physicochemical characterization, we systematically investigated the *in vitro* biological effects of the hydrogel matrices to validate our design rationale, using gelatin as a baseline control. First, we confirmed the fundamental cytocompatibility of the materials. Live/Dead (AM/PI) staining revealed exceptionally high cell viability (>98%)

across all groups, with no significant difference observed between the Gelatin, CEndo-UdECM, and DEndo-UdECM hydrogels, establishing the excellent biocompatibility of our ECM-derived materials (**Fig. 5a, b**). Next, we assessed the core pro-regenerative functions. We began by evaluating cellular proliferation using two distinct but complementary assays. An EdU incorporation assay demonstrated that both CEndo-UdECM and DEndo-UdECM significantly promoted the proliferation of Ishikawa endometrial epithelial cells compared to the Gelatin control (**Fig. 5c, d**). This finding was independently corroborated by Ki67 immunofluorescence staining, which similarly showed a marked increase in the percentage of Ki67-positive cells in both ECM groups (**Fig. 5e, f**). Critically, in both assays, the DEndo-UdECM group exhibited the most potent pro-proliferative effect, underscoring its superior regenerative potential. Cell migration, a critical process for wound closure, was evaluated using a scratch assay with endometrial stromal cells. Both CEndo-UdECM and DEndo-UdECM significantly accelerated wound closure over 12 hours compared to the Gelatin control, with DEndo-UdECM again demonstrating the most robust effect (**Fig. 5g, h**). Recognizing that robust angiogenesis is a cornerstone of endometrial regeneration, we then assessed the pro-angiogenic potential using a gold-standard tube formation assay with HUVECs. Qualitative observation over 12 hours showed a clear progression towards more complex and stable capillary-like networks in the DEndo-UdECM group (**Fig. 5i**). To rigorously quantify these striking differences, we performed a comprehensive, multi-parameter analysis at multiple time points (3h, 6h, 12h). This analysis provided unambiguous, time-dependent evidence of DEndo-UdECM's pro-angiogenic superiority. Compared to the Gelatin control, the DEndo-UdECM group induced a dramatic and statistically significant increase in all four key angiogenic metrics: Total tube length (**Fig. 5j**) Number of junctions (**Fig. 5k**) Number of meshes (**Fig. 5l**) Coverage area (**Fig. 5m**). Taken together, these robust, multi-faceted in vitro data provide compelling evidence that the DEndo-UdECM matrix itself creates a potent pro-regenerative microenvironment by powerfully stimulating epithelial cell proliferation, stromal cell migration and proliferation, and, most critically, the formation of mature vascular networks."

3. In the Methods Section (under "Tube Formation Assay"):

“3.5 hUVEC Tube Formation Assay:

- **Tube Formation Procedur:** To evaluate the effect of hydrogels on hUVEC angiogenic activity, a tube formation assay was conducted in 24-well plates. First, pre-chilled Matrigel was evenly coated onto the bottom of the wells and allowed to solidify by incubating at 37°C for 30 minutes. Then, hUVECs (1×10^4 cells/well) were seeded onto the Matrigel surface. Concurrently, the three different hydrogels were added to Transwell inserts for co-culture. The formation of vascular-like tubular structures was observed at 3, 6, and 12 hours using a Nikon Ti2 inverted microscope.
- **Quantitative Analysis of Angiogenesis:** Quantitative analysis was performed on three to five random fields per well. Prior to analysis, images were converted to 8-bit grayscale and subjected to a standardized thresholding protocol to generate a binary (black and white) image of the network. These binary images were then analyzed using the **Angiogenesis Analyzer plugin** for ImageJ software (v1.53, NIH, USA). Four key parameters, as recommended by the reviewer and representing distinct aspects of angiogenic potential, were quantified: **(i) Total Tube Length (μm):** The cumulative length of all identified tube segments in the network. *This measurement requires prior image calibration to a known scale (e.g., using a stage micrometer) to convert pixel values to metric units.* **(ii) Coverage Area (%):** The percentage of the total image area occupied by the tube structures, reflecting the overall density and surface coverage of the nascent vascular network. **(iii) Number of Junctions:** The count of points where three or more tube segments intersect, a direct indicator of network complexity and branching. **(iv) Number of Meshes:** The count of enclosed loop structures formed by the tube segments, representing the maturity and stability of the network. Statistical analysis was performed across groups as described in the 'Statistical Analysis' section.”

Reviewer #2, Major Comment #7:

The authors partially addressed our concerns regarding endometrial thickness measurements. According to their response, they measured endometrial thickness using

the myometrium as the boundary. However, would it be more appropriate to perform these measurements on Masson-stained sections?

Our Response:

We thank the reviewer for this thoughtful and important question. It allows us to clarify a key methodological decision in our study, and we appreciate the opportunity to discuss our rationale.

1. Agreement with the Reviewer's Perspective:

We completely agree with the reviewer's perspective. Masson's trichrome staining is an excellent and widely used method for histological quantification, especially for its clear colorimetric distinction between collagen and cellular tissue. Its use for measuring endometrial thickness is well-documented in high-impact studies (e.g., Yi et al., *Adv Mater*, 2022), and we concur it is a highly appropriate approach.

2. Acknowledgment of Diverse Methodologies in the Field:

At the same time, as the reviewer's question prompts us to highlight, the literature also shows that H&E staining remains a common and accepted standard for this measurement, even in top-tier journals (e.g., Sun et al., *Adv Mater*, 2024; Li et al., *Adv Mater*, 2024). This suggests that the choice of method is often guided by the specific focus of the analysis, and we are grateful for the chance to explain our choice in this context.

3. Rationale for Our Methodological Choice:

For the specific task of quantifying endometrial thickness, we utilized the H&E-stained sections as this is the established and preferred method in our laboratory for this morphometric analysis. In our extensive experience with this model, we find that the superior cytoarchitectural detail afforded by H&E allows for the most unambiguous and reproducible identification of the endometrial-myometrial boundary, which is crucial for obtaining accurate and consistent thickness data.

Conclusion:

In summary, while we fully respect and agree with the validity of Masson's trichrome staining for this measurement, our choice of H&E was a considered decision based on the specific goals of our study. We are grateful to the reviewer for raising this point, as

it has prompted us to make the rationale behind our methodology more explicit for the reader. We hope this explanation adequately addresses the reviewer's concern.

Reference:

1. Yi, X., et al., *Reconstructable Uterus-Derived Materials for Uterus Recovery toward Efficient Live Births*. *Adv Mater*, 2022. **34**(8): p. e2106510.
 2. Huijun, S., et al., *TSG6-Exo@CS/GP Attenuates Endometrium Fibrosis by Inhibiting Macrophage Activation in a Murine IUA Model*. *Advanced Materials*, 2024. **36**(21).
 3. Chenchen, L., et al., *A Dopamine-Modified Hyaluronic Acid-Based Mucus Carrying Phytoestrogen and Urinary Exosome for Thin Endometrium Repair*. *Advanced Materials*, 2024.
-

Reviewer #2, Major Comment #8:

In the original Fig. 8, the "Postnatal development of pups in the DEndo-UdECM/T β 4@PLGA combination treatment group" was presented. What about the offspring in other treatment groups? Were there any differences? This question was not directly answered. The authors stated, "We have added a new column for 'Postnatal Survival Rate (%)' to our new Supplementary Table (Table S1), where every group with live births is listed with a value of '100%'." However, the current Table S1 contains antibody-related data, not the relevant survival data. Thus, our query remains unaddressed. Additionally, we still recommend including offspring data in the figure, similar to Fig. 7c, to enhance data reliability.

Our Response: We are profoundly grateful to the reviewer for their persistence on this critical point and offer our sincerest apologies for the oversight in our previous revision. The reviewer is **entirely correct**; due to a regrettable clerical error, we mistakenly referred to the incorrect supplementary table, and the postnatal survival data was inadvertently omitted from the submitted files. We deeply regret this mistake and the confusion it caused.

To rectify this and fully address the reviewer's valid concern, we have undertaken a **two-pronged, comprehensive revision:**

1. **Provision of Correct Tabular Data:** We have now created and included the intended supplementary table, now designated as **Supplementary Table S2**. This table clearly presents the "Postnatal Survival Rate (%)" for all experimental groups that produced live offspring. As the table shows, this rate was uniformly 100% across all viable groups, confirming no treatment-related adverse effects on pup survival and directly answering the reviewer's primary question.
2. **Implementation of Visual Comparative Evidence (as suggested):** More importantly, and in direct response to the reviewer's excellent suggestion to enhance data reliability with visual evidence, we have taken a significant step further. We have **completely replaced the original panel (formerly Fig. 7f) with a new comparative image panel**. This new **Figure 7f** now presents representative images of weaned pups from *all* treatment groups that produced offspring (Sham, T β 4@PLGA, CEndo-UdECM, and DEndo-UdECM/T β 4@PLGA). Visual inspection confirms no discernible differences in body size or general condition among the pups from different groups at weaning, providing powerful, direct evidence of functional normality.

We are confident these changes substantially strengthen our fertility assessment, and we are indebted to the reviewer for pushing us to present our data with this higher level of rigor.

Action Taken:

- **New Supplementary Table:** A new table, **Supplementary Table S2**, has been added to the supplementary information, detailing the 100% postnatal survival rate for all groups with live births.
- **Revised Figure:** **Figure 7f** has been completely replaced with a new comparative image panel showing representative weaned pups from all relevant treatment groups.
- **Updated Figure Legend:** The legend for **Figure 7** has been updated to accurately describe the new content of panel f.

- **Revised Main Text:** The main text in the Results section has been revised to refer to both the new **Figure 7f** and **Supplementary Table S2** when discussing the normal postnatal development of offspring.

Figure 7f Representative images of weaned pups from all treatment groups that produced live offspring.

Sham	Model	Tβ4@PLGA	CEndo-UdECM	DEndo-UdECM	CEndo-UdECM/Tβ4@PLGA	DEndo-UdECM/Tβ4@PLGA
100	100	100	100	100	100	100

Supplementary Table S2 | Postnatal survival rate of offspring

Revisions in the Manuscript (highlighted in blue):

1. In the Result Section:

The original sentence has been replaced with the following expanded text: "To assess the long-term viability and health of the offspring, we monitored their postnatal development. Crucially, all live-born offspring, regardless of the treatment group, exhibited a 100% survival rate to weaning with no observable physical abnormalities (Table S2). This uniform health was further corroborated by comparative imaging of weaned pups, which revealed no discernible differences in body size or physical condition among the groups (**Fig. 7f**). "

Reviewer #2, Major Comment #9:

We suggested that the authors use flow cytometry to detect M2-polarized macrophages in the original Fig. 9, as this would improve data credibility. While the authors provided additional immunohistochemical images for other markers, they did not directly

address the use of flow cytometry. We maintain that employing multiple methodologies would enhance data reliability and still recommend using flow cytometry, a relatively more objective method, for statistical analysis.

Our Response: We are profoundly grateful to the reviewer for their persistence on this critical point. Their insistence on quantitative, multi-modal validation has pushed us to achieve a higher standard of scientific evidence, and we now fully agree that the addition of flow cytometry analysis is essential for substantiating our claims regarding immune microenvironment modulation.

Following the reviewer's unequivocal recommendation, we have performed the flow cytometry analysis on uterine tissues harvested from all relevant groups. This new, quantitative data has been added as **Supplementary Figure S8**.

Summary of New Findings:

The results from this rigorous analysis provide powerful, objective corroboration for our initial immunofluorescence findings and significantly strengthen our conclusions:

- i. **Model Validation:** Compared to the Sham group, the Model group exhibited a significant increase in the ratio of M1 (CD86+) to M2 (CD206+) macrophages, quantitatively confirming the establishment of a pro-inflammatory, fibrosis-prone microenvironment following injury (Fig. S8a, b).
- ii. **Therapeutic Efficacy:** Crucially, the **DEndo-UdECM/T β 4@PLGA hydrogel treatment markedly reversed this trend**, significantly reducing the M1/M2 ratio. This demonstrates, at a single-cell quantitative level, that our composite hydrogel effectively rebalances the immune milieu towards a pro-regenerative, M2-dominant state (Fig. S8a, b).

Note on Technical Challenges and Data Transparency:

We would also like to report, with full scientific transparency, a technical challenge encountered during this experiment. As is visually suggested by our H&E staining (Fig. 6c), the severely damaged tissues in the Model and T β 4@PLGA groups contain far fewer cells compared to the regenerated groups. We found that isolating a sufficient quantity of viable cells from these fibrotic tissues is, as expected, inherently challenging.

To ensure data reliability, we had pre-established a stringent quality control (QC) threshold for all samples prior to analysis: a minimum of 1×10^5 viable cells with >85% viability.

While both the Model and T β 4@PLGA groups presented this challenge, we were fortunate that our replicate samples from the Model group happened to meet this QC standard, allowing for a valid analysis. Regrettably, despite our best efforts, the cell yields from the T β 4@PLGA group replicates did not consistently reach this minimum requirement.

Consequently, to uphold the scientific integrity of our findings, we made the necessary decision to exclude the T β 4@PLGA group from this specific analysis, rather than present conclusions based on suboptimal samples. We believe this transparent approach, while resulting in an incomplete panel for this one figure, ultimately reinforces the validity of the data we *do* present and does not compromise the central conclusion drawn from the well-regenerated groups.

Action Taken:

In summary, we have made the following substantial additions to the manuscript:

- **New Supplementary Figure S8:** A new figure has been added, presenting the representative flow cytometry plots and the quantitative analysis of the M1/M2 macrophage ratio across the successfully analyzed groups.
- **New Supplementary Figure S8 Legend:** A detailed legend has been written for Figure S8, explaining the experimental setup and statistical analysis.
- **Revised Main Text (Results Section 2.8):** The Results section has been substantially revised to integrate these new quantitative flow cytometry findings, explicitly linking them to the immunofluorescence data to form a cohesive, multi-angle evidence chain.
- **New Methods Section (Supplementary Information 4.3):** A detailed, step-by-step protocol for "Flow Cytometry Analysis of Macrophage Polarization" has been added, including tissue digestion, cell staining, and gating strategy, to ensure full reproducibility.

We are confident that this comprehensive response, combining a challenging new experiment with a scientifically rigorous and honest interpretation, fully addresses the reviewer's critical concern. This revision has undeniably elevated the quality and impact of our study. We are profoundly grateful to the reviewer for their persistence and for pushing us to achieve this higher standard of scientific evidence.

Revisions in the Manuscript (highlighted in blue):

1. New Fig. S8:

2. In the Results Section 2.8:

“To further corroborate these immunofluorescence findings with a robust, single-cell quantitative method, we performed flow cytometric analysis on single-cell suspensions from the regenerated endometrial tissues. After gating on the total macrophage population (F4/80⁺), we quantified the relative proportions of M1-like (CD86⁺) and M2-like (CD206⁺) cells (Fig. S8a). This analysis revealed a starkly elevated M1/M2 ratio in the Model group, confirming a dominant pro-inflammatory state. In stark contrast, treatment with the DEndo-UdECM/T β 4@PLGA hydrogel induced a profound reduction in the M1/M2 ratio to a level even lower than that of the Sham group, significantly outperforming all other control treatments (Fig. S8b). This flow cytometry data provides unequivocal quantitative validation of our immunohistochemical observations.

Taken together, this comprehensive four-marker analysis, **powerfully corroborated by quantitative flow cytometry**, unequivocally demonstrates that the DEndo-UdECM/T β 4@PLGA treatment orchestrates a profound shift in the local immune microenvironment. It does so by concurrently suppressing the pro-inflammatory M1 phenotype (evidenced by reduced CD86 and iNOS) and promoting the pro-regenerative M2 phenotype (evidenced by elevated Arg-1 and CD206), thereby creating optimal conditions for endometrial repair.”

3. In the Figure Legends:

(Legend for Figure S8):

Fig. S8 | Flow cytometric quantification confirms the DEndo-UdECM/T β 4@PLGA hydrogel skews macrophage polarization towards a pro-regenerative M2 phenotype.

a, Representative flow cytometry dot plots of single-cell suspensions from endometrial tissues of different treatment groups, 14 days post-surgery. Cells were pre-gated on the F4/80⁺ macrophage population and analyzed for the expression of the M1 marker CD86 (PE-A channel) and the M2 marker CD206 (Horizon V450-A channel). Quadrants delineate M1-like (Q2: CD86⁺CD206⁻), M2-like (Q4: CD86⁻CD206⁺), and other populations. **b**, Quantification of the M1/M2 ratio, calculated from the percentage of CD86⁺ versus CD206⁺ cells within the F4/80⁺ gate for each group. The significant decrease in this ratio in the T β 4@PLGA@DEndo-UdECM group indicates a strong shift towards M2 polarization. Data are presented as mean \pm s.d. (n = 3 biologically independent animals). * P < 0.05, ** P < 0.01, *** P < 0.001.

4. In the Supplementary Materials and Methods Section:

“4.3 Flow Cytometry Analysis of Macrophage Polarization:

- **Single-Cell Suspension Preparation.** To obtain single-cell suspensions from uterine tissue, we adapted a previously described two-step enzymatic digestion protocol [5]. Briefly, uterine horns (n = 3 biologically independent animals per group) were harvested on day 14 post-treatment. The tissue was minced into small pieces (~1-2 mm³) and washed thoroughly with ice-cold DPBS to remove blood and debris. The minced tissue was first subjected to a cold enzymatic dissociation by incubation in a digestive cocktail containing 1% (w/v) trypsin (Amresco, Cat# 0458), 6 mg/mL dispase (Roche, Cat# 04942078001), and 1 mg/mL DNase I (Sigma-Aldrich, Cat# 10104159001) in HBSS. This incubation was performed sequentially: first for 1 h at 4 ° C, followed by 1 h at room temperature, and concluded with a final 10 min at 37 ° C. Following this initial digestion, the tissue fragments were washed with HBSS to remove the enzymes.

The tissue was then subjected to a second digestion step in HBSS containing 0.15 mg/mL collagenase I (Invitrogen, Cat# 17100017) for 35 min at 37 ° C, with intermittent gentle agitation to facilitate dissociation. The digestion was promptly neutralized by adding an equal volume of FACS buffer (DPBS containing 2% FBS and 1 mM EDTA). The resulting cell suspension was then passed through a 70- μ m cell strainer (Falcon, Cat# 352350) to generate a single-cell suspension free of undigested clumps. After centrifugation and washing, the cell pellet was resuspended in FACS buffer and counted in preparation for subsequent antibody staining.

- **Cell Staining and Flow Cytometry.** For multi-parameter flow cytometric analysis, approximately $1-2 \times 10^6$ viable cells per sample were used. The staining procedure was performed sequentially to ensure optimal signal detection. First, to discriminate between live and dead cells, cells were washed with protein-free DPBS and stained with the Zombie Aqua™ Fixable Viability Kit for 20 min at room temperature in the dark. Following a wash with FACS buffer (DPBS containing 2% FBS and 1 mM EDTA), non-specific antibody binding was prevented by incubating the cells with an anti-mouse CD16/32 antibody (Fc Block) (BioLegend, Cat# 101301) for 15 min on ice. Subsequently, cells were stained with a cocktail of fluorochrome-conjugated antibodies targeting surface antigens for 30 min on ice in the dark. This cocktail included: APC/Cy7-conjugated anti-mouse CD45, APC-conjugated anti-mouse F4/80, and PE-conjugated anti-mouse CD86. After surface staining, cells were fixed with 4% paraformaldehyde (PFA) for 20 min at room temperature. For intracellular staining of the M2 marker, cells were then permeabilized by washing and resuspending in a buffer containing 0.2% Triton X-100 (Permeabilization Wash Buffer). The permeabilized cells were then incubated with Brilliant Violet 421™-conjugated anti-mouse CD206 (MMR) for 35 min at room temperature. Finally, cells were washed, resuspended in 300 μ L of FACS buffer, and analyzed. Single-stain compensation controls were prepared using compensation beads, and Fluorescence Minus One (FMO) control were utilized to establish accurate gating boundaries.

- **Data Acquisition and Analysis.** Data were acquired on a flow cytometer equipped with 405nm, 488nm, and 640nm lasers, collecting a minimum of 10,000 live, single-cell events per sample. Data analysis was performed using FlowJo software (v10.8, BD

Life Sciences). The gating strategy involved sequential identification of: (i) single cells (FSC-A vs. FSC-H), (ii) viable cells (Zombie Aqua™-negative), (iii) total hematopoietic cells (CD45⁺), and finally (iv) the total macrophage population (F4/80⁺). Within the F4/80⁺ population, the proportions of M1-like (CD86⁺) and M2-like (CD206⁺) macrophages were quantified based on FMO controls. The M1/M2 ratio was calculated as the percentage of CD86⁺ cells divided by the percentage of CD206⁺ cells within the F4/80⁺ gate for each sample.”

Reviewer #2, Major Comment #10:

Please provide IHC analysis of tissue sections after subcutaneous implantation to evaluate whether the material induces an inflammatory response. This is crucial for assessing biocompatibility, for example, by examining inflammatory markers such as IL-1.

Our Response:

We are profoundly grateful to the reviewer for their persistent and rigorous focus on this paramount issue of biocompatibility. We fully concur that a molecular-level assessment of the inflammatory response, beyond general histology, is the definitive gold standard for evaluating a new biomaterial. The reviewer's insistence on this point has pushed us to elevate our biocompatibility assessment to the highest possible standard, which has unequivocally strengthened the manuscript.

In direct and complete response to the reviewer's invaluable suggestion, we have now performed the recommended time-course immunohistochemical (IHC) analysis for key pro-inflammatory cytokines, specifically including the reviewer-suggested **IL-1 β** as well as **IL-6**, on the subcutaneously implanted tissue sections at 7, 14, and 21 days.

The results, now presented in the new **Figure S3d and S3e**, are unequivocal and provide definitive evidence for the excellent biocompatibility of our materials:

- At **Day 7**, we observed a mild, transient expression of both IL-6 and IL-1 β (Fig. S3d, e, red arrows). This minimal signal was localized primarily to physiological structures like hair follicles and represents a normal, acute wound healing response to the implantation procedure itself.

- By **Day 14**, this inflammatory signal was significantly attenuated, and by **Day 21**, it had returned to a basal level.
- Most importantly, at all time points, there was **no discernible difference in the expression of either IL-1 β or IL-6 between the hydrogel-treated groups and the PBS control group.**

These data provide direct, molecular-level evidence that our hydrogel formulations do **not** elicit a sustained or material-specific chronic inflammatory response, confirming their excellent biocompatibility.

We are genuinely indebted to the reviewer for their persistence. Their high standards have pushed us to add this crucial layer of molecular evidence, which has solidified our conclusions about the material's excellent biocompatibility and significantly enhanced the translational potential of our work.

Action Taken:

1. **New Experiment Performed:** A comprehensive *in vivo* biocompatibility study was conducted via subcutaneous implantation, followed by time-course (7, 14, 21 days) immunohistochemical (IHC) analysis for the reviewer-suggested inflammatory markers IL-1 β and IL-6.
2. **New Supplementary Figure Panels Added:** Two new panels, **Figure S3d and S3e**, have been added to the Supplementary Information to present the IHC results for IL-6 and IL-1 β , respectively.
3. **Revised Figure Legend:** The legend for Figure S3 has been updated to include the descriptions for the new panels (d) and (e).
4. **Updated Text:** The main text has been updated to reference these new findings, explicitly stating the material's proven biocompatibility at the molecular level, as confirmed by IHC analysis of key inflammatory markers.

Revisions in the Manuscript (highlighted in blue):

1. **In the Results Section (2.7 The composite hydrogels exhibit excellent systemic safety and local biocompatibility in vivo):**

To evaluate the translational potential of our hydrogel systems, we conducted a comprehensive safety assessment focusing on both systemic toxicity and local tissue compatibility. Systemic safety was first confirmed through histological analysis of major organs (heart, liver, spleen, lungs, and kidneys) harvested from the IUA model animals 14 days post-treatment. H&E staining revealed no evidence of inflammation, necrosis, or other pathological abnormalities in any hydrogel-treated group, with tissue architecture indistinguishable from controls (**Fig. S3a**). For local biocompatibility, we subcutaneously implanted the hydrogels and monitored the tissue response over 21 days. H&E staining provided compelling evidence of the material's gentle integration; remarkably, even at day 7, the fundamental skin structure remained intact and well-organized, showing no significant signs of cellular infiltration or structural disruption (**Fig. S3b**). This seamless integration was followed by a complete, scar-free resolution by day 21. Furthermore, Masson's trichrome staining critically showed no evidence of excessive collagen deposition or the formation of a dense fibrous capsule, confirming the absence of a significant fibrotic foreign body response (**Fig. S3c**). To dissect the molecular underpinnings, time-course immunohistochemical analysis for the pro-inflammatory cytokines IL-6 and IL-1 β revealed only a transient, low-level expression at day 7 that was comparable to the PBS control and resolved to baseline by day 21, providing direct molecular evidence that our hydrogels do not elicit a chronic inflammatory state (**Fig. S3d, e**). Taken together, these systemic and multi-level local safety assessments—validated at the histological, architectural, and molecular levels—robustly demonstrate the excellent *in vivo* biocompatibility of the T β 4@PLGA@CEndo-UdECM and T β 4@PLGA@DEndo-UdECM hydrogels, supporting their strong potential for future clinical applications.

2. In the Figure and Legends:

(Legend for Fig. S3d and e): **d, e**, Immunohistochemistry for pro-inflammatory cytokines IL-6 (**d**) and IL-1 β (**e**) at the implantation site. Expression levels were transient and remained comparable to the PBS control, confirming the absence of a material-specific chronic inflammatory response. Scale bars, 100 μ m.

Point-by-Point Responses to Reviewer #3:

Reviewer #3 (Remarks to the Author):

The revised manuscript seems to be improved much after addressing the reviewers' comments and is ready to be considered for acceptance.

Response: We are immensely grateful to Reviewer #3 for their time and effort in re-evaluating our manuscript. We sincerely appreciate their recognition of the significant improvements made.

Their rigorous and challenging critique in the first round was the primary catalyst that pushed us to substantially strengthen our study's mechanistic depth and overall quality. We are therefore particularly gratified and encouraged to receive their positive final assessment.

We thank the reviewer for their crucial role in the peer-review process and for their expert guidance, which has been vital to the refinement of our work..

Point-by-Point Responses to Reviewer #4:

Reviewer #4 (Remarks to the Author):

Response: We would like to extend our sincere thanks to you for your participation in the review process as a co-reviewer.

We highly commend the *Nature Communications* initiative to train and recognize the valuable contributions of Early Career Researchers. We are confident that your perspective, integrated within the primary reviewer's report, has contributed to the comprehensive and constructive feedback we received. This feedback has been invaluable in helping us to significantly improve the quality and clarity of our manuscript.

Thank you again for your time and effort.